# The Implicit Bias of Steepest Descent with Mini-batch Stochastic Gradient

**Jichu Li** [1]  **Xuan Tang** [2]  **Difan Zou** [2 3]

## Abstract

A variety of widely used optimization methods like SignSGD and Muon can be interpreted as instances of steepest descent under different norm-induced geometries. In this work, we study the implicit bias of mini-batch stochastic steepest descent in multi-class classification, characterizing how batch size, momentum, and variance reduction shape the limiting max-margin behavior and convergence rates under general entry-wise and Schatten-$p$ norms. We show that, without momentum, worst-case convergence and successful classification can only be guaranteed with full-batch gradient. In contrast, momentum enables small-batch convergence to an approximate max-margin solution through a batch-momentum trade-off, though it slows convergence. This approach provides fully explicit, dimension-free rates that improve upon prior results. Moreover, we prove that variance reduction can recover the exact full-batch implicit bias for any batch size, albeit at a slower convergence rate. Finally, we further investigate the batch-size-one steepest descent without momentum, and reveal its convergence to a fundamentally different bias via a concrete data example, which reveals a key limitation of purely stochastic updates. Overall, our unified analysis clarifies when stochastic optimization aligns with full-batch behavior, and paves the way for perform deeper explorations of the training behavior of stochastic gradient steepest descent algorithms.

## 1  Introduction

In large-scale language model pretraining, the choice of optimizer plays a crucial role in training stability and efficiency, and directly impacts the performance of the final model. Among modern optimization algorithms, Adam and AdamW (Kingma, 2014; Loshchilov & Hutter, 2017) have become the de facto standard optimizers for large-scale pretraining. More recently, Muon (Jordan et al., 2024; Liu et al., 2025) has emerged as a promising alternative. Developing a theoretical understanding of these optimizers is therefore essential for improving large-scale training. Prior work has shown that Adam can exhibit sign-like update behavior and can be approximated by SignGD in certain settings (Bernstein et al., 2018; Balles & Hennig, 2018; Zou et al., 2021). Moreover, SignGD can be interpreted as steepest descent under $\ell_\infty$ norm constraint, while Muon corresponds to steepest descent under spectral norm constraint. These connections motivate our study of a general class of steepest descent methods induced by different norms.

The implicit bias of an optimization algorithm is fundamental to understanding its performance (Neyshabur et al., 2014; Soudry et al., 2018; Ji & Telgarsky, 2019). In over-parameterized learning settings, where the training objective admits infinitely many global minima, the implicit bias determines which solution is ultimately selected. Understanding this phenomenon has become central to explaining why over-parameterized models can achieve near-zero training loss while still exhibiting strong generalization performance.

The theoretical studies of the implicit bias have been extensively employed for full-batch gradient descent (GD) (Soudry et al., 2018; Nacson et al., 2019b; Ji & Telgarsky, 2019; 2020; Wu et al., 2023; Cai et al., 2025), which prove its tendency to certain $\ell_2$-norm regularized solutions. More recently, the implicit bias of steepest descent optimization algorithms has received increasing attention. To name a few, Zhang et al. (2024) showed that Adam converges to the $\ell_\infty$ max-margin solution in linear logistic regression, exhibiting an implicit bias similar to SignSGD. Tsilivis et al. (2024) showed that the iterates of steepest descent converge to a KKT point of a generalized margin maximization problem in homogeneous neural networks. Fan et al. (2025) analyzed a general class of full-batch steepest descent algorithms in multi-class linear classification and showed that steepest descent with entry-wise or Schatten-$p$ norms maximizes the margin w.r.t. the corresponding norm.

However, most existing theoretical analyses of implicit bias

[1] School of Statistics, Renmin University of China, Beijing, China [2] School of Computing and Data Science, The University of Hong Kong, Hong Kong [3] Institute of Data Science, The University of Hong Kong, Hong Kong. Correspondence to: Jichu Li <lijichu52@gmail.com>, Difan Zou <dzou@hku.hk>.

*Proceedings of the 43rd International Conference on Machine Learning*, Seoul, South Korea. PMLR 306, 2026. Copyright 2026 by the author(s).

focus on full-batch gradients, which differs substantially from practical training settings that rely on stochastic gradients. More importantly, optimization dynamics under full-batch updates can exhibit different behavior from their stochastic counterparts, leading to fundamentally different implicit biases. This distinction is supported by recent work analyzing per-sample Adam (Tang et al., 2025b; Baek et al., 2025). However, their analyses typically rely on proxy algorithms that differ from practical implementations, requiring access to full-batch information at each update. As a result, such approaches are difficult to extend to other steepest descent methods and offer limited insight into the implicit bias of practical optimization algorithms, including how algorithmic parameters such as batch size and momentum shape the induced bias.

In this paper, we consider a general class of steepest descent algorithms and study the multi-class classification problem with cross-entropy and exponential loss (Fan et al., 2025). In particular, we develop a unified framework that covers a range of well-known stochastic optimization methods such as SignSGD (Bernstein et al., 2018), Normalized-SGD (Hazan et al., 2015), Muon (Jordan et al., 2024), and their related variants. We investigate how different algorithmic parameter settings, including batch size and momentum, influence the implicit bias of steepest descent methods and characterize the corresponding convergence rates. Motivated by the central role of batch size, we further study the effect of incorporating variance reduction and analyze its impact on implicit bias. In addition, we consider the extreme batch-size-one setting without momentum using a specially constructed dataset. Finally, our experiments highlight the critical role of momentum and variance reduction in implicit bias convergence.

Our contributions can be summarized as follows:

- We first establish a large-batch condition under which stochastic normalized steepest descent without momentum guarantees worst-case convergence and successful classification. We then observe that this condition necessarily collapses to the full-batch-gradient regime, which implies the number of mini-batches per epoch $m = 1$, i.e., $b = n$. Moreover, we show that this restriction is not merely due to looseness of the condition by constructing a counterexample in which random-reshuffling normalized steepest descent without momentum fails when $m > 1$.

- We show that incorporating momentum enables mini-batch normalized steepest descent to converge to an approximate max margin $\rho < \gamma$, even in the small-batch regime. Specifically, momentum stabilizes mini-batch noise, resulting in an approximation margin gap $\gamma - \rho$ that vanishes as either the batch size increases or the momentum parameter $\beta_1 \to 1$, revealing an interplay between mini-batch size and momentum. Furthermore, our analysis removes the dimension dependence in the

rate developed in Fan et al. (2025), offering a technical improvement of independent interest.

- Beyond momentum, we prove that incorporating variance reduction ensures convergence to the same norm-induced max-margin solution as full-batch steepest descent, regardless of batch size and with or without momentum. However, this exactness presents a trade-off: the convergence rate is slower than that of standard stochastic methods without variance reduction.

- Finally, we explore the implicit bias of stochastic steepest descent with mini-batch size 1 on a specifically designed orthogonal data model. We show that the algorithm can converge to an implicit bias fundamentally different from that of full-batch methods. This discrepancy implies there may not exist a unified implicit bias theory covering steepest descent with small batch sizes, aligning with similar observations for per-sample Adam in (Baek et al., 2025).

**Notations.** Scalars, vectors, and matrices are denoted by $x$, $\mathbf{x}$, and $\mathbf{X}$, respectively, with $\mathbf{X}[i, j]$ and $\mathbf{x}[i]$ denoting their entries. For $k \in \mathbb{N}^+$, let $[k] = \{1, \ldots, k\}$. For sequences $\{a_t\}$ and $\{b_t\}$, we use $a_t = \mathcal{O}(b_t)$ if there exist constants $C, N > 0$ such that $a_t \leq Cb_t$ for all $t \geq N$. Throughout the paper, $\|\mathbf{X}\|$ denotes a matrix norm determined by context. The entrywise $p$-norm is $\|\mathbf{X}\|_p = (\sum_{i,j} |\mathbf{X}[i, j]|^p)^{1/p}$, and the Schatten $p$-norm is $\|\mathbf{X}\|_{S_p} = (\sum_{i=1}^r \sigma_i^p)^{1/p}$, where $\sigma_i$ is the singular values of $\mathbf{X}$ and $r = \text{rank}(\mathbf{X})$; special cases include $p = 1$ (nuclear), $p = 2$ (Frobenius), and $p = \infty$ (spectral). We use the standard matrix inner product $\langle \mathbf{A}, \mathbf{B} \rangle = \text{tr}(\mathbf{A}^\top \mathbf{B})$, with dual norm denoted by $\| \cdot \|_*$. Let $\mathbb{S} : \mathbb{R}^k \to \triangle^{k-1}$ denote the softmax map, and let $\{\mathbf{e}_c\}_{c=1}^k$ denote the standard basis of $\mathbb{R}^k$. *A complete list of notation and precise definitions is deferred to Appendix B.*

## 2 Related Work

**Steepest Descent Methods for Optimization.** A variety of optimization methods can be viewed as instances of steepest descent induced by different norms. Normalized-SGD (Hazan et al., 2015) corresponds to steepest descent under the $\ell_2$ norm, but its convergence typically requires either large batch sizes or low-variance gradient estimates. Cutkosky & Mehta (2020) showed that momentum can relax these requirements and improve convergence. Similarly, SignSGD and Signum (Bernstein et al., 2018) can be viewed as steepest descent under the $\ell_\infty$ norm. As with Normalized-SGD, SignSGD's convergence relies on large batch sizes or restrictive noise assumptions (Karimireddy et al., 2019), while subsequent work demonstrated that momentum substantially improves convergence guarantees under weaker conditions (Sun et al., 2023b; Jiang et al., 2025). Beyond entry-wise normalization, spectral-norm-based steepest descent methods have also been studied, alongside recent work on generalized $p$-norm regularization (Outmezguine & Levi,

2024). More recently, Jordan et al. (2024) proposed Muon, which can be viewed as steepest descent induced by the spectral norm. Its convergence properties have been studied and refined by a series of subsequent works (Shen et al., 2025; Chang et al., 2025; Sato et al., 2025; Tang et al., 2025a; Pethick et al., 2025; 2026). While convergence properties have been extensively studied, they are insufficient to theoretically understand steepest descent methods.

**Implicit Bias of Optimization Algorithms.** The implicit bias of gradient descent (GD) has been extensively studied. For linearly separable data, GD converges in direction to the $\ell_2$ max-margin solution for both linear models and networks (Soudry et al., 2018; Ji & Telgarsky, 2018; Gunasekar et al., 2018b; Nacson et al., 2019b), with extensions to stochastic settings (Nacson et al., 2019c). Beyond separability, GD's bias has also been characterized for nonseparable data (Ji & Telgarsky, 2019), under larger learning rates via primal-dual analysis (Ji & Telgarsky, 2021), and in the edge-of-stability regime (Wu et al., 2023). Implicit bias results further extend beyond linear models to homogeneous and non-homogeneous neural networks (Lyu & Li, 2019; Nacson et al., 2019a; Ji & Telgarsky, 2020; Cao et al., 2023; Kou et al., 2023; Cai et al., 2024; 2025). For steepest descent, Gunasekar et al. (2018a) characterized its implicit bias on separable linear models via margin maximization, and Nacson et al. (2019b) further analyzed the corresponding convergence rates. More recently, Tsilivis et al. (2024) proved steepest descent converges to a KKT point of a generalized margin maximization problem in homogeneous neural networks, while Fan et al. (2025) showed that steepest descent with entry-wise or Schatten-$p$ norms maximizes the corresponding norm-induced margin in multiclass linear classification. Motivated by the $\ell_\infty$ geometry underlying SignGD and its connection to Adam (Balles & Hennig, 2018; Zou et al., 2021), recent work studied the implicit bias of Adam/AdamW (Wang et al., 2021; Cattaneo et al., 2023) and its $\ell_\infty$-induced geometry (Zhang et al., 2024; Xie & Li, 2024). However, most existing analyses focus on full-batch optimization, and the implicit bias of mini-batch stochastic methods remains far less understood.

**Stochasticity in Margin Maximization.** Nacson et al. (2019c) shows that mini-batch SGD converges to the $\ell_2$ max-margin direction under a sufficiently small stepsize controlling stochastic noise (depending on batch size). Wang et al. (2022) further shows that SGD with momentum preserves the same $\ell_2$ max-margin direction under a small constant stepsize depending on both batch size and momentum. Jin et al. (2024) establishes a similar max-margin result for stochastic AdaGrad-Norm under general mini-batch noise assumption. Baek et al. (2025) shows that per-sample Adam can induce an implicit bias that differs from its full-batch counterpart and that Signum exhibits similar behavior in our paper under a restrictive fixed mini-batch cycle. In contrast, our work provides a unified analysis of stochastic normalized steepest descent (NSD) under general geometries and is, to the best of our knowledge, the first to systematically characterize how multiple stochastic factors, including batch size, momentum, and variance reduction, interact to shape the implicit bias.

**Variance Reduction Methods.** Variance reduction methods have been extensively studied in stochastic optimization for reducing gradient variance and accelerating convergence (Roux et al., 2012; Defazio et al., 2014; Gower et al., 2020). In this work, we focus on an SVRG-style estimator (Johnson & Zhang, 2013). To the best of our knowledge, its role in shaping implicit bias has not been explicitly studied. We show that variance reduction recovers the exact full-batch implicit bias regardless of batch size and momentum, while it may lead to a slower margin convergence rate.

## 3  Preliminaries

**Problem Setup.**  We consider a multi-class classification problem with training data $\{(\mathbf{x}_i, y_i)\}_{i=1}^n$, where each datapoint $\mathbf{x}_i \in \mathbb{R}^d$ lies in a $d$-dimensional embedding space and each label $y_i \in [k]$ denotes one of the $k$ classes; we assume that each class is represented by at least one datapoint. We consider a linear multi-class classifier parameterized by a weight matrix $\mathbf{W} \in \mathbb{R}^{k \times d}$. Given an input $\mathbf{x}_i \in \mathbb{R}^d$, the model produces logits $\boldsymbol{\ell}_i := \mathbf{W}\mathbf{x}_i \in \mathbb{R}^k$, which are passed through the softmax map to yield class probabilities $\hat{p}(c \mid \mathbf{x}_i) := \mathbb{S}_c(\boldsymbol{\ell}_i)$. We aim to learn $\mathbf{W}$ by minimizing the following empirical loss:

$$L(\mathbf{W}) := \frac{1}{n} \sum_{i=1}^n \ell\left(\mathbf{W}\mathbf{x}_i; y_i\right).$$

where $\ell\left(\mathbf{W}\mathbf{x}_i; y_i\right)$ is the loss function value on the data point $(\mathbf{x}_i, y_i)$. In the main text, we focus on the cross-entropy loss due to its widespread use in practice, while all of our theoretical results also extend to the exponential loss. (see Appendix I for details.) Specifically, the empirical cross-entropy loss is given by

$$L(\mathbf{W}) = -\frac{1}{n} \sum_{i=1}^n \log \hat{p}(y_i \mid \mathbf{x}_i) = -\frac{1}{n} \sum_{i=1}^n \log \mathbb{S}_{y_i}(\mathbf{W}\mathbf{x}_i).$$

Finally, we define the maximum margin $\gamma$ of the training data $\{(\mathbf{x}_i, y_i)\}_{i=1}^n$ under any entry-wise or Schatten $p$-norm $\|\cdot\|$ as:

$$\gamma := \max_{\|\mathbf{W}\| \leq 1} \min_{i \in [n], c \neq y_i} (\mathbf{e}_{y_i} - \mathbf{e}_c)^\top \mathbf{W}\mathbf{x}_i.$$

### 3.1  Optimization Algorithms

We study stochastic optimization algorithms for training linear multi-class classifiers under random shuffling (sampling

without replacement in each epoch). Our analysis covers both momentum-based and non-momentum methods, as well as their variance-reduced counterparts, under a unified steepest descent framework induced by general norms. For clarity, we provide the pseudocode of the algorithms in Appendix A.

**Mini-batch Stochastic Algorithm.** In this paper, we focus on stochastic optimization algorithms without replacement. Let $b$ denote the mini-batch size and assume for simplicity that the data size $n = mb$ for some integer $m$. At each epoch, the training set is uniformly randomly partitioned into $m$ disjoint mini-batches of size $b$. Then the algorithm performs $m$ consecutive updates, one for each mini-batch. Specifically, at the $k$-th epoch, let the mini-batch index sets be $\mathcal{B}_{k,0}, \ldots, \mathcal{B}_{k,m-1}$, where $|\mathcal{B}_{k,j}| = b$ and $\bigcup_{j=1}^{m} \mathcal{B}_{k,j} = \{1, 2, \ldots, n\}$. We index iterations by $t = km + j$, corresponding to the $j$-th mini-batch at epoch $k$. At iteration $t$, the stochastic gradient is computed as $\nabla L_{\mathcal{B}_t}(\mathbf{W}_t) := \frac{1}{b} \sum_{i \in \mathcal{B}_{k,j}} \nabla \ell(\mathbf{W}_t \mathbf{x}_i; y_i)$.

**Gradient Signal Construction.** Beyond the vanilla mini-batch stochastic gradient $\nabla L_{\mathcal{B}_t}(\mathbf{W}_t)$, we investigate more advanced gradient estimators that incorporate momentum acceleration and variance reduction techniques.

*Momentum.* We first consider the standard exponential moving average (EMA) momentum to stabilize the update direction. The momentum buffer $\mathbf{M}_t$ is updated as:

$$\mathbf{M}_t = \beta_1 \mathbf{M}_{t-1} + (1 - \beta_1) \nabla L_{\mathcal{B}_t}(\mathbf{W}_t),$$

where $\beta_1 \in [0, 1)$ is the momentum decay parameter.

*Variance Reduction.* We construct variance-reduced gradient estimators $\mathbf{V}_t$ using a control variate strategy similar to SVRG (Johnson & Zhang, 2013). Let $\tilde{\mathbf{W}}_t$ denote the *snapshot* of the model parameters taken at the beginning of the epoch to which iteration $t$ belongs. We compute the variance-reduced estimator as:

$$\mathbf{V}_t = \nabla L_{\mathcal{B}_t}(\mathbf{W}_t) - \nabla L_{\mathcal{B}_t}(\tilde{\mathbf{W}}_t) + \nabla L(\tilde{\mathbf{W}}_t),$$

We apply variance reduction in two settings: without momentum (using $\mathbf{V}_t$ directly) and with momentum. In the latter case, $\mathbf{V}_t$ is accumulated into the buffer $\mathbf{M}_t^V$ as

$$\mathbf{M}_t^V = \beta_1 \mathbf{M}_{t-1}^V + (1 - \beta_1) \mathbf{V}_t.$$

**Steepest Descent.** Building on the stochastic gradient signals constructed above, we unify all optimization algorithms considered in this work under a stochastic steepest descent framework. At each iteration $t$, the model parameters are updated as

$$\mathbf{W}_{t+1} = \mathbf{W}_t - \eta_t \mathbf{\Delta}_t,$$

where $\eta_t > 0$ is the step size and $\mathbf{\Delta}_t$ is a descent direction derived from a stochastic signal $\mathbf{G}_t$. Here, $\mathbf{G}_t$ may correspond to the mini-batch stochastic gradient $\nabla L_{\mathcal{B}_t}(\mathbf{W}_t)$, the momentum buffer $\mathbf{M}_t$, or their variance-reduced counterparts $\mathbf{V}_t$ and $\mathbf{M}_t^V$ defined above.

Given any entry-wise or Schatten $p$-norm $\| \cdot \|$, the descent direction is obtained by applying the associated steepest descent mapping

$$\mathbf{\Delta}_t := \phi_{\|\cdot\|}(\mathbf{G}_t) := \arg \max_{\|\mathbf{\Delta}\| \leq 1} \langle \mathbf{G}_t, \mathbf{\Delta} \rangle.$$

This operator selects, at each stochastic iteration, the unit-norm direction that maximally aligns with the current signal $\mathbf{G}_t$ under the geometry induced by $\| \cdot \|$. By norm duality, $\max_{\|\mathbf{\Delta}\| \leq 1} \langle \mathbf{G}_t, \mathbf{\Delta} \rangle = \|\mathbf{G}_t\|_*$. Note that for $p \in (1, \infty)$, the corresponding steepest descent direction is uniquely defined, while for $p = 1$ or $p = \infty$ the maximizer may not be unique and our theoretical results hold for any choice within the set of maximizers (Fan et al., 2025).

Different choices of the norm constraint recover a range of well-known stochastic optimization methods, depending on the choice of the driving signal $\mathbf{G}_t$. Under the Frobenius norm $\| \cdot \|_2$, the steepest descent map reduces to normalization, $\phi_2(\mathbf{G}_t) = \mathbf{G}_t/\|\mathbf{G}_t\|_2$, which yields Normalized-SGD (Hazan et al., 2015) when $\mathbf{G}_t = \nabla L_{\mathcal{B}_t}$ and its momentum variant (Cutkosky & Mehta, 2020) when $\mathbf{G}_t = \mathbf{M}_t$. For the max-norm $\| \cdot \|_\infty$, the map acts entry-wise as the sign operator, $\phi_\infty(\mathbf{G}_t) = \text{sign}(\mathbf{G}_t)$, recovering SignSGD when $\mathbf{G}_t = \nabla L_{\mathcal{B}_t}$ and Signum when $\mathbf{G}_t = \mathbf{M}_t$ (Bernstein et al., 2018). Under the spectral norm $\|\cdot\|_{S_\infty}$, the map projects onto the leading singular directions: if $\mathbf{G}_t = \mathbf{U}_t \mathbf{\Sigma}_t \mathbf{V}_t^\top$, then $\phi_{\text{spec}}(\mathbf{G}_t) = \mathbf{U}_t \mathbf{V}_t^\top$, corresponding to spectral descent (Spectral-SGD) when $\mathbf{G}_t = \nabla L_{\mathcal{B}_t}$ (Carlson et al., 2015) and to Muon when $\mathbf{G}_t = \mathbf{M}_t$ (Jordan et al., 2024). In all cases, variance-reduced variants are obtained by replacing $\mathbf{G}_t$ with $\mathbf{V}_t$ or $\mathbf{M}_t^V$ while applying the same mapping.

**Assumptions.** Here we present the assumptions required for our analysis to establish the implicit bias.

**Assumption 3.1.** There exists $\mathbf{W} \in \mathbb{R}^{k \times d}$ such that $\min_{c \neq y_i} (\mathbf{e}_{y_i} - \mathbf{e}_c)^\top \mathbf{W} \mathbf{x}_i > 0$ for all $i \in [n]$.

Assumption 3.1 ensures linear separability and a strictly positive margin $\gamma$, a standard assumption in implicit bias studies (Soudry et al., 2018; Ji & Telgarsky, 2019; Zhang et al., 2024; Fan et al., 2025; Baek et al., 2025).

**Assumption 3.2.** There exists constant $R > 0$ such that $\max_{i \in [n]} \|\mathbf{x}_i\|_1 \leq R$.

Assumption 3.2 is a commonly used boundedness condition on the data. Similar assumptions have been used in prior work (Ji & Telgarsky, 2019; Nacson et al., 2019b; Wu et al., 2023; Zhang et al., 2024; Fan et al., 2025; Baek et al., 2025).

**Assumption 3.3.** The learning rate schedule $\{\eta_t\}$ is decreasing with respect to $t$ and satisfies the following conditions: $\lim_{t \to \infty} \eta_t = 0$ and $\sum_{t=0}^{\infty} \eta_t = \infty$.

**Assumption 3.4.** The learning rate schedule satisfies the following: let $\beta \in (0,1)$ and $c_1 > 0$ be two constants, there exist time $t_0 \in \mathbb{N}_+$ and constant $c_2 = c_2(c_1, \beta) > 0$ such that $\sum_{s=0}^{t} \beta^s (e^{c_1 \sum_{\tau=1}^{s} \eta_{t-\tau}} - 1) \leq c_2 \eta_t$ for all $t \geq t_0$.

Assumption 3.3 and 3.4 on the learning rate schedule are commonly used in analyses of implicit bias for gradient-based methods (Zhang et al., 2024; Fan et al., 2025; Baek et al., 2025). In our work, we mainly consider learning rate schedule $\eta_t = \Theta(\frac{1}{t^a})$, where $a \in (0,1]$ which have been extensively studied in prior work on convergence and implicit bias (Nacson et al., 2019b; Sun et al., 2023a; Zhang et al., 2024; Fan et al., 2025). We show that this learning rate schedule satisfies Assumptions 3.3 and 3.4; see Lemma C.15 for details.

# 4 Main Results

## 4.1 Normalized Steepest Descent without Momentum: Full-Batch Guarantee

In this subsection, We establish a large batch condition for convergence and successful classification. We show that, without momentum, convergence and successful classification can only be guaranteed when the algorithm uses the full-batch gradient. The detailed proofs of Theorem 4.1 and Corollary 4.3 are deferred to Appendix D.

**Theorem 4.1** (Margin Convergence of Stochastic Steepest Descent without Momentum). *Suppose Assumptions 3.1, 3.2, and 3.3 hold. Assume the batch size $b$ satisfies the large batch condition: $\rho := \gamma - 4(\frac{n}{b} - 1)R > 0$ ($b > \frac{4Rn}{\gamma + 4R}$). There exist $t_2 = t_2(n, b, \gamma, \mathbf{W}_0, R)$ that for all $t > t_2$, the margin gap of the iterates satisfies:*

$$\rho - \frac{\min_{i \in [n], c \neq y_i}(\mathbf{e}_{y_i} - \mathbf{e}_c)^\top \mathbf{W}_t \mathbf{x}_i}{\|\mathbf{W}_t\|}$$
$$\leq \mathcal{O}\left( \frac{\sum_{s=0}^{t_2-1} \eta_s + \sum_{s=t_2}^{t-1} \eta_s^2 + \sum_{s=t_2}^{t-1} \eta_s e^{-\frac{\rho}{4} \sum_{\tau=t_2}^{s-1} \eta_\tau}}{\sum_{s=0}^{t-1} \eta_s} \right).$$

Theorem 4.1 should be interpreted as a full-batch guarantee rather than a genuine large-batch stochastic guarantee. Indeed, the stated condition $\rho := \gamma - 4\left(\frac{n}{b} - 1\right)R > 0$ is equivalent to $m = \frac{n}{b} < 1 + \frac{\gamma}{4R}$. On the other hand, under Assumption 3.2, the maximum margin satisfies the universal upper bound $\gamma \leq 2R$, hence $1 + \frac{\gamma}{4R} \leq \frac{3}{2}$. Since $m = n/b$ is assumed to be a positive integer, the above condition can hold only when $m = 1$, i.e., when $b = n$. In this case, the stochastic gradient coincides with the full-batch gradient and $\rho = \gamma$. Consequently, Theorem 4.1 shows that, without momentum, worst-case convergence and successful classification can be guaranteed only in the full-batch-gradient regime. The large batch condition is not a vacuous technicality. It ensures that after a sufficiently large time, the empirical loss becomes monotonically decreasing; see Lemma D.1.

This monotone descent guarantees successful classification of all training samples and enables the subsequent margin growth analysis.

This restriction is not merely due to looseness of the large-batch condition. In fact, the following counterexample shows that, already when $m = 2$, random-reshuffling stochastic steepest descent without momentum can fail to converge and correctly classify linearly separable data.

**Proposition 4.2** (Failure of random reshuffling SignSGD when $m = 2$). *Fix any $0 < \varepsilon < 1$. Consider the linearly separable dataset with $n = 2$, $b = 1$, and hence $m = 2$:*

$$(\mathbf{x}_1, y_1) = ((1, -\varepsilon), 1), \qquad (\mathbf{x}_2, y_2) = ((\varepsilon, -1), 2).$$

*For random-reshuffling SignSGD without momentum, initialized at $\mathbf{W}_0 = 0$, for any positive stepsize sequence and every realization of random reshuffling, the iterates **never strictly correctly classify both training samples**.*

This failure already at $m = 2$ highlights the importance of stabilization mechanisms, motivating the momentum and variance-reduction analyses in the following subsections. Although Theorem 4.1 reduces to the full-batch-gradient regime, we state the explicit rate below to facilitate comparison with the momentum and variance-reduction results in later sections.

**Corollary 4.3.** *(Corollary 1 in Fan et al. (2025)) Consider a learning rate schedule of the form $\eta_t = c \cdot t^{-a}$ where $a \in (0,1]$ and $c > 0$. Under the same setting as Theorem 4.1, the margin gap converges with the following rates:*

$$\gamma - \frac{\min_{i \in [n], c \neq y_i}(\mathbf{e}_{y_i} - \mathbf{e}_c)^\top \mathbf{W}_t \mathbf{x}_i}{\|\mathbf{W}_t\|}$$
$$= \begin{cases} \mathcal{O}\left( \frac{t^{1-2a} + n}{t^{1-a}} \right) & \text{if } a < \frac{1}{2} \\ \mathcal{O}\left( \frac{\log t + n}{t^{1/2}} \right) & \text{if } a = \frac{1}{2} \\ \mathcal{O}\left( \frac{n}{t^{1-a}} \right) & \text{if } \frac{1}{2} < a < 1 \\ \mathcal{O}\left( \frac{n}{\log t} \right) & \text{if } a = 1 \end{cases}$$

## 4.2 Implicit Bias with Momentum

In this subsection, we show that momentum enables mini-batch steepest descent to converge to an approximate max-margin solution even with small batches, with the margin gap vanishing as either the batch size increases or $\beta_1 \to 1$. Notably, in addition to building the theory for mini-batch setting, our technical analysis also removes the dependence on the problem dimension $d$ present in prior work (Fan et al., 2025), leading to a tighter bound on the convergence rate. The detailed proofs are deferred to Appendix E.

**Theorem 4.4** (Margin Convergence of Stochastic Steepest Descent with Momentum). *Suppose Assumptions 3.1, 3.2, 3.3, and 3.4 hold. Assume the momentum parameter $\beta_1 \in$*

$(0, 1)$ *and batch size* $b$ *satisfy the positive effective margin condition* $\rho := \gamma - 2(1-\beta_1)m(m^2-1)R > 0$, *where* $m = n/b$. *Let* $D = \frac{4R\eta_0}{1-\sqrt{\beta_1}}$ *denote the constant bound on the momentum drift. Consider a learning rate schedule of the form* $\eta_t = c \cdot t^{-a}$ *with* $a \in (0, 1]$ *and* $\eta_0 \leq c$. *There exists* $t_2 = t_2(n, b, \beta_1, \gamma, \mathbf{W}_0, R)$ *such that for all* $t > t_2$, *the margin gap of the iterates satisfies:*

$$
\rho - \frac{\min_{i \in [n], c \neq y_i}(\mathbf{e}_{y_i} - \mathbf{e}_c)^\top \mathbf{W}_t \mathbf{x}_i}{\|\mathbf{W}_t\|}
$$
$$
\leq \mathcal{O}\left( \frac{\sum_{s=0}^{t_2-1}\eta_s + \frac{m}{1-\beta_1}\sum_{s=t_2}^{t-1}\eta_s^2 + \sum_{s=t_2}^{t-1}\eta_s e^{-\frac{\rho}{4}\sum_{\tau=t_2}^{s-1}\eta_\tau} + D}{\sum_{s=0}^{t-1}\eta_s} \right).
$$

*Remark* 4.5. The momentum-based analysis does not recover the no-momentum result by setting $\beta_1 = 0$ because it exploits smooth exponential decay across epochs and geometric-series bounds, which break down in the degenerate case $\beta_1 = 0$.

Theorem 4.4 shows that momentum fundamentally alters the regime under which the implicit bias of stochastic steepest descent emerges, by stabilizing the descent direction under mini-batch noise, as temporal gradient accumulation exploits the epoch-wise zero-sum structure of Random Reshuffling to cancel stochastic fluctuations and enforce alignment with the full gradient (Lemma C.12 and E.1). In contrast to the no-momentum case, momentum enables convergence to an approximate norm-induced max-margin solution even with small batches, provided that the effective margin $\rho$ is positive, which typically requires sufficiently large momentum $\beta_1$. Specifically, while no-momentum worst-case guarantee $\Omega(\frac{nR}{\gamma+R})$ collapses to $b = n$ under the stated assumptions, the momentum condition only requires $b = \Omega\left(n\left(\frac{(1-\beta_1)R}{\gamma}\right)^{1/3}\right)$. As $\beta_1 \to 1$, the required batch size can become much smaller; e.g., if $1 - \beta_1 = O(n^{-3})$, the condition permits $b = \Omega(1)$. When $\rho > 0$, momentum restores the monotonic loss decay and subsequent margin growth typical of large-batch settings, but at a later threshold $t_2$. The bound further reveals a tradeoff between batch size and momentum: increasing $\beta_1$ or increasing $b$ both reduce the gap between $\rho$ and $\gamma$. While momentum stabilizes the descent direction under stochastic gradients, it also introduces additional terms in the margin bound scaling with $\frac{m}{1-\beta_1}$ and the momentum drift constant $D$, which collectively slow down the convergence compared to the no-momentum case.

Notably, Baek et al. (2025) also analyzed Signum with mini-batch stochastic gradients and obtained margin convergence results related to Theorem 4.4. However, their analysis is limited to Signum and relies on a fixed mini-batch partition that is cycled deterministically throughout training, which departs from standard stochastic training protocols. Moreover, their results characterize only asymptotic convergence behavior and do not provide convergence rate guarantees.

**Corollary 4.6.** *Consider the learning rate schedule* $\eta_t = c \cdot t^{-a}$ *with* $a \in (0, 1]$. *Under the setting of Theorem 4.4, the margin gap converges with the following rates:*

$$
\rho - \frac{\min_{i \in [n], c \neq y_i}(\mathbf{e}_{y_i} - \mathbf{e}_c)^\top \mathbf{W}_t \mathbf{x}_i}{\|\mathbf{W}_t\|}
$$
$$
= \begin{cases}
\mathcal{O}\left( \frac{n[\frac{m}{1-\beta_1}]^{\frac{1}{a}-1} + \frac{m}{1-\beta_1}t^{1-2a}}{t^{1-a}} \right) & \text{if} \quad a < \frac{1}{2} \\[3mm]
\mathcal{O}\left( \frac{\frac{m}{1-\beta_1}(n+\log t)}{t^{1/2}} \right) & \text{if} \quad a = \frac{1}{2} \\[3mm]
\mathcal{O}\left( \frac{n[\frac{m}{1-\beta_1}]^{\frac{1}{a}-1} + \frac{m}{1-\beta_1}}{t^{1-a}} \right) & \text{if} \quad \frac{1}{2} < a < 1 \\[3mm]
\mathcal{O}\left( \frac{n\log(\frac{m}{1-\beta_1}) + \frac{m}{1-\beta_1}}{\log t} \right) & \text{if} \quad a = 1
\end{cases}
$$

Corollary 4.6 builds on Theorem 4.4 by translating the general margin gap bound into explicit convergence rates under specific decaying learning rates. In contrast to existing analyses such as Zhang et al. (2024); Fan et al. (2025), which do not account for the effect of momentum on convergence rates, this corollary explicitly characterizes how the momentum parameter $\beta_1$ and the batch-dependent factor $m = n/b$ jointly influence the convergence rate by carefully controlling the time scale at which the underlying Assumption 3.4 become effective, a step that is technically challenging in the presence of momentum.

The corollary reveals that introducing momentum slows down convergence through multiplicative factors scaling with $\frac{m}{1-\beta_1}$. This behavior is fundamentally different from the well-known acceleration effects of momentum in standard convergence analyses. Here, margin growth is governed by the Taylor expansion $L(\mathbf{W}_{t+1}) \leq L(\mathbf{W}_t) - \eta_t \langle \nabla L(\mathbf{W}_t), \boldsymbol{\Delta}_t \rangle + O(\eta_t^2)$, so the effective progress depends on the alignment inner product $\langle \nabla L(\mathbf{W}_t), \boldsymbol{\Delta}_t \rangle$. While full-gradient steepest descent achieves $\langle \nabla L(\mathbf{W}_t), \boldsymbol{\Delta}_t \rangle = \|\nabla L(\mathbf{W}_t)\|_*$, momentum replaces this with the steepest direction induced by the history-averaged gradient $\mathbf{M}_t = \beta_1 \mathbf{M}_{t-1} + (1-\beta_1)\mathbf{G}_t$, yielding in general $\langle \nabla L(\mathbf{W}_t), \boldsymbol{\Delta}_t \rangle \leq \|\nabla L(\mathbf{W}_t)\|_*$. Controlling this persistent alignment loss under stochasticity introduces factors scaling with $\frac{m}{1-\beta_1}$, delaying the monotonic loss-decreasing regime and leading to a larger threshold time $t_2$ for the margin analysis to apply.

For a direct comparison with the full-batch results in Fan et al. (2025), we set $m = 1$ (full-batch) and treat the momentum parameter $\beta_1$ as a constant. Under this setting, the convergence rates in Corollary 4.6 recover the same rates as those obtained in the no-momentum full-batch case (Corollary 4.3). More importantly, our bound is sharper, as it avoids explicit dependence on the problem dimension $d$ in the numerator. For example, when $a = \frac{1}{2}$, our rate is $\mathcal{O}(\frac{\log t + n}{t^{1/2}})$, whereas the corresponding bound in Fan et al. (2025) is $\mathcal{O}(\frac{d\log t + nd}{t^{1/2}})$. This sharper dependence originates

from a different way of bounding the gradient difference $\nabla L(\mathbf{W}_1) - \nabla L(\mathbf{W}_2)$. Under the same assumption that the data $\|\mathbf{x}_i\|_1$ has upper bound, their analyses apply entry-wise bounds $|\nabla L(\mathbf{W}_1)[i,j] - \nabla L(\mathbf{W}_2)[i,j]|$ and use only coordinate-wise control of $\mathbf{x}_i$, whereas our analysis exploits the global $\ell_1$-norm $\|\nabla L(\mathbf{W}_1) - \nabla L(\mathbf{W}_2)\|_1$ to obtain a tighter, dimension-free scaling.

### 4.3  Implicit Bias with Variance Reduction

In this subsection, we show that incorporating variance reduction restores the implicit bias of stochastic steepest descent to that of full-batch counterpart, yielding convergence to the same norm-induced max-margin solution regardless of batch size or momentum, at the cost of a slower worst-case convergence rate than those obtained with or without momentum. The detailed proofs are deferred to Appendix F and G.

**Theorem 4.7** (Margin Convergence of VR-Stochastic Steepest Descent). *Suppose Assumptions 3.1, 3.2, and 3.3 hold, and that Assumption 3.4 holds when momentum is used. Consider $\eta_t = ct^{-a}$ with $a \in (0, 1]$ and $\eta_0 \leq c$. Define $D = \frac{4R\eta_0}{1 - \sqrt{\beta_1}}$. Then there exists $t_2 = t_2(n, b, \beta_1, \gamma, \mathbf{W}_0, R)$ such that for all $t > t_2$, with $\beta_1 = 0$ corresponding to the case without momentum, variance reduction recovers the full max-margin solution $\gamma$, and the margin gap satisfies*

$$\gamma - \frac{\min_{i \in [n], c \neq y_i}(\mathbf{e}_{y_i} - \mathbf{e}_c)^\top \mathbf{W}_t \mathbf{x}_i}{\|\mathbf{W}_t\|}$$

$$\leq \mathcal{O}\left( \frac{\sum_{s=0}^{t_2-1} \eta_s + C_m \sum_{s=t_2}^{t-1} \eta_s^2 + \sum_{s=t_2}^{t-1} \eta_s e^{-\frac{\gamma}{4}\sum_{\tau=t_2}^{s-1}\eta_\tau} + C_D}{\sum_{s=0}^{t-1} \eta_s} \right),$$

*where* $(C_m, C_D) = \begin{cases} (m^{2+a}, 0), & \text{without momentum}, \\ \left(\dfrac{m^{2+a}}{1-\beta_1}, D\right), & \text{with momentum}. \end{cases}$

Theorem 4.7 show that variance reduction removes the dependence of the implicit bias on algorithmic parameters such as the batch size and momentum. Without requiring either a large batch size or momentum, variance reduction guarantees convergence to the same norm-induced max-margin solution as full-batch steepest descent. This is achieved by correcting the stochastic dynamics so that the descent direction asymptotically aligns with the full gradient, thereby restoring the full-batch steepest descent trajectory under mini-batch sampling (Lemma C.13, C.14, F.1 and G.1). This robustness comes at the cost of a longer transient phase. Compared to stochastic methods without variance reduction, the onset of the monotonic loss-decreasing regime and correct classification occurs later, which is reflected in more conservative convergence bounds involving larger batch-dependent factors, scaling as $m^{2+a}$ and $\frac{m^{2+a}}{1-\beta_1}$.

**Corollary 4.8.** *Under the setting of Theorem 4.7, the following margin convergence rates hold. Setting $\beta_1 = 0$ recovers the corresponding rates for variance-reduced stochastic steepest descent without momentum.*

$$\gamma - \frac{\min_{i \in [n], c \neq y_i}(\mathbf{e}_{y_i} - \mathbf{e}_c)^\top \mathbf{W}_t \mathbf{x}_i}{\|\mathbf{W}_t\|}$$

$$= \begin{cases} \mathcal{O}\left( \dfrac{n(\frac{m^{2+a}}{1-\beta_1})^{\frac{1}{a}-1} + \frac{m^{2+a}}{1-\beta_1}t^{1-2a}}{t^{1-a}} \right) & \text{if } a < \frac{1}{2} \\[2ex] \mathcal{O}\left( \dfrac{\frac{m^{5/2}}{1-\beta_1}(n + \log t)}{t^{1/2}} \right) & \text{if } a = \frac{1}{2} \\[2ex] \mathcal{O}\left( \dfrac{n(\frac{m^{2+a}}{1-\beta_1})^{\frac{1}{a}-1} + \frac{m^{2+a}}{1-\beta_1}}{t^{1-a}} \right) & \text{if } \frac{1}{2} < a < 1 \\[2ex] \mathcal{O}\left( \dfrac{n\log\frac{m^3}{1-\beta_1} + \frac{m^3}{1-\beta_1}}{\log t} \right) & \text{if } a = 1 \end{cases}$$

This corollary shows that while variance reduction guarantees convergence to the full-batch max-margin solution independently of batch size and momentum, it yields more conservative worst-case convergence rates. As in the momentum case, this slowdown is not at odds with the well-known acceleration effects of variance reduction in optimization, but instead stems from a similar alignment mismatch between the full gradient and the update direction induced by the variance-reduced gradient. Controlling this mismatch uniformly over iterations introduces larger batch-dependent factors in the margin gap bounds.

### 4.4  Implicit Bias in Batch-size-one Regime

In previous subsections, we showed that momentum and variance reduction recover the full-batch implicit bias in small-batch regimes. We now ask what implicit bias arises without these mechanisms. Focusing on the extreme case of batch size 1 with plain stochastic steepest descent, we show that it can converge to a fundamentally different implicit bias from its full-batch counterpart.

However, analyzing implicit bias in the batch size 1 regime is particularly challenging. Even algorithmically, stochastic steepest descent driven by per-sample gradients may fail to converge. Convergence often requires increasing batch sizes, restrictive noise assumptions, or momentum (Bernstein et al., 2018; Karimireddy et al., 2019; Cutkosky & Mehta, 2020; Sun et al., 2023b; Jiang et al., 2025). More generally, per-sample updates can deviate substantially from the full gradient and remain highly stochastic, precluding guaranteed descent or tractable dynamics. We therefore focus on a carefully constructed dataset that admits explicit analysis of the induced implicit bias.

A key observation is that when the batch size is 1, Spectral-SGD reduces exactly to Normalized-SGD. (See Appendix H.1) Consequently, in this subsection we focus on per-sample SignSGD and Normalized-SGD. The detailed

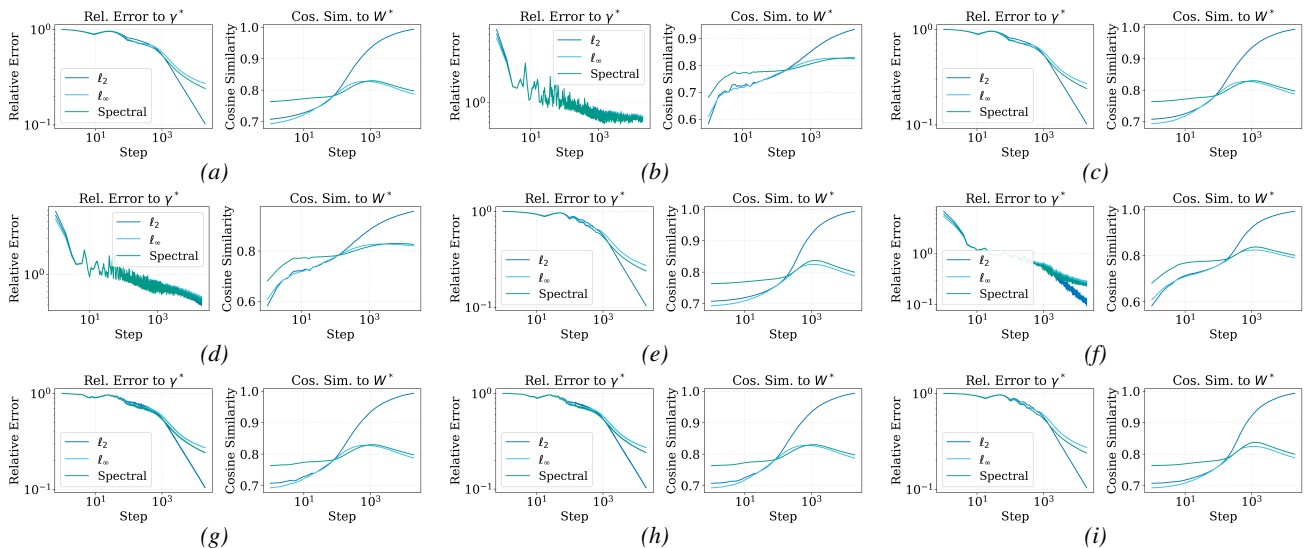

*Figure 1.* Empirical validation of the implicit bias of normalized steepest descent under the $\ell_2$ norm. (a) N-SGD with full-batch size $b = 200$. (b) N-SGD with mini-batch size $b = 20$. (c) N-MSGD with momentum $\beta_1 = 0.5$ and full-batch size $b = 200$. (d) N-MSGD with momentum $\beta_1 = 0.5$ and mini-batch size $b = 20$. (e) N-MSGD with momentum $\beta_1 = 0.99$ and full-batch size $b = 200$. (f) N-MSGD with momentum $\beta_1 = 0.99$ and mini-batch size $b = 20$. (g) VR-N-SGD with mini-batch size $b = 20$. (h) VR-N-MSGD with momentum $\beta_1 = 0.5$ and mini-batch size $b = 20$. (i) VR-N-MSGD with momentum $\beta_1 = 0.99$ and mini-batch size $b = 20$.

proofs of Theorem 4.10 are deferred to Appendix H.2 and the empirical validation is provided in Appendix J.

**Data Construction.** We consider a dataset $\mathcal{D} = \{(\mathbf{x}_i, y_i)\}_{i=1}^n$ with $K$ classes constructed under an *orthogonal scale-skewed* setting. Specifically, for the $i$-th sample with label $y_i$, the input feature is aligned with the canonical basis vector of its class, given by $\boldsymbol{x}_i = \alpha_i \mathbf{e}_{y_i}$, where $\mathbf{e}_{y_i} \in \mathbb{R}^K$ denotes the standard basis vector and $\alpha_i > 0$ represents the arbitrary, heterogeneous scale of the sample.

To characterize the implicit bias induced by per-sample stochastic updates, we introduce a bias matrix constructed by aggregating the normalized per-sample loss gradient directions underlying SignSGD and Normalized-SGD.

**Definition 4.9** (Bias Directions). Define the bias matrix $\bar{\mathbf{W}}$ associated with per-sample SignSGD and per-sample Normalized-SGD as:

$$\bar{\mathbf{W}} \triangleq \begin{cases} \sum_{i=1}^n \big(2\mathbf{e}_{y_i} - \mathbf{1}\big) \operatorname{sign}(\mathbf{x}_i)^\top, & \text{(SignSGD)}, \\ \sum_{i=1}^n \dfrac{\mathbf{e}_{y_i} - \frac{1}{K}\mathbf{1}}{\left\|\mathbf{e}_{y_i} - \frac{1}{K}\mathbf{1}\right\|_2} \dfrac{\mathbf{x}_i^\top}{\|\mathbf{x}_i\|_2}, & \text{(Normalized-SGD)}. \end{cases}$$

**Theorem 4.10** (Implicit Bias of Per-sample SignSGD and Per-sample Normalized-SGD). *Consider per-sample SignSGD and per-sample Normalized-SGD with random reshuffling initialized at $\mathbf{W}_0 = \mathbf{0}$, and learning rate $\eta_t = c\, t^{-a}$ with $c > 0$ and $a \in (0, 1]$. On the Orthogonal Scale-Skewed dataset $\mathcal{D}$, **the training loss converges to zero** and*

$$\lim_{t \to \infty} \frac{\mathbf{W}_t}{\|\mathbf{W}_t\|_F} = \frac{\bar{\mathbf{W}}}{\|\bar{\mathbf{W}}\|_F},$$

*where $\bar{\mathbf{W}}$ is defined in Definition 4.9, corresponding to Per-sample SignSGD or Per-sample Normalized-SGD.*

From Theorem 4.10, we observe a fundamental difference between per-sample methods and full-batch gradient descent. Per-sample updates induce an *averaging effect* over individual samples. As a result, the limiting direction is independent of the sample scales $\alpha_i$ and depends only on the class labels and their frequencies. In particular, class imbalance directly biases the asymptotic solution. This behavior contrasts with full-batch gradient descent, which converges to an $l_p$ max-margin solution driven by the geometry of the hard samples regardless of their frequency.

## 5 Experiment

**Synthetic data.** We consider a synthetic multi-class linear classification problem with 10 classes and 20 samples per class ($n = 200$), feature dimension $d = 5$, and i.i.d. Gaussian features with variance $\sigma = 0.1$, which are linearly separable. We study convergence to the multi-class max-margin solution under steepest descent dynamics, comparing stochastic steepest descent and its variance-reduced variants initialized at $\mathbf{W}_0 = \mathbf{0}$. We sweep batch sizes $b \in \{20, 200\}$ and momentum parameters $\beta \in \{0, 0.5, 0.99\}$. All methods are run for $T = 20{,}000$ iterations using a decaying step size $\eta_t = \eta_0 t^{-\alpha}$ with $\alpha = 0.5$. The base step

size is set to $\eta_0 = 0.5$ for the $\ell_2$ and spectral norms, and $\eta_0 = 0.05$ for the $\ell_\infty$ norm to ensure stability. Convergence is evaluated under the $\ell_2$, $\ell_\infty$, and spectral norms using the relative error to the max-margin $\gamma^*$ and cosine similarity to the max-margin solution $\mathbf{W}^*$ defined in Section 3, with results reported across all batch size and momentum settings.

**Empirical validation of theory.** In the main text, we present empirical results for normalized-SGD (N-SGD), normalized momentum SGD (N-MSGD) and variance reduction variants (VR-N-SGD/VR-N-MSGD) under the $\ell_2$ norm constraint. Results for $\ell_\infty$ and spectral norm are deferred to the Appendix K.

Figure 1 illustrates how batch size, momentum, and variance reduction affect convergence to the $\ell_2$ max-margin solution. (a)-(b) show that without momentum, the implicit bias converges to $\ell_2$ max-margin solution in the full-batch case but fails under mini-batch sampling. (c)-(d) show that moderate momentum ($\beta_1 = 0.5$) partially improves convergence, yielding smaller relative error and higher cosine similarity than the no-momentum case, though discrepancies remain. (e)-(f) show that sufficiently large momentum ($\beta_1 = 0.99$) enables convergence even with small batches. Finally, (g)-(i) confirm that variance reduction eliminates the dependence on batch size and momentum, ensuring convergence to the full-batch max-margin solution across all settings.

**Real-world data.** We further validate our findings on a two-layer non-linear NN trained on MNIST dataset (LeCun et al., 1998). Motivated by Fan et al. (2025), we sample $n = 1000$ data points from $K = 10$ classes (100 per class), and train a two-layer network with hidden dimension $m = 100$, with first-layer weights $\mathbf{W_1}$ and second-layer weights $\mathbf{W_2}$, using cross-entropy loss. We measure the spectral margin $\gamma^{\mathbf{W_1}, \mathbf{W_2}} := \min_{i \in [n], c \neq y_i} \frac{(\mathbf{e}_{y_i} - \mathbf{e}_c)^\top \mathbf{W_2}\, \sigma(\mathbf{W_1}\mathbf{h}_i)}{\max\{\|\|\mathbf{W_1}\|\|_{S_\infty}, \|\|\mathbf{W_2}\|\|_{S_\infty}\}}$. where $\sigma(\cdot)$ denotes the Sigmoid activation. We report its evolution over training steps under different batch sizes $b \in \{10, 100, 500, 1000\}$ and momentum $\beta \in \{0, 0.5, 0.9\}$, along with their variance-reduced counterparts. We observe qualitatively consistent behaviors with our theoretical predictions. See details in Appendix K.

## 6 Conclusions and Limitations

We investigated the implicit bias of mini-batch stochastic steepest descent in multi-class classification, characterizing how batch size, momentum, and variance reduction jointly determine the limiting max-margin behavior. Our results reveal a sharp distinction between stochastic normalized steepest descent with and without stabilization mechanisms. Key theoretical findings are in order: In the absence of momentum, our large-batch condition for convergence and successful classification necessarily reduces to the full-

batch-gradient regime; moreover, a counterexample shows that failure can already occur in the smallest non-full-batch regime $m = 2$. Introducing momentum eliminates the large-batch requirement, allowing small-batch convergence at the expense of slower, dimension-free rates. Variance reduction further strengthens this result, recovering the exact full-batch solution for any batch sizes, albeit with more conservative rates. Conversely, we showed that lacking these mechanisms, per-sample steepest descent converges to a distinct implicit bias governed by sample averaging rather than max-margin geometry.

**Limitations.** Our analysis is subject to several limitations. First, we focus on linear classifiers trained on linearly separable data, which allows a precise characterization of implicit bias but does not extend directly to nonlinear models. Second, our positive convergence guarantee without momentum is limited to the full-batch-gradient regime; outside this regime, our counterexample rules out a general guarantee, but does not provide a complete characterization of all possible stochastic dynamics. Finally, in the vanilla batch-size-one regime, we restrict attention to a carefully constructed dataset to enable explicit analysis.

## Acknowledgments

We would like to thank the anonymous reviewers and area chairs for their helpful comments. We acknowledge the support from NSFC 62306252, Hong Kong ECS award 27309624, Guangdong NSF 2024A1515012444, and the central fund from HKU.

## Impact Statement

This paper presents work whose goal is to advance the field of Machine Learning. There are many potential societal consequences of our work, none which we feel must be specifically highlighted here.

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

# A    Optimization Algorithms

## A.1    Detailed Algorithmic Procedures

---

**Algorithm 1** Unified Stochastic Steepest Descent (Momentum / VR as options)

---

**Require:** $\mathcal{D} = \{(\mathbf{x}_i, y_i)\}_{i=1}^n$, batch size $b$, epochs $K$, stepsizes $\{\eta_t\}$.
**Require:** Norm $\|\cdot\|$ and $\phi_{\|\cdot\|}(\mathbf{G}) = \arg\max_{\|\Delta\| \leq 1} \langle \mathbf{G}, \Delta \rangle$.
**Require:** Switches $\nu, \mu \in \{0, 1\}$ (VR / Momentum) and $\beta_1 \in [0, 1)$.
1: $(\nu, \mu) = (0, 0)$: vanilla; $(0, 1)$: momentum; $(1, 0)$: VR; $(1, 1)$: VR+momentum.
2: Initialize $\mathbf{W}_0$; set $m \leftarrow n/b$, $t \leftarrow 0$, and $\mathbf{H}_{-1} \leftarrow \mathbf{0}$.
3: **for** $k = 0$ to $K - 1$ **do**
4:     Randomly reshuffle $[n]$ and form mini-batches $\{\mathcal{B}_{k,j}\}_{j=0}^{m-1}$.            ▷ random reshuffling
5:     **if** $\nu = 1$ **then**
6:         $\tilde{\mathbf{W}} \leftarrow \mathbf{W}_t$; compute $\nabla L(\tilde{\mathbf{W}})$.            ▷ epoch snapshot / full gradient
7:     **end if**
8:     **for** $j = 0$ to $m - 1$ **do**
9:         $\nabla L_{\mathcal{B}_{k,j}}(\mathbf{W}_t) := \frac{1}{b} \sum_{i \in \mathcal{B}_{k,j}} \nabla \ell(\mathbf{W}_t \mathbf{x}_i; y_i)$.
10:        **if** $\nu = 1$ **then**
11:            $\mathbf{V}_t \leftarrow \nabla L_{\mathcal{B}_{k,j}}(\mathbf{W}_t) - \nabla L_{\mathcal{B}_{k,j}}(\tilde{\mathbf{W}}) + \nabla L(\tilde{\mathbf{W}})$.            ▷ SVRG-style
12:        **end if**
13:        $\mathbf{G}_t \leftarrow (1 - \nu)\nabla L_{\mathcal{B}_{k,j}}(\mathbf{W}_t) + \nu \mathbf{V}_t$.            ▷ $\nu = 0 : \mathbf{G}_t = \nabla L_{\mathcal{B}_{k,j}}(\mathbf{W}_t); \nu = 1 : \mathbf{G}_t = \mathbf{V}_t$
14:        $\mathbf{H}_t \leftarrow (1 - \mu)\mathbf{G}_t + \mu(\beta_1 \mathbf{H}_{t-1} + (1 - \beta_1)\mathbf{G}_t)$.            ▷ $\mu = 0 : \mathbf{H}_t = \mathbf{G}_t; \mu = 1 : \mathbf{H}_t = \beta_1 \mathbf{H}_{t-1} + (1 - \beta_1)\mathbf{G}_t$
15:        $\Delta_t \leftarrow \phi_{\|\cdot\|}(\mathbf{H}_t); \quad \mathbf{W}_{t+1} \leftarrow \mathbf{W}_t - \eta_t \Delta_t; \quad t \leftarrow t + 1$.
16:     **end for**
17: **end for**
18: **return** $\mathbf{W}_t$.

---

# B    Complete Notations

**Notations.**    Scalars, vectors, and matrices are denoted by $x$, $\mathbf{x}$, and $\mathbf{X}$. We denote the $(i, j)$-th entry of $\mathbf{X}$ by $\mathbf{X}[i, j]$ and the $i$-th entry of $\mathbf{x}$ by $\mathbf{x}[i]$. For $k \in \mathbb{N}^+$, let $[k] = \{1, 2, \ldots, k\}$. For real sequences $\{a_t\}$ and $\{b_t\}$, we write $a_t = \mathcal{O}(b_t)$ if there exist constants $C, N > 0$ such that $a_t \leq C b_t$ for all $t \geq N$; $a_t = \Omega(b_t)$ if $b_t = \mathcal{O}(a_t)$; $a_t = \Theta(b_t)$ if both $a_t = \mathcal{O}(b_t)$ and $a_t = \Omega(b_t)$. The entry-wise matrix $p$-norm is defined as $\|\mathbf{X}\|_p := (\sum_{i,j} |\mathbf{X}[i, j]|^p)^{1/p}$ for $p \geq 1$ with the corresponding vector $\ell_p$ norm defined analogously. Of particular interest are the max-norm $\|\mathbf{X}\|_{\max} := \|\mathbf{X}\|_\infty := \max_{i,j} |\mathbf{X}[i, j]|$ and the entry-wise $\ell_1$ norm $\|\mathbf{X}\|_{\text{sum}} := \|\mathbf{X}\|_1 := \sum_{i,j} |\mathbf{X}[i, j]|$, which is dual to the max-norm. For vectors, $\|\mathbf{x}\|_\infty$ and $\|\mathbf{x}\|_1$ denote the usual $\ell_\infty$ and $\ell_1$ norms. The Schatten $p$-norm of $\mathbf{X}$ is defined as $\|\mathbf{X}\|_{S_p} := (\sum_{i=1}^r \sigma_i^p)^{1/p}$, where $\sigma_1 \geq \cdots \geq \sigma_r > 0$ are the singular values of $\mathbf{X}$ and $r = \text{rank}(\mathbf{X})$; special cases include the nuclear ($p = 1$), Frobenius ($p = 2$), and spectral ($p = \infty$) norms. When the specific norm is clear from context, we write $\|\mathbf{X}\|$ to denote any entry-wise or Schatten $p$-norm with $p \geq 1$. The dual norm with respect to the standard matrix inner product $\langle \mathbf{X}, \mathbf{M} \rangle := \text{tr}(\mathbf{X}^\top \mathbf{M})$ is denoted by $\|\mathbf{X}\|_*$. Let $\mathbb{S} : \mathbb{R}^k \to \triangle^{k-1}$ denote the softmax map defined by $\mathbb{S}_c(\mathbf{x}) := \frac{\exp(\mathbf{x}[c])}{\sum_{j \in [k]} \exp(\mathbf{x}[j])}$ for $c \in [k]$. We denote by $\mathbb{S}'(\mathbf{x}) := \text{diag}(\mathbb{S}(\mathbf{x})) - \mathbb{S}(\mathbf{x})\mathbb{S}(\mathbf{x})^\top$ the softmax Jacobian. Let $\{\mathbf{e}_c\}_{c=1}^k$ be the standard basis of $\mathbb{R}^k$, and indicator $\delta_{ij}$ be such that $\delta_{ij} = 1$ if and only if $i = j$.

# C    Technical Lemmas

**Construction of Proxy Function.**    Our construction of the proxy function follows the multiclass formulation introduced by Fan et al. (2025), which relates the gradient magnitude of the cross-entropy loss to the margin. Specifically, we define

$$\mathcal{G}(\mathbf{W}) := \frac{1}{n} \sum_{i=1}^n \big(1 - \mathbb{S}_{y_i}(\mathbf{W}\mathbf{x}_i)\big),$$

and show that it serves as an effective surrogate for our analysis.

### C.1 Properties of Loss function and Norm

In this section, we establish the functional properties and lemmas required for our analysis. Several lemmas presented here are derived from or inspired by Fan et al. (2025). However, we include their proofs here for completeness and to ensure consistency with our notation.

**Lemma C.1.** *(Lemma 11 in (Fan et al., 2025)) For any matrix $\mathbf{A} \in \mathbb{R}^{m \times n}$ and any entry-wise or Schatten p-norm $\|\cdot\|$ with $p \geq 1$, it holds that*

$$\|\mathbf{A}\|_{\max} \leq \|\mathbf{A}\| \leq \|\mathbf{A}\|_{\mathrm{sum}}.$$

*Proof.* The entry-wise p-norm case is trivial. Here, we focus the Schatten p-norm case. Note that $\|\mathbf{A}\|_{S_2}$ coincides with the entrywise 2-norm $\|\mathbf{A}\|_2$, but in general Schatten norms are different from entry-wise norms. On the other hand, Schatten norms preserve the ordering of norms. Specifically, por any $p \geq 1$, it holds:

$$\|\mathbf{A}\|_{S_\infty} = \sigma_1 \leq \|\mathbf{A}\|_{S_p} = \left(\sum_{i=1}^r \sigma_i^p\right)^{1/p} \leq \sum_{i=1}^r \sigma_i = \|\mathbf{A}\|_{S_1}. \tag{1}$$

It is also well-known that

$$\|\mathbf{A}\|_{S_\infty} = \max_{\|\mathbf{u}\|_2 = \|\mathbf{v}\|_2 = 1} \mathbf{u}^\top \mathbf{A} \mathbf{v} \geq \max_{i,j} |\mathbf{A}[i,j]| = \|\mathbf{A}\|_{\max} \tag{2}$$

where the inequality follows by selecting $\mathbf{u} = \mathrm{sign}(\mathbf{A}[i',j']) \cdot \mathbf{e}_{i'}$ and $\mathbf{v} = \mathbf{e}_{j'}$ for $(i',j')$ such that $|\mathbf{A}[i',j']| = \|\mathbf{A}\|_{\max}$ and $\mathbf{e}_{i'}, \mathbf{e}_{j'}$ corresponding basis vectors.

Using this together with duality, it also holds that

$$\|\mathbf{A}\|_{S_1} \leq \|\mathbf{A}\|_{\mathrm{sum}}. \tag{3}$$

This follows from the following sequnece of inequalities

$$\|\mathbf{A}\|_{S_1} = \max_{\|\mathbf{B}\|_{S_\infty} \leq 1} \langle \mathbf{A}, \mathbf{B} \rangle \leq \|\mathbf{A}\|_{\mathrm{sum}} \cdot \max_{\|\mathbf{B}\|_{S_\infty} \leq 1} \|\mathbf{B}\|_{\max} \leq \|\mathbf{A}\|_{\mathrm{sum}} \cdot \max_{\|\mathbf{B}\|_{S_\infty} \leq 1} \|\mathbf{B}\|_{S_\infty} \leq \|\mathbf{A}\|_{\mathrm{sum}}, \tag{4}$$

where the first inequality follows from generalized Cauchy-Scwhartz and the second inequality by (2). $\square$

**Lemma C.2** (Gradient and Hessian). *(Lemma 9 and 10 in (Fan et al., 2025)) Let CE loss*

$$L(\mathbf{W}) := -\frac{1}{n} \sum_{i \in [n]} \log\left(\mathbb{S}_{y_i}(\mathbf{W}\mathbf{x}_i)\right),$$

*and simplify $\mathbf{S} := \mathbb{S}(\mathbf{W}\mathbf{x}) = [\mathbf{s}_1, \ldots, \mathbf{s}_n] \in \mathbb{R}^{k \times n}$. Then, for any $\mathbf{W}$, it holds*

- $\nabla L(\mathbf{W}) = -\frac{1}{n} \sum_{i \in [n]} (\mathbf{e}_{y_i} - \mathbf{s}_i) \mathbf{x}_i^\top = -\frac{1}{n}(\mathbf{Y} - \mathbf{S})\mathbf{x}^\top.$

- $\mathbb{1}_k^\top \nabla L(\mathbf{W}) = 0.$

- *For any matrix $\mathbf{A} \in \mathbb{R}^{k \times d}$,*

$$\langle \mathbf{A}, -\nabla L(\mathbf{W}) \rangle = \frac{1}{n} \sum_{i \in [n]} \sum_{c \neq y_i} s_{ic} (\mathbf{e}_{y_i} - \mathbf{e}_c)^\top \mathbf{A} \mathbf{x}_i \tag{5}$$

  *and consequently,*

$$\langle \mathbf{A}, -\nabla L(\mathbf{W}) \rangle \geq \frac{1}{n} \sum_{i \in [n]} (1 - s_{iy_i}) \cdot \min_{c \neq y_i} (\mathbf{e}_{y_i} - \mathbf{e}_c)^\top \mathbf{A} \mathbf{x}_i. \tag{6}$$

- *Let perturbation $\mathbf{\Delta} \in \mathbb{R}^{k \times d}$ and denote $\mathbf{W}' = \mathbf{W} + \mathbf{\Delta}$. Then,*

$$L(\mathbf{W}') = L(\mathbf{W}) - \frac{1}{n} \sum_{i \in [n]} \langle (\mathbf{e}_{y_i} - \mathbb{S}(\mathbf{W}\mathbf{x}_i))\mathbf{x}_i^\top, \mathbf{\Delta} \rangle$$
$$+ \frac{1}{2n} \sum_{i \in [n]} \mathbf{x}_i^\top \mathbf{\Delta}^\top \left( \mathrm{diag}(\mathbb{S}(\mathbf{W}\mathbf{x}_i)) - \mathbb{S}(\mathbf{W}\mathbf{x}_i)\mathbb{S}(\mathbf{W}\mathbf{x}_i)^\top \right) \mathbf{\Delta}\, \mathbf{x}_i + o(\|\mathbf{\Delta}\|^2)\,.$$

*Proof.* **(Gradient part).** The first bullet is by direct calculation. The second bullet uses the fact that $\mathbb{1}^\top(\mathbf{e}_{y_i} - \mathbf{s}_i) = 1 - 1 = 0$ since $\mathbb{1}^\top \mathbf{s}_i = 1$. For the third bullet, write

$$\mathbf{s}_i^\top \mathbf{A}\mathbf{x}_i = \left( \sum_{c \in [k]} s_{ic}\mathbf{e}_c \right)^\top \mathbf{A}\mathbf{x}_i = \sum_{c \in [k]} s_{ic}\, \mathbf{e}_c^\top \mathbf{A}\mathbf{x}_i,$$

and expand $\langle \mathbf{A}, -\nabla L(\mathbf{W}) \rangle = \frac{1}{n} \sum_{i \in [n]} (\mathbf{e}_{y_i} - \mathbf{s}_i)^\top \mathbf{A}\mathbf{x}_i$ to obtain (5). Then (6) follows by lower bounding the weighted average over $c \neq y_i$ by the minimum term and using $\sum_{c \neq y_i} s_{ic} = 1 - s_{iy_i}$.

**(Hessian / Taylor part).** Define function $\ell_y : \mathbb{R}^k \to \mathbb{R}$ parameterized by $y \in [k]$ as follows:

$$\ell_y(\boldsymbol{l}) := -\log(\mathbb{S}_y(\boldsymbol{l}))\,.$$

From the gradient computation above (equivalently Lemma C.2),

$$\nabla \ell_y(\boldsymbol{l}) = -(\mathbf{e}_y - \mathbb{S}(\boldsymbol{l}))\,.$$

Thus,

$$\nabla^2 \ell_y(\boldsymbol{l}) = \nabla \mathbb{S}(\boldsymbol{l}) = \mathrm{diag}(\mathbb{S}(\boldsymbol{l})) - \mathbb{S}(\boldsymbol{l})\mathbb{S}(\boldsymbol{l})^\top\,.$$

Combining these, the second-order Taylor expansion of $\ell_y$ writes as follows for any $\boldsymbol{l}, \boldsymbol{\delta} \in \mathbb{R}^k$:

$$\ell_y(\boldsymbol{l} + \boldsymbol{\delta}) = \ell_y(\boldsymbol{l}) - (\mathbf{e}_y - \mathbb{S}(\boldsymbol{l}))^\top \boldsymbol{\delta} + \frac{1}{2} \boldsymbol{\delta}^\top \left( \mathrm{diag}(\mathbb{S}(\boldsymbol{l})) - \mathbb{S}(\boldsymbol{l})\mathbb{S}(\boldsymbol{l})^\top \right) \boldsymbol{\delta} + o(\|\boldsymbol{\delta}\|^2)\,.$$

To evaluate this with respect to a change on the classifier parameters, set $\boldsymbol{l} = \mathbf{W}\mathbf{x}$ and $\boldsymbol{\delta} = \mathbf{\Delta}\mathbf{x}$ for $\mathbf{\Delta} \in \mathbb{R}^{k \times d}$. Denoting $\mathbf{W}' = \mathbf{W} + \mathbf{\Delta}$, we then have

$$\ell_y(\mathbf{W}') = \ell_y(\mathbf{W}) - \langle (\mathbf{e}_y - \mathbb{S}(\boldsymbol{l}))\mathbf{x}^\top, \mathbf{\Delta} \rangle + \frac{1}{2}\mathbf{x}^\top \mathbf{\Delta}^\top \left( \mathrm{diag}(\mathbb{S}(\boldsymbol{l})) - \mathbb{S}(\boldsymbol{l})\mathbb{S}(\boldsymbol{l})^\top \right) \mathbf{\Delta}\mathbf{x} + o(\|\mathbf{\Delta}\|^2)\,.$$

Summing over $i \in [n]$ and dividing by $n$ yields the stated expansion for $L(\mathbf{W}')$.

Moreover, using the mean-value form of the remainder, we can further obtain

$$\ell_y(\mathbf{W}') = \ell_y(\mathbf{W}) - \langle (\mathbf{e}_y - \mathbb{S}(\boldsymbol{l}))\mathbf{x}^\top, \mathbf{\Delta} \rangle + \frac{1}{2}\mathbf{x}^\top \mathbf{\Delta}^\top \left( \mathrm{diag}(\mathbb{S}(\boldsymbol{l}')) - \mathbb{S}(\boldsymbol{l}')\mathbb{S}(\boldsymbol{l}')^\top \right) \mathbf{\Delta}\mathbf{x}, \tag{7}$$

where $\boldsymbol{l}' = \boldsymbol{l} + \zeta\boldsymbol{\delta}$ for some $\zeta \in [0, 1]$. $\qquad \square$

Lemma C.3 is used in bounding the second order term in the Taylor expansion of $L(\mathbf{W})$.

**Lemma C.3** (Hessian Quadratic Bound). *Let $\|\cdot\|$ denote any entry-wise or Schatten $p$-norm with $p \geq 1$. Consider a probability vector $\mathbf{s} \in \Delta^{k-1}$, an arbitrary matrix $\mathbf{\Delta} \in \mathbb{R}^{k \times d}$, and a data vector $\mathbf{x} \in \mathbb{R}^d$ bounded by $\|\mathbf{x}\|_* \leq R$ (where $\|\cdot\|_*$ is the dual of the vector norm induced by the problem geometry). For any class index $c \in [k]$, the quadratic form associated with the softmax Hessian satisfies:*

$$\mathbf{x}^\top \mathbf{\Delta}^\top \left( \mathrm{diag}(\mathbf{s}) - \mathbf{s}\mathbf{s}^\top \right) \mathbf{\Delta}\mathbf{x} \leq 4R^2(1 - s_c)\|\mathbf{\Delta}\|^2\,.$$

*Proof.* Let $\mathbf{v} := \boldsymbol{\Delta}\mathbf{x} \in \mathbb{R}^k$ and $\mathbf{H} := \mathrm{diag}(\mathbf{s}) - \mathbf{s}\mathbf{s}^\top \in \mathbb{R}^{k \times k}$. The term of interest can be written as the trace of a product:

$$\mathcal{Q} := \mathbf{v}^\top \mathbf{H}\mathbf{v} = \mathrm{tr}(\mathbf{H}\mathbf{v}\mathbf{v}^\top).$$

Let $q$ be the conjugate exponent of $p$ such that $1/p + 1/q = 1$. By the generalized Hölder's inequality for matrix norms (where $\|\cdot\|_q$ denotes the dual norm of $\|\cdot\|$), we have:

$$\mathcal{Q} \leq \|\mathbf{H}\|_q \|\mathbf{v}\mathbf{v}^\top\|. \tag{8}$$

We proceed by bounding the two terms in (8) separately.

Fisrt, we need to Bound the Hessian Norm $\|\mathbf{H}\|_q$. Recall the norm inequality $\|\mathbf{A}\|_q \leq \|\mathbf{A}\|_1$ (entry-wise 1-norm) for any matrix $\mathbf{A}$ and $q \geq 1$ (since $\|\cdot\|_1$ dominates other p-norms and Schatten norms). We compute the entry-wise 1-norm of $\mathbf{x}$:

$$\|\mathbf{H}\|_1 = \sum_{i,j} |H_{ij}| = \sum_i |s_i - s_i^2| + \sum_{i \neq j} |-s_i s_j| = \sum_i s_i(1 - s_i) + \sum_{i \neq j} s_i s_j.$$

Since $\sum_{j \neq i} s_j = 1 - s_i$, the second term becomes $\sum_i s_i(1 - s_i)$. Thus:

$$\|\mathbf{H}\|_q \leq \|\mathbf{H}\|_1 = 2\sum_{i=1}^k s_i(1 - s_i).$$

To relate this sum to a specific index $c$, we utilize the constraint $\sum_i s_i = 1$:

$$\begin{aligned}
\sum_i s_i(1 - s_i) &= s_c(1 - s_c) + \sum_{i \neq c} s_i - \sum_{i \neq c} s_i^2 \\
&= s_c(1 - s_c) + (1 - s_c) - \sum_{i \neq c} s_i^2 \\
&= (1 - s_c)(1 + s_c) - \sum_{i \neq c} s_i^2 = 1 - s_c^2 - \sum_{i \neq c} s_i^2.
\end{aligned}$$

Alternatively, a simpler algebraic bound suffices: $\sum_i s_i(1-s_i) = 1 - \sum s_i^2 \leq 2(1-s_c)$ is equivalent to $(1-s_c)^2 + \sum_{i \neq c} s_i^2 \geq 0$, which is trivially true. Hence, we establish:

$$\|\mathbf{H}\|_q \leq 4(1 - s_c). \tag{9}$$

Next, we bound the Rank-1 Norm $\|\mathbf{v}\mathbf{v}^\top\|$. We consider two cases for the norm definition:

- **Case I: Schatten p-norms.** The matrix $\mathbf{v}\mathbf{v}^\top$ is rank-1. Its only non-zero singular value is $\|\mathbf{v}\|_2^2$. Thus, for any $p \geq 1$, $\|\mathbf{v}\mathbf{v}^\top\|_S = \|\mathbf{v}\|_2^2$. Using the spectral norm property $\|\boldsymbol{\Delta}\|_{S_\infty} \leq \|\boldsymbol{\Delta}\|_{S_p}$ and $\|\mathbf{x}\|_2 \leq R$:

$$\|\mathbf{v}\mathbf{v}^\top\|_S = \|\mathbf{v}\|_2^2 \leq \|\boldsymbol{\Delta}\|_{S_\infty}^2 \|\mathbf{x}\|_2^2 \leq \|\boldsymbol{\Delta}\|_S^2 \|\mathbf{x}\|_2^2 \leq R^2 \|\boldsymbol{\Delta}\|_S^2 \ .$$

- **Case II: Entry-wise p-norms.** Using the consistency of vector and matrix norms:

$$\|\mathbf{v}\mathbf{v}^\top\| = \|\mathbf{v}\|_p^2 \leq (\|\boldsymbol{\Delta}\|\|\mathbf{x}\|_*)^2 \leq R^2\|\boldsymbol{\Delta}\|^2.$$

In both cases, we obtain:

$$\|\mathbf{v}\mathbf{v}^\top\| \leq R^2\|\boldsymbol{\Delta}\|^2. \tag{10}$$

**Conclusion.** Substituting (9) and (10) back into (8) yields the desired result:

$$\mathbf{x}^\top \boldsymbol{\Delta}^\top \mathbf{H}\boldsymbol{\Delta}\mathbf{x} \leq 4(1 - s_c) \cdot R^2\|\boldsymbol{\Delta}\|^2.$$

$\square$

**Lemma C.4.** *(Lemma 15 in (Fan et al., 2025)) For any $\mathbf{v}, \mathbf{v}', \mathbf{q}, \mathbf{q}' \in \mathbb{R}^k$ and $c \in [k]$, the following inequalities hold:*

*(i)* $|\frac{\mathbb{S}_c(\mathbf{v}')}{\mathbb{S}_c(\mathbf{v})} - 1| \leq e^{2\|\mathbf{v}-\mathbf{v}'\|_\infty} - 1$

*(ii)* $|\frac{1-\mathbb{S}_c(\mathbf{v}')}{1-\mathbb{S}_c(\mathbf{v})} - 1| \leq e^{2\|\mathbf{v}-\mathbf{v}'\|_\infty} - 1$

*Proof.* We prove each inequality:

(i) First, observe that

$$\begin{aligned}
|\frac{\mathbb{S}_c(\mathbf{v}')}{\mathbb{S}_c(\mathbf{v})} - 1| &= |\frac{e^{v'_c}\sum_{i\in[k]}e^{v_i}}{e^{v_c}\sum_{i\in[k]}e^{v'_i}} - 1| \\
&= |\frac{\sum_{i\in[k]}e^{v'_c+v_i} - \sum_{i\in[k]}e^{v_c+v'_i}}{\sum_{i\in[k]}e^{v_c+v'_i}}| \\
&\leq \frac{\sum_{i\in[k]}|e^{v'_c+v_i} - e^{v_c+v'_i}|}{\sum_{i\in[k]}e^{v_c+v'_i}}
\end{aligned}$$

For any $i \in [k]$, we have $\frac{|e^{v'_c+v_i}-e^{v_c+v'_i}|}{e^{v_c+v'_i}} = |e^{v'_c-v_c+v_i-v'_i} - 1| \leq e^{|v'_c-v_c+v_i-v'_i|} - 1 \leq e^{2\|\mathbf{v}-\mathbf{v}'\|_\infty} - 1$. This implies $\sum_{i\in[k]}|e^{v'_c+v_i} - e^{v_c+v'_i}| \leq (e^{2\|\mathbf{v}-\mathbf{v}'\|_\infty} - 1)\sum_{i\in[k]}e^{v_c+v'_i}$, from which we obtain the desired inequality.

(ii) For the second inequality:

$$\begin{aligned}
|\frac{1-\mathbb{S}_c(\mathbf{v}')}{1-\mathbb{S}_c(\mathbf{v})} - 1| &= |\frac{1 - \frac{e^{v'_c}}{\sum_{i\in[k]}e^{v'_i}}}{1 - \frac{e^{v_c}}{\sum_{i\in[k]}e^{v_i}}} - 1| \\
&= |\frac{(\sum_{j\in[k],j\neq c}e^{v'_j})(\sum_{i\in[k]}e^{v_i})}{(\sum_{j\in[k],j\neq c}e^{v_j})(\sum_{i\in[k]}e^{v'_i})} - 1| \\
&= |\frac{\sum_{j\in[k],j\neq c}\sum_{i\in[k]}[e^{v'_j+v_i} - e^{v_j+v'_i}]}{\sum_{j\in[k],j\neq c}\sum_{i\in[k]}e^{v_j+v'_i}}| \\
&\leq \frac{\sum_{j\in[k],j\neq c}\sum_{i\in[k]}|e^{v'_j+v_i} - e^{v_j+v'_i}|}{\sum_{j\in[k],j\neq c}\sum_{i\in[k]}e^{v_j+v'_i}}
\end{aligned}$$

For any $j \in [k]$, $j \neq c$, and $i \in [k]$, we have $\frac{|e^{v'_j+v_i}-e^{v_j+v'_i}|}{e^{v_j+v'_i}} \leq e^{2\|\mathbf{v}-\mathbf{v}'\|_\infty} - 1$. This implies that $\sum_{j\in[k],j\neq c}\sum_{i\in[k]}|e^{v'_j+v_i} - e^{v_j+v'_i}| \leq (e^{2\|\mathbf{v}-\mathbf{v}'\|_\infty} - 1)\sum_{j\in[k],j\neq c}\sum_{i\in[k]}e^{v_j+v'_i}$, from which the result follows.

$\square$

## C.2 Lemmas for Loss and Proxy Function

Lemma C.5 shows that $\mathcal{G}(\mathbf{W})$ upper and lower bound the dual norm of the loss gradient.

**Lemma C.5** ($\mathcal{G}(\mathbf{W})$ as proxy to the loss-gradient norm)**.** *(Lemma 16 in (Fan et al., 2025)) Under Assumption 3.2. For any* $\mathbf{W} \in \mathbb{R}^{k\times d}$*, it holds that*

$$2R \cdot \mathcal{G}(\mathbf{W}) \geq \|\nabla L(\mathbf{W})\|_* \geq \gamma \cdot \mathcal{G}(\mathbf{W}).$$

*Proof.* First, we prove the lower bound. By duality and direct application of (6)

$$\begin{aligned}
\|\nabla L(\mathbf{W})\|_* &= \max_{\|\mathbf{A}\|\leq 1} \langle \mathbf{A}, -\nabla L(\mathbf{W}) \rangle \\
&\geq \max_{\|\mathbf{A}\|\leq 1} \frac{1}{n}\sum_{i\in[n]}(1 - s_{iy_i})\min_{c\neq y_i}(\mathbf{e}_{y_i} - \mathbf{e}_c)^\top \mathbf{A}\mathbf{x}_i \\
&\geq \frac{1}{n}\sum_{i\in[n]}(1 - s_{iy_i}) \cdot \max_{\|\mathbf{A}\|\leq 1}\min_{i\in[n],c\neq y_i}(\mathbf{e}_{y_i} - \mathbf{e}_c)^\top \mathbf{A}\mathbf{x}_i.
\end{aligned}$$

Second, for the upper bound, it holds by triangle inequality and Lemma C.1 that

$$\|\nabla L(\mathbf{W})\|_* \leq \|\nabla L(\mathbf{W})\|_{\mathrm{sum}} \leq \frac{1}{n}\sum_{i\in[n]}\|\nabla\ell_i(\mathbf{W})\|_{\mathrm{sum}},$$

where $\ell_i(\mathbf{W}) = -\log(\mathbb{S}_{y_i}(\mathbf{W}\mathbf{x}_i))$. Recall that

$$\nabla\ell_i(\mathbf{W}) = -(\mathbf{e}_y - \mathbb{S}_{y_i}(\mathbf{W}\mathbf{x}_i))\mathbf{x}_i^\top,$$

and, for two vectors $\mathbf{v}, \mathbf{u}$: $\left\|\mathbf{u}\mathbf{v}^\top\right\|_{\mathrm{sum}} = \|\mathbf{u}\|_1\|\mathbf{v}\|_1$. Combining these and noting that

$$\|\mathbf{e}_{y_i} - \mathbb{S}_{y_i}(\mathbf{W}\mathbf{x}_i)\|_1 = 2(1 - s_{y_i})$$

together with using the assumption $\|\mathbf{x}_i\| \leq R$ yields the advertised upper bound. $\qquad\square$

Built upon Lemma C.5, we obtain a simple bound on the loss difference at two points.

**Lemma C.6.** *(Lemma 17 in (Fan et al., 2025)) For any* $\mathbf{W}, \mathbf{W}_0 \in \mathbb{R}^{k\times d}$, *suppose that* $L(\mathbf{W})$ *is convex, we have*

$$|L(\mathbf{W}) - L(\mathbf{W}_0)| \leq 2R\|\mathbf{W} - \mathbf{W}_0\|.$$

*Proof.* By convexity of $L$, we have

$$L(\mathbf{W}_0) - L(\mathbf{W}) \leq \langle\nabla L(\mathbf{W}_0), \mathbf{W}_0 - \mathbf{W}\rangle \leq \|\nabla L(\mathbf{W}_0)\|_*\|\mathbf{W}_0 - \mathbf{W}\| \leq 2R\|\mathbf{W}_0 - \mathbf{W}\|,$$

where the last inequality is by Lemma C.5. Similarly, we can also show that $L(\mathbf{W}) - L(\mathbf{W}_0) \leq 2R\|\mathbf{W}_0 - \mathbf{W}\|$. $\qquad\square$

Lemma C.7 shows the close relationships between $\mathcal{G}(\mathbf{W})$ and $L(\mathbf{W})$. The proxy $\mathcal{G}(\mathbf{W})$ not only lower bounds $L(\mathbf{W})$, but also upper bounds $L(\mathbf{W})$ up to a factor depending on $L(\mathbf{W})$. Moreover, the rate of convergence $\frac{\mathcal{G}(\mathbf{W})}{L(\mathbf{W})}$ depends on the rate of decrease in the loss.

**Lemma C.7** ($\mathcal{G}(\mathbf{W})$ as proxy to the loss). *(Lemma 18 in (Fan et al., 2025)) Let* $\mathbf{W} \in \mathbb{R}^{k\times d}$, *we have*

*(i)* $1 \geq \frac{\mathcal{G}(\mathbf{W})}{L(\mathbf{W})} \geq 1 - \frac{nL(\mathbf{W})}{2}$

*(ii) Suppose that* $\mathbf{W}$ *satisfies* $L(\mathbf{W}) \leq \frac{\log 2}{n}$ *or* $\mathcal{G}(\mathbf{W}) \leq \frac{1}{2n}$, *then* $L(\mathbf{W}) \leq 2\mathcal{G}(\mathbf{W})$.

*Proof.* **(i)** Let $s_i := \mathbb{S}_{y_i}(\mathbf{W}\mathbf{x}_i)$. Then $L(\mathbf{W}) = \frac{1}{n}\sum_i \log(1/s_i)$ and $\mathcal{G}(\mathbf{W}) = \frac{1}{n}\sum_i(1 - s_i)$.

*Upper Bound* ($\frac{\mathcal{G}}{L} \leq 1$)*:* Using the inequality $\log x \leq x - 1$ with $x = 1/s_i$, we have $\log(1/s_i) \geq 1 - s_i$. Summing over $i$ and dividing by $n$ yields $L(\mathbf{W}) \geq \mathcal{G}(\mathbf{W})$.

*Lower Bound:* We first establish the scalar inequality $1 - s \geq \log(1/s) - \frac{1}{2}\log^2(1/s)$ for $s \in (0, 1]$. Letting $u = \log(1/s) \geq 0$, this is equivalent to $e^{-u} \leq 1 - u + \frac{1}{2}u^2$, which holds by the second-order Taylor expansion of $e^{-u}$. Applying this to each sample:

$$\mathcal{G}(\mathbf{W}) = \frac{1}{n}\sum_{i=1}^n(1 - s_i) \geq \frac{1}{n}\sum_{i=1}^n\left(\log(1/s_i) - \frac{1}{2}\log^2(1/s_i)\right)$$

$$= L(\mathbf{W}) - \frac{1}{2n}\sum_{i=1}^n(\log(1/s_i))^2.$$

Using the inequality $\sum a_i^2 \leq (\sum|a_i|)^2$, we have $\sum_i\log^2(1/s_i) \leq (\sum_i\log(1/s_i))^2 = (nL(\mathbf{W}))^2$. Substituting this back:

$$\mathcal{G}(\mathbf{W}) \geq L(\mathbf{W}) - \frac{1}{2n}(nL(\mathbf{W}))^2 = L(\mathbf{W})\left(1 - \frac{nL(\mathbf{W})}{2}\right).$$

**(ii)** We prove $L(\mathbf{W}) \leq 2\mathcal{G}(\mathbf{W})$ under either condition.

*Case 1:* $L(\mathbf{W}) \leq \frac{\log 2}{n}$. Since $\log 2 < 1$, we have $L(\mathbf{W}) < 1/n$. Thus, $1 - \frac{nL(\mathbf{W})}{2} > 1/2$. Applying the lower bound from (i):

$$\frac{\mathcal{G}(\mathbf{W})}{L(\mathbf{W})} \geq 1 - \frac{nL(\mathbf{W})}{2} > \frac{1}{2} \implies L(\mathbf{W}) < 2\mathcal{G}(\mathbf{W}).$$

*Case 2:* $\mathcal{G}(\mathbf{W}) \leq \frac{1}{2n}$. For any $i \in [n]$, observe that $1 - s_i \leq \sum_j (1 - s_j) = n\mathcal{G}(\mathbf{W}) \leq 1/2$, which implies $s_i \geq 1/2$. Consider the function $h(s) = 2(1 - s) - \log(1/s)$. Its derivative is $h'(s) = -2 + 1/s$. For $s \in [1/2, 1]$, $h'(s) \leq 0$, so $h(s)$ is non-increasing. Since $h(1) = 0$, we have $h(s) \geq 0$ for all $s \in [1/2, 1]$. Therefore, $\log(1/s_i) \leq 2(1 - s_i)$ holds for all $i$. Averaging over $n$ samples yields $L(\mathbf{W}) \leq 2\mathcal{G}(\mathbf{W})$. $\qquad\square$

Lemma C.8 shows that the data becomes separable when the loss is small. It is used in deriving the lower bound on the un-normalized margin.

**Lemma C.8** (Low $L(\mathbf{W})$ implies separability). *(Lemma 19 in (Fan et al., 2025)) Suppose that there exists $\mathbf{W} \in \mathbb{R}^{k \times d}$ such that $L(\mathbf{W}) \leq \frac{\log 2}{n}$, then we have*

$$(\mathbf{e}_{y_i} - \mathbf{e}_c)^\top \mathbf{W}\mathbf{x}_i \geq 0, \quad \text{for all } i \in [n] \text{ and for all } c \in [k] \text{ such that } c \neq y_i. \tag{11}$$

*Proof.* We rewrite the loss into the form:

$$L(\mathbf{W}) = -\frac{1}{n} \sum_{i \in [n]} \log\Big(\frac{e^{\boldsymbol{\ell}_i[y_i]}}{\sum_{c \in [k]} e^{\boldsymbol{\ell}_i[c]}}\Big) = \frac{1}{n} \sum_{i \in [n]} \log\Big(1 + \sum_{c \neq y_i} e^{-(\boldsymbol{\ell}_i[y_i] - \boldsymbol{\ell}_i[c])}\Big).$$

Fix any $i \in [n]$, by the assumption that $L(\mathbf{W}) \leq \frac{\log 2}{n}$, we have the following:

$$\log\Big(1 + \sum_{c \neq y_i} e^{-(\boldsymbol{\ell}_i[y_i] - \boldsymbol{\ell}_i[c])}\Big) \leq nL(\mathbf{W}) \leq \log(2).$$

This implies:

$$e^{-\min_{c \neq y_i}(\boldsymbol{\ell}_i[y_i] - \boldsymbol{\ell}_i[c])} = \max_{c \neq y_i} e^{-(\boldsymbol{\ell}_i[y_i] - \boldsymbol{\ell}_i[c]) \leq} \leq \sum_{c \neq y_i} e^{-(\boldsymbol{\ell}_i[y_i] - \boldsymbol{\ell}_i[c])} \leq 1.$$

After taking $\log$ on both sides, we obtain the following: $\boldsymbol{\ell}_i[y_i] - \boldsymbol{\ell}_i[c] = (\mathbf{e}_{y_i} - \mathbf{e}_c)^\top \mathbf{W}\mathbf{x}_i \geq 0$ for any $c \in [k]$ such that $c \neq y_i$. $\qquad\square$

**Lemma C.9** (Stability of Proxy Function). *For any two weight matrices $\mathbf{W}, \mathbf{W}' \in \mathbb{R}^{k \times d}$, assuming the data satisfies $\|\mathbf{x}_i\|_1 \leq R$, the ratio of the proxy function values is bounded by:*

$$\frac{\mathcal{G}(\mathbf{W}')}{\mathcal{G}(\mathbf{W})} \leq e^{2R\|\mathbf{W}' - \mathbf{W}\|_{\max}} \leq e^{2R\|\mathbf{W}' - \mathbf{W}\|},$$

*where $\|\cdot\|$ denotes any entry-wise or Schatten $p$-norm ($p \geq 1$), and $\|\cdot\|_{\max}$ is the entry-wise max-norm.*

*Proof.* The second inequality follows from the norm dominance relationship $\|\mathbf{A}\|_{\max} \leq \|\mathbf{A}\|$ established in Lemma C.1. We focus on proving the first inequality.

Recall the definition $\mathcal{G}(\mathbf{W}) = \frac{1}{n} \sum_{i \in [n]} \big(1 - \mathbb{S}_{y_i}(\mathbf{W}\mathbf{x}_i)\big)$. Let $\mathbf{v}_i = \mathbf{W}\mathbf{x}_i$ and $\mathbf{v}'_i = \mathbf{W}'\mathbf{x}_i$ be the logit vectors for the $i$-th sample. From the properties of the softmax function (specifically Lemma C.4 (ii)), for any class index $c$, the ratio of the complements of softmax probabilities satisfies:

$$\frac{1 - \mathbb{S}_c(\mathbf{v}'_i)}{1 - \mathbb{S}_c(\mathbf{v}_i)} \leq e^{2\|\mathbf{v}'_i - \mathbf{v}_i\|_\infty}.$$

We now bound the exponent term $\|\mathbf{v}'_i - \mathbf{v}_i\|_\infty$. By definition, $\mathbf{v}'_i - \mathbf{v}_i = (\mathbf{W}' - \mathbf{W})\mathbf{x}_i$. Consider the $j$-th component of this difference vector, given by $\mathbf{e}_j^\top(\mathbf{W}' - \mathbf{W})\mathbf{x}_i$. Using Hölder's inequality with the assumption $\|\mathbf{x}_i\|_1 \leq R$:

$$|(\mathbf{v}'_i - \mathbf{v}_i)_j| = |\mathbf{e}_j^\top(\mathbf{W}' - \mathbf{W})\mathbf{x}_i| \leq \|\mathbf{e}_j^\top(\mathbf{W}' - \mathbf{W})\|_\infty \|\mathbf{x}_i\|_1.$$

Note that $\|\mathbf{e}_j^\top(\mathbf{W}' - \mathbf{W})\|_\infty$ represents the maximum absolute value in the $j$-th row of the difference matrix, which is naturally upper bounded by the global matrix max-norm $\|\mathbf{W}' - \mathbf{W}\|_{\max}$. Therefore:

$$|(\mathbf{v}_i' - \mathbf{v}_i)_j| \le \|\mathbf{W}' - \mathbf{W}\|_{\max} R.$$

Since this bound holds for all components $j \in [k]$, we have:

$$\|\mathbf{v}_i' - \mathbf{v}_i\|_\infty \le R\|\mathbf{W}' - \mathbf{W}\|_{\max}.$$

Substituting this back into the softmax ratio inequality with $c = y_i$:

$$1 - \mathbb{S}_{y_i}(\mathbf{W}'\mathbf{x}_i) \le e^{2R\|\mathbf{W}' - \mathbf{W}\|_{\max}}\big(1 - \mathbb{S}_{y_i}(\mathbf{W}\mathbf{x}_i)\big).$$

Summing over all samples $i \in [n]$ and dividing by $n$:

$$\mathcal{G}(\mathbf{W}') = \frac{1}{n}\sum_{i\in[n]}\big(1 - \mathbb{S}_{y_i}(\mathbf{W}'\mathbf{x}_i)\big) \le e^{2R\|\mathbf{W}' - \mathbf{W}\|_{\max}}\frac{1}{n}\sum_{i\in[n]}\big(1 - \mathbb{S}_{y_i}(\mathbf{W}\mathbf{x}_i)\big)$$

$$= e^{2R\|\mathbf{W}' - \mathbf{W}\|_{\max}}\mathcal{G}(\mathbf{W}).$$

Rearranging the terms yields the desired result. $\qquad\square$

**Lemma C.10** (Gradient Stability via Proxy Function). *For any two weight matrices $\mathbf{W}, \mathbf{W}' \in \mathbb{R}^{k\times d}$, let $\boldsymbol{\Delta} = \mathbf{W}' - \mathbf{W}$. Suppose the data satisfies $\|\mathbf{x}_i\|_1 \le R$. Then, the entry-wise 1-norm of the gradient difference is bounded by:*

$$\|\nabla L(\mathbf{W}') - \nabla L(\mathbf{W})\|_{\mathrm{sum}} \le \frac{1}{n}\sum_{i=1}^n \|\nabla\ell_i(\mathbf{W}') - \nabla\ell_i(\mathbf{W})\|_{\mathrm{sum}} \le 2R\left(e^{2R\|\boldsymbol{\Delta}\|_{\max}} - 1\right)\mathcal{G}(\mathbf{W}).$$

*Proof.* Let $\mathbf{v}_i = \mathbf{W}\mathbf{x}_i$ and $\mathbf{v}_i' = \mathbf{W}'\mathbf{x}_i$ denote the logits for the $i$-th sample. The gradient of the loss at $\mathbf{W}$ is given by $\nabla L(\mathbf{W}) = \frac{1}{n}\sum_{i\in[n]}(\mathbb{S}(\mathbf{v}_i) - \mathbf{e}_{y_i})\mathbf{x}_i^\top$. We proceed in three steps. We consider the entry-wise 1-norm of the gradient difference. By the triangle inequality:

$$\|\nabla L(\mathbf{W}') - \nabla L(\mathbf{W})\|_{\mathrm{sum}} = \sum_{c=1}^k\sum_{j=1}^d\left|\frac{1}{n}\sum_{i=1}^n(\mathbb{S}_c(\mathbf{v}_i') - \mathbb{S}_c(\mathbf{v}_i))x_{ij}\right|$$

$$\le \frac{1}{n}\sum_{i=1}^n\sum_{c=1}^k\sum_{j=1}^d|(\mathbb{S}_c(\mathbf{v}_i') - \mathbb{S}_c(\mathbf{v}_i))x_{ij}| = \frac{1}{n}\sum_{i=1}^n\|\nabla\ell_i(\mathbf{W}') - \nabla\ell_i(\mathbf{W})\|_{\mathrm{sum}}$$

$$= \frac{1}{n}\sum_{i=1}^n\sum_{c=1}^k|\mathbb{S}_c(\mathbf{v}_i') - \mathbb{S}_c(\mathbf{v}_i)|\cdot\underbrace{\sum_{j=1}^d|x_{ij}|}_{\|\mathbf{x}_i\|_1}$$

$$\le \frac{R}{n}\sum_{i=1}^n\underbrace{\sum_{c=1}^k|\mathbb{S}_c(\mathbf{v}_i') - \mathbb{S}_c(\mathbf{v}_i)|}_{\text{Sum of probability shifts}}.$$

Then, we analyze the exponent term appearing in the softmax perturbation bounds. By the definition of $\boldsymbol{\Delta}$ and the data bound:

$$\|\mathbf{v}_i' - \mathbf{v}_i\|_\infty = \|(\mathbf{W}' - \mathbf{W})\mathbf{x}_i\|_\infty \le \|\boldsymbol{\Delta}\|_{\max}\|\mathbf{x}_i\|_1 \le R\|\boldsymbol{\Delta}\|_{\max}.$$

Now, we split the sum over classes $c \in [k]$ into the target class $y_i$ and non-target classes $c \ne y_i$:

*(i) Target class $c = y_i$:* Using Lemma C.4 (ii):

$$|\mathbb{S}_{y_i}(\mathbf{v}_i') - \mathbb{S}_{y_i}(\mathbf{v}_i)| = |(1 - \mathbb{S}_{y_i}(\mathbf{v}_i)) - (1 - \mathbb{S}_{y_i}(\mathbf{v}_i'))|$$

$$= (1 - \mathbb{S}_{y_i}(\mathbf{v}_i))\left|1 - \frac{1 - \mathbb{S}_{y_i}(\mathbf{v}_i')}{1 - \mathbb{S}_{y_i}(\mathbf{v}_i)}\right|$$

$$\le (1 - \mathbb{S}_{y_i}(\mathbf{v}_i))\left(e^{2\|\mathbf{v}_i' - \mathbf{v}_i\|_\infty} - 1\right) \le (1 - \mathbb{S}_{y_i}(\mathbf{v}_i))\left(e^{2R\|\boldsymbol{\Delta}\|_{\max}} - 1\right).$$

*(ii) Non-target classes $c \neq y_i$:* Using Lemma C.4 (i):

$$|\mathbb{S}_c(\mathbf{v}_i') - \mathbb{S}_c(\mathbf{v}_i)| = \mathbb{S}_c(\mathbf{v}_i) \left| \frac{\mathbb{S}_c(\mathbf{v}_i')}{\mathbb{S}_c(\mathbf{v}_i)} - 1 \right|$$

$$\leq \mathbb{S}_c(\mathbf{v}_i) \left( e^{2\|\mathbf{v}_i' - \mathbf{v}_i\|_\infty} - 1 \right) \leq \mathbb{S}_c(\mathbf{v}_i) \left( e^{2R\|\mathbf{\Delta}\|_{\max}} - 1 \right).$$

Summing these parts together for a single sample $i$, and noting that $\sum_{c \neq y_i} \mathbb{S}_c(\mathbf{v}_i) = 1 - \mathbb{S}_{y_i}(\mathbf{v}_i)$:

$$\sum_{c=1}^k |\mathbb{S}_c(\mathbf{v}_i') - \mathbb{S}_c(\mathbf{v}_i)| = |\mathbb{S}_{y_i}(\mathbf{v}_i') - \mathbb{S}_{y_i}(\mathbf{v}_i)| + \sum_{c \neq y_i} |\mathbb{S}_c(\mathbf{v}_i') - \mathbb{S}_c(\mathbf{v}_i)|$$

$$\leq \left( e^{2R\|\mathbf{\Delta}\|_{\max}} - 1 \right) \left[ (1 - \mathbb{S}_{y_i}(\mathbf{v}_i)) + \sum_{c \neq y_i} \mathbb{S}_c(\mathbf{v}_i) \right]$$

$$= 2 \left( e^{2R\|\mathbf{\Delta}\|_{\max}} - 1 \right) (1 - \mathbb{S}_{y_i}(\mathbf{v}_i)).$$

Substitute the result from back into the inequality:

$$\|\nabla L(\mathbf{W}') - \nabla L(\mathbf{W})\|_{\text{sum}} \leq \frac{R}{n} \sum_{i=1}^n 2 \left( e^{2R\|\mathbf{\Delta}\|_{\max}} - 1 \right) (1 - \mathbb{S}_{y_i}(\mathbf{v}_i))$$

$$= 2R \left( e^{2R\|\mathbf{\Delta}\|_{\max}} - 1 \right) \cdot \underbrace{\frac{1}{n} \sum_{i=1}^n (1 - \mathbb{S}_{y_i}(\mathbf{v}_i))}_{\mathcal{G}(\mathbf{W})}.$$

This completes the proof. $\qquad\square$

## C.3 Lemmas for mini-batch algorithm

**Lemma C.11** (Bound for Mini-batch Gradient Noise). *Consider the mini-batch gradient $\nabla L_{\mathcal{B}}(\mathbf{W}) = \frac{1}{b} \sum_{i \in \mathcal{B}} \nabla \ell_i(\mathbf{W})$ computed on a batch $\mathcal{B}$ of size $b$, and the full gradient $\nabla L(\mathbf{W})$. Let $m = n/b$ be an integer. The gradient error matrix $\mathbf{\Xi} := \nabla L_{\mathcal{B}}(\mathbf{W}) - \nabla L(\mathbf{W})$ satisfies the following bound with respect to the entry-wise 1-norm:*

$$\|\mathbf{\Xi}\|_{\text{sum}} \leq 2(m-1)R\mathcal{G}(\mathbf{W}).$$

*Consequently, for any entry-wise or Schatten $p$-norm $\|\cdot\|$, we have $\|\mathbf{\Xi}\| \leq 2(m-1)R\mathcal{G}(\mathbf{W})$.*

*Proof.* We first establish the upper bound for the gradient of a single sample. Recall that $\nabla \ell_i(\mathbf{W}) = -(\mathbf{e}_{y_i} - \mathbf{s}_i)\mathbf{x}_i^\top$. Using the sub-multiplicativity of the entry-wise 1-norm and the data bound $\|\mathbf{x}_i\|_1 \leq R$:

$$\|\nabla \ell_i(\mathbf{W})\|_{\text{sum}} = \|\mathbf{e}_{y_i} - \mathbf{s}_i\|_1 \|\mathbf{x}_i\|_1 \leq 2(1 - s_{iy_i})R.$$

Summing over the entire dataset yields a bound related to the proxy function $\mathcal{G}(\mathbf{W})$:

$$\sum_{i \in [n]} \|\nabla \ell_i(\mathbf{W})\|_{\text{sum}} \leq 2R \sum_{i \in [n]} (1 - s_{iy_i}) = 2nR\mathcal{G}(\mathbf{W}). \tag{12}$$

Next, we utilize the finite population correction identity. The full gradient can be decomposed into the weighted sum of the gradients of the current batch $\mathcal{B}$ and its complement $\mathcal{B}^c$ (where $|\mathcal{B}^c| = n - b$):

$$\nabla L(\mathbf{W}) = \frac{b}{n} \nabla L_{\mathcal{B}}(\mathbf{W}) + \frac{n-b}{n} \nabla L_{\mathcal{B}^c}(\mathbf{W}) = \frac{1}{m} \nabla L_{\mathcal{B}}(\mathbf{W}) + \frac{m-1}{m} \nabla L_{\mathcal{B}^c}(\mathbf{W}).$$

Substituting this into the definition of $\boldsymbol{\Xi}$:

$$\boldsymbol{\Xi} = \nabla L_{\mathcal{B}}(\mathbf{W}) - \left( \frac{1}{m} \nabla L_{\mathcal{B}}(\mathbf{W}) + \frac{m-1}{m} \nabla L_{\mathcal{B}^c}(\mathbf{W}) \right) = \frac{m-1}{m} \left( \nabla L_{\mathcal{B}}(\mathbf{W}) - \nabla L_{\mathcal{B}^c}(\mathbf{W}) \right).$$

Taking the entry-wise 1-norm and applying the triangle inequality:

$$\|\boldsymbol{\Xi}\|_{\text{sum}} = \frac{m-1}{m} \left\| \frac{1}{b} \sum_{i \in \mathcal{B}} \nabla \ell_i(\mathbf{W}) - \frac{1}{n-b} \sum_{j \in \mathcal{B}^c} \nabla \ell_j(\mathbf{W}) \right\|_{\text{sum}}$$

$$\leq \frac{m-1}{m} \left( \frac{1}{b} \sum_{i \in \mathcal{B}} \|\nabla \ell_i(\mathbf{W})\|_{\text{sum}} + \frac{1}{n-b} \sum_{j \in \mathcal{B}^c} \|\nabla \ell_j(\mathbf{W})\|_{\text{sum}} \right).$$

Substituting $b = n/m$ and $n - b = n(m-1)/m$, the coefficients simplify as follows:

$$\frac{m-1}{m} \cdot \frac{1}{b} = \frac{m-1}{n}, \quad \text{and} \quad \frac{m-1}{m} \cdot \frac{1}{n-b} = \frac{1}{n}.$$

Thus, the bound becomes:

$$\|\boldsymbol{\Xi}\|_{\text{sum}} \leq \frac{m-1}{n} \sum_{i \in \mathcal{B}} \|\nabla \ell_i(\mathbf{W})\|_{\text{sum}} + \frac{1}{n} \sum_{j \in \mathcal{B}^c} \|\nabla \ell_j(\mathbf{W})\|_{\text{sum}}.$$

We consider the worst-case distribution of gradient norms.

- If $m = 1$ (full-batch), the coefficient $\frac{m-1}{n} = 0$, yielding $\|\boldsymbol{\Xi}\|_{\text{sum}} = 0$.

- If $m \geq 2$, we have $\frac{m-1}{n} \geq \frac{1}{n}$. The upper bound is maximized when the gradient norms are concentrated in the set with the larger coefficient ($\mathcal{B}$).

Using the total sum bound from (12):

$$\|\boldsymbol{\Xi}\|_{\text{sum}} \leq \frac{m-1}{n} \left( \sum_{i \in [n]} \|\nabla \ell_i(\mathbf{W})\|_{\text{sum}} \right) \leq \frac{m-1}{n} \cdot 2nR\mathcal{G}(\mathbf{W}) = 2(m-1)R\mathcal{G}(\mathbf{W}).$$

The final statement for general norms follows from Lemma C.1 ($\|\cdot\| \leq \|\cdot\|_{\text{sum}}$). $\qquad \square$

**Lemma C.12** (Accumulated Sampling Error of Momentum). *Let $\{\boldsymbol{\Xi}_j\}_{j=0}^t$ be a sequence of gradient error matrices where $\boldsymbol{\Xi}_j := \nabla L_{\mathcal{B}_j}(\mathbf{W}) - \nabla L(\mathbf{W})$ is computed on the mini-batch $\mathcal{B}_j$ at a fixed weight $\mathbf{W}$. Consider the exponentially weighted sum of these errors:*

$$\mathbf{E}_t(\mathbf{W}) := \sum_{j=0}^t (1 - \beta_1) \beta_1^{t-j} \boldsymbol{\Xi}_j.$$

*Under the Random Reshuffling sampling scheme, the entry-wise 1-norm of this accumulated error is bounded by:*

$$\|\mathbf{E}_t(\mathbf{W})\|_{\text{sum}} \leq (1 - \beta_1) m(m^2 - 1) R\mathcal{G}(\mathbf{W}),$$

*where $m = n/b$ denotes the number of mini-batches per epoch.*

*Proof.* We decompose the time horizon $[0, t]$ into full epochs and the current partial epoch. Let $K = \lfloor t/m \rfloor$ be the number of completed epochs. The summation can be split as:

$$\frac{\mathbf{E}_t(\mathbf{W})}{1 - \beta_1} = \sum_{j=0}^t \beta_1^{t-j} \boldsymbol{\Xi}_j = \underbrace{\sum_{k=0}^{K-1} \sum_{j=km}^{(k+1)m-1} \beta_1^{t-j} \boldsymbol{\Xi}_j}_{\text{Historical Full Epochs}} + \underbrace{\sum_{j=Km}^t \beta_1^{t-j} \boldsymbol{\Xi}_j}_{\text{Current Partial Epoch}}.$$

We bound these two parts separately using the uniform bound from Lemma C.11, denoted as $K_\Xi := 2(m-1)R\mathcal{G}(\mathbf{W})$.

First, bound historical full epochs. A fundamental property of Random Reshuffling is that the mini-batches within a full epoch partition the dataset. Consequently, the sum of gradient errors over any full epoch $k$ is zero:

$$\sum_{j=km}^{(k+1)m-1} \Xi_j = \sum_{j=km}^{(k+1)m-1} \left(\nabla L_{\mathcal{B}_j}(\mathbf{W}) - \nabla L(\mathbf{W})\right) = m\nabla L(\mathbf{W}) - m\nabla L(\mathbf{W}) = \mathbf{0}.$$

Leveraging this zero-sum property, we can subtract a constant baseline $\beta_1^{t-km}$ from the coefficients within each epoch $k$ without changing the sum's value:

$$\left\| \sum_{j=km}^{(k+1)m-1} \beta_1^{t-j}\Xi_j \right\|_{\text{sum}} = \left\| \sum_{j=km}^{(k+1)m-1} (\beta_1^{t-j} - \beta_1^{t-km})\Xi_j \right\|_{\text{sum}}$$

$$\leq \sum_{j=km}^{(k+1)m-1} |\beta_1^{t-j} - \beta_1^{t-km}| \cdot \|\Xi_j\|_{\text{sum}}.$$

Let $j = km + l$ with $0 \leq l \leq m-1$. The difference in coefficients is bounded by:

$$|\beta_1^{t-km-l} - \beta_1^{t-km}| = \beta_1^{t-km-l}(1 - \beta_1^l) \leq \beta_1^{t-(k+1)m} \cdot l(1 - \beta_1),$$

where we used $\beta_1^{t-km-l} \leq \beta_1^{t-(k+1)m}$ and the inequality $1 - x^l \leq l(1-x)$. Summing over $l$ for epoch $k$:

$$\text{Sum}_k \leq \sum_{l=0}^{m-1} \beta_1^{t-(k+1)m} l(1 - \beta_1)K_\Xi$$

$$= \beta_1^{t-(k+1)m}(1 - \beta_1)K_\Xi \frac{m(m-1)}{2}.$$

Now, summing over all historical epochs $k = 0, \ldots, K-1$:

$$\text{Total}_{\text{hist}} \leq (1 - \beta_1)K_\Xi \frac{m(m-1)}{2} \sum_{k=0}^{K-1} \beta_1^{t-(k+1)m}.$$

The geometric series $\sum_k \beta_1^{t-(k+1)m}$ is bounded by $\frac{1}{1-\beta_1^m}$. Using $1 - \beta_1^m \geq 1 - \beta_1$, we have $\sum(\ldots) \leq \frac{1}{1-\beta_1}$. Thus:

$$\text{Total}_{\text{hist}} \leq \frac{m(m-1)}{2}K_\Xi.$$

For the current partial epoch, we simply bound the coefficients by 1 and the number of terms by $m$:

$$\text{Total}_{\text{curr}} = \left\| \sum_{j=Km}^{t} \beta_1^{t-j}\Xi_j \right\|_{\text{sum}} \leq \sum_{j=Km}^{t} 1 \cdot K_\Xi \leq mK_\Xi.$$

Combining the bounds and multiplying by the factor $(1 - \beta_1)$:

$$\|\mathbf{E}_t(\mathbf{W})\|_{\text{sum}} \leq (1 - \beta_1)\left(\frac{m(m-1)}{2}K_\Xi + mK_\Xi\right)$$

$$= (1 - \beta_1)\left(\frac{m^2 - m + 2m}{2}\right)K_\Xi$$

$$= (1 - \beta_1)\frac{m(m+1)}{2}K_\Xi.$$

Substituting $K_\Xi = 2(m-1)R\mathcal{G}(\mathbf{W})$ from Lemma C.11:

$$\|\mathbf{E}_t(\mathbf{W})\|_{\text{sum}} \leq (1-\beta_1)\frac{m(m+1)}{2} \cdot 2(m-1)R\mathcal{G}(\mathbf{W})$$
$$= (1-\beta_1)m(m^2-1)R\mathcal{G}(\mathbf{W}).$$

$\square$

**Lemma C.13** (Deviation Bound for the Variance-Reduced Estimator). *Let $\nabla L_{\mathcal{B}}(\cdot)$ denote the mini-batch gradient computed on a batch $\mathcal{B}$ of size $b$, and let $m = n/b$. For any $\mathbf{W}, \mathbf{W}' \in \mathbb{R}^{k \times d}$, let $\Delta = \mathbf{W}' - \mathbf{W}$. We have the following bound on the gradient difference estimator:*

$$\|\nabla L_{\mathcal{B}}(\mathbf{W}) - \nabla L_{\mathcal{B}}(\mathbf{W}') + \nabla L(\mathbf{W}') - \nabla L(\mathbf{W})\|_{\text{sum}} \leq 2(m-1)R(e^{2R\|\Delta\|_{\max}} - 1)\mathcal{G}(\mathbf{W}).$$

*Proof.* Let $\phi_i := \nabla \ell_i(\mathbf{W}) - \nabla \ell_i(\mathbf{W}')$ be the gradient difference for a single sample $i$. Let $\mathbf{A}$ denote the term of interest. We can rewrite it using the definition of mini-batch and full-batch gradients:

$$\mathbf{A} := \frac{1}{b}\sum_{i \in \mathcal{B}} \phi_i - \frac{1}{n}\sum_{i=1}^{n} \phi_i.$$

Using the finite population identity, the full sum can be decomposed into the sum over the current batch $\mathcal{B}$ and the remaining batch $\mathcal{B}^c$ (where $|\mathcal{B}^c| = n - b$):

$$\frac{1}{n}\sum_{i=1}^{n} \phi_i = \frac{b}{n}\left(\frac{1}{b}\sum_{i \in \mathcal{B}} \phi_i\right) + \frac{n-b}{n}\left(\frac{1}{n-b}\sum_{j \in \mathcal{B}^c} \phi_j\right).$$

Substituting this back into the expression for $\mathbf{A}$:

$$\mathbf{A} = \left(1 - \frac{b}{n}\right)\left(\frac{1}{b}\sum_{i \in \mathcal{B}} \phi_i\right) - \frac{n-b}{n}\left(\frac{1}{n-b}\sum_{j \in \mathcal{B}^c} \phi_j\right)$$
$$= \frac{n-b}{n}\left(\frac{1}{b}\sum_{i \in \mathcal{B}} \phi_i - \frac{1}{n-b}\sum_{j \in \mathcal{B}^c} \phi_j\right).$$

Now we take the entry-wise 1-norm. Using the triangle inequality:

$$\|\mathbf{A}\|_{\text{sum}} \leq \frac{n-b}{n}\left(\frac{1}{b}\sum_{i \in \mathcal{B}} \|\phi_i\|_{\text{sum}} + \frac{1}{n-b}\sum_{j \in \mathcal{B}^c} \|\phi_j\|_{\text{sum}}\right)$$
$$= \frac{n-b}{nb}\sum_{i \in \mathcal{B}} \|\phi_i\|_{\text{sum}} + \frac{n-b}{n(n-b)}\sum_{j \in \mathcal{B}^c} \|\phi_j\|_{\text{sum}}.$$

Using the relations $b = n/m$ and $n - b = n(m-1)/m$, the coefficients simplify to:

$$\frac{n-b}{nb} = \frac{n(m-1)/m}{n \cdot (n/m)} = \frac{m-1}{n}, \quad \text{and} \quad \frac{1}{n}.$$

Thus:

$$\|\mathbf{A}\|_{\text{sum}} \leq \frac{m-1}{n}\sum_{i \in \mathcal{B}} \|\phi_i\|_{\text{sum}} + \frac{1}{n}\sum_{j \in \mathcal{B}^c} \|\phi_j\|_{\text{sum}}.$$

Assuming $m \geq 2$ (otherwise the term is 0), we have $\frac{m-1}{n} \geq \frac{1}{n}$. The worst-case bound occurs when the gradient differences are concentrated in the batch $\mathcal{B}$. Thus, we can upper bound the expression by summing over the entire dataset $[n]$ with the larger coefficient:

$$\|\mathbf{A}\|_{\text{sum}} \leq \frac{m-1}{n}\sum_{i=1}^{n} \|\phi_i\|_{\text{sum}} = \frac{m-1}{n}\sum_{i=1}^{n} \|\nabla \ell_i(\mathbf{W}) - \nabla \ell_i(\mathbf{W}')\|_{\text{sum}}. \tag{13}$$

Now we apply the single-sample result derived in the proof of Lemma C.10 and get:

$$\sum_{i=1}^{n} \|\phi_i\|_{\text{sum}} \leq 2R(e^{2R\|\mathbf{W}'-\mathbf{W}\|_{\max}} - 1) \sum_{i=1}^{n} (1 - \mathbb{S}_{y_i}(\mathbf{W}\mathbf{x}_i))$$
$$= 2nR(e^{2R\|\mathbf{W}'-\mathbf{W}\|_{\max}} - 1)\mathcal{G}(\mathbf{W}).$$

Substituting this back into Eq. (13):

$$\|\mathbf{A}\|_{\text{sum}} \leq \frac{m-1}{n} \cdot 2nR(e^{2R\|\mathbf{W}'-\mathbf{W}\|_{\max}} - 1)\mathcal{G}(\mathbf{W})$$
$$= 2(m-1)R(e^{2R\|\mathbf{W}'-\mathbf{W}\|_{\max}} - 1)\mathcal{G}(\mathbf{W}).$$

$\square$

**Lemma C.14** (Momentum Approximation Error for SVR). *Consider the SVR-Momentum estimator* $\mathbf{M}_t^V = \sum_{\tau=0}^{t}(1 - \beta_1)\beta_1^{\tau}\mathbf{V}_{t-\tau}$ *and the full-gradient momentum* $\mathbf{M}_t^{full} = \sum_{\tau=0}^{t}(1 - \beta_1)\beta_1^{\tau}\nabla L(\mathbf{W}_{t-\tau})$. *Suppose Assumptions 3.4 and 3.2 hold. Let* $\eta_t = ct^{-a}$ *with* $a \in (0, 1]$. *Then there exist a time* $t_0$ *such that for all* $t \geq t_0$,

$$\left\|\mathbf{M}_t^V - \mathbf{M}_t^{full}\right\|_{\text{sum}} \leq 2(m-1)(1-\beta_1)Rc_2'\eta_t\mathcal{G}(\mathbf{W}_t) + 16(m^2 - m)(1+2m)^aR^2(1 - \beta_1^m)\eta_t\mathcal{G}(\mathbf{W}_t).$$

*where* $c_2'$ *is the constant from Assumption 3.4 depends on* $R, m, a, \beta_1$.

*Proof.* Let $\mathbf{E}_t := \mathbf{M}_t^V - \mathbf{M}_t^{full} = \sum_{\tau=0}^{t}(1 - \beta_1)\beta_1^{\tau}(\mathbf{V}_{t-\tau} - \nabla L(\mathbf{W}_{t-\tau}))$. Taking the sum-norm and applying the triangle inequality:

$$\|\mathbf{E}_t\|_{\text{sum}} \leq \sum_{\tau=0}^{t}(1 - \beta_1)\beta_1^{\tau}\|\mathbf{V}_{t-\tau} - \nabla L(\mathbf{W}_{t-\tau})\|_{\text{sum}}.$$

Recalling the definition $\mathbf{V}_{t-\tau} = \nabla L_{\mathcal{B}_{t-\tau}}(\mathbf{W}_{t-\tau}) - \nabla L_{\mathcal{B}_{t-\tau}}(\tilde{\mathbf{W}}_{t-\tau}) + \nabla L(\tilde{\mathbf{W}}_{t-\tau})$, the inner term is exactly the deviation bound form. Applying Lemma C.13:

$$\|\mathbf{V}_{t-\tau} - \nabla L(\mathbf{W}_{t-\tau})\|_{\text{sum}} \leq 2(m-1)R\left(e^{2R\|\mathbf{W}_{t-\tau}-\tilde{\mathbf{W}}_{t-\tau}\|_{\max}} - 1\right)\mathcal{G}(\mathbf{W}_{t-\tau}).$$

Next, we relate $\mathcal{G}(\mathbf{W}_{t-\tau})$ to $\mathcal{G}(\mathbf{W}_t)$ using Lemma C.9:

$$\mathcal{G}(\mathbf{W}_{t-\tau}) \leq e^{2R\|\mathbf{W}_{t-\tau}-\mathbf{W}_t\|_{\max}}\mathcal{G}(\mathbf{W}_t).$$

Substituting these back into the summation:

$$\|\mathbf{E}_t\|_{\text{sum}} \leq 2(m-1)(1-\beta_1)R\mathcal{G}(\mathbf{W}_t) \sum_{\tau=0}^{t} \beta_1^{\tau} \underbrace{\left(e^{2R\|\mathbf{W}_{t-\tau}-\tilde{\mathbf{W}}_{t-\tau}\|_{\max}} - 1\right) e^{2R\|\mathbf{W}_{t-\tau}-\mathbf{W}_t\|_{\max}}}_{\text{Term } (\star)}.$$

Since the updates are normalized ($\|\mathbf{\Delta}\|_{\max} \leq 1$), the distance is bounded by the sum of step sizes.

- Distance to snapshot: $\tilde{\mathbf{W}}_{t-\tau}$ is at most $m$ steps away from $\mathbf{W}_{t-\tau}$.

$$\left\|\mathbf{W}_{t-\tau} - \tilde{\mathbf{W}}_{t-\tau}\right\|_{\max} \leq \sum_{k=1}^{m} \eta_{t-\tau-k}.$$

- Distance to current:

$$\|\mathbf{W}_{t-\tau} - \mathbf{W}_t\|_{\max} \leq \sum_{j=1}^{\tau} \eta_{t-j}.$$

Using the algebraic inequality $(e^x - 1)e^y = e^{x+y} - e^y \le e^{x+y} - 1$ (for $x, y \ge 0$):

$$(\star) \le \exp\left(2R\left(\sum_{k=1}^{m} \eta_{t-\tau-k} + \sum_{j=1}^{\tau} \eta_{t-j}\right)\right) - 1$$

$$= \exp\left(2R\sum_{p=1}^{\tau+m} \eta_{t-p}\right) - 1.$$

We now split the summation into two ranges: $\tau \in \{0, \ldots, m-1\}$ and $\tau \in \{m, \ldots, t\}$:

$$\sum_{\tau=0}^{t} \beta_1^{\tau}(\star) \le \sum_{\tau=0}^{m-1} \beta_1^{\tau}\left(e^{2R\sum_{p=1}^{\tau+m} \eta_{t-p}} - 1\right) + \sum_{\tau=m}^{t} \beta_1^{\tau}\left(e^{2R\sum_{p=1}^{\tau+m} \eta_{t-p}} - 1\right).$$

*Remark on indices:* In the summation above, indices $t - p$ may become non-positive when $p \ge t$. consistent with the algorithm initialization, we define $\mathbf{W}_k = \mathbf{W}_0$ for all $k \le 0$. Consequently, the weight difference is zero for these terms. To maintain the validity of the upper bounds, we formally define the effective step size $\eta_k = 0$ for $k < 0$.

**(I) Short range:** $\tau \in \{0, \ldots, m-1\}$. Since $\tau + m \le 2m$, we have

$$\sum_{p=1}^{\tau+m} \eta_{t-p} \le \sum_{p=1}^{2m} \eta_{t-p}.$$

Choose $t_{0,1} := 2m + \left\lceil \left(\frac{4Rmc}{\ln 2}\right)^{1/a} \right\rceil$, so that for all $t \ge t_{0,1}$ we have

$$\sum_{p=1}^{2m} \eta_{t-p} \le 2m\, \eta_{t-2m} = 2mc\,(t - 2m)^{-a} \le \frac{\ln 2}{2R},$$

which implies

$$2R\sum_{p=1}^{2m} \eta_{t-p} \le \ln 2 \quad \implies \quad e^{2R\sum_{p=1}^{2m} \eta_{t-p}} \le 2.$$

Then for all $t \ge t_{0,1}$ and all $\tau \in \{0, \ldots, m-1\}$,

$$e^{2R\sum_{p=1}^{\tau+m} \eta_{t-p}} - 1 \le e^{2R\sum_{p=1}^{2m} \eta_{t-p}} - 1 \le 2 \cdot 2R\sum_{p=1}^{2m} \eta_{t-p} = 4R\sum_{p=1}^{2m} \eta_{t-p},$$

where we used $e^x - 1 \le xe^x \le 2x$ when $x \le \ln 2$. Moreover, for $t \ge 2m+1$, we have the ratio:

$$\frac{\eta_{t-2m}}{\eta_t} = \left(\frac{t}{t - 2m}\right)^a = \left(1 + \frac{2m}{t - 2m}\right)^a \le (1 + 2m)^a,$$

So that:

$$\sum_{p=1}^{2m} \eta_{t-p} \le 2m\, \eta_{t-2m} \le 2m(1 + 2m)^a \eta_t,$$

hence for all $t \ge \max\{t_{0,1}, 2m+1\}$,

$$\sum_{\tau=0}^{m-1} \beta_1^{\tau}\left(e^{2R\sum_{p=1}^{\tau+m} \eta_{t-p}} - 1\right) \le \left(\sum_{\tau=0}^{m-1} \beta_1^{\tau}\right) \cdot 4R \cdot 2m(1 + 2m)^a \eta_t = \frac{8Rm(1 + 2m)^a(1 - \beta_1^m)}{1 - \beta_1}\, \eta_t.$$

**(II) Long range:** $\tau \in \{m, \ldots, t\}$. Fix any $\tau \ge m$. Split the window:

$$\sum_{p=1}^{\tau+m} \eta_{t-p} = \sum_{p=1}^{\tau} \eta_{t-p} + \sum_{q=1}^{m} \eta_{t-\tau-q}.$$

Since $\eta_s$ is non-increasing in $s$ and $t - \tau - q \geq t - \tau - m$, we have

$$\sum_{q=1}^{m} \eta_{t-\tau-q} \leq m \, \eta_{t-\tau-m}.$$

Moreover, for $\eta_t = ct^{-a}$ and $t - \tau \geq 1$,

$$\eta_{t-\tau-m} \leq (1+m)^a \eta_{t-\tau}.$$

$$\sum_{p=1}^{\tau} \eta_{t-p} \geq \eta_{t-\tau}.$$

Combining the last three displays yields

$$\sum_{q=1}^{m} \eta_{t-\tau-q} \leq m(1+m)^a \sum_{p=1}^{\tau} \eta_{t-p},$$

hence

$$\sum_{p=1}^{\tau+m} \eta_{t-p} \leq \left(1 + m(1+m)^a\right) \sum_{p=1}^{\tau} \eta_{t-p}.$$

Let $c_1' := 2R\left(1 + m(1+m)^a\right)$. Then for all $\tau \geq m$,

$$e^{2R \sum_{p=1}^{\tau+m} \eta_{t-p}} - 1 \leq e^{c_1' \sum_{p=1}^{\tau} \eta_{t-p}} - 1.$$

Therefore,

$$\sum_{\tau=m}^{t} \beta_1^\tau \left(e^{2R \sum_{p=1}^{\tau+m} \eta_{t-p}} - 1\right) \leq \sum_{\tau=m}^{t} \beta_1^\tau \left(e^{c_1' \sum_{p=1}^{\tau} \eta_{t-p}} - 1\right) \leq \sum_{\tau=0}^{t} \beta_1^\tau \left(e^{c_1' \sum_{p=1}^{\tau} \eta_{t-p}} - 1\right).$$

By Assumption 3.4 with parameter $c_1'$, there exists a constant $c_2' > 0$ such that for all $t \geq t_{0,2}$,

$$\sum_{\tau=0}^{t} \beta_1^\tau \left(e^{c_1' \sum_{p=1}^{\tau} \eta_{t-p}} - 1\right) \leq c_2' \eta_t.$$

Combining (I) and (II), for all $t \geq t_0 = \max\{t_{0,1}, t_{0,2}, 2m+1\}$,

$$\sum_{\tau=0}^{t} \beta_1^\tau (\star) \leq \left(\frac{8Rm(1+2m)^a(1-\beta_1^m)}{1-\beta_1} + c_2'\right) \eta_t =: c_2 \, \eta_t.$$

Substituting the bound for the sum back into the expression for $\|\mathbf{E}_t\|_{\text{sum}}$:

$$\|\mathbf{E}_t\|_{\text{sum}} \leq 2(m-1)(1-\beta_1)Rc_2'\eta_t\mathcal{G}(\mathbf{W}_t) + 16(m^2-m)(1+2m)^aR^2(1-\beta_1^m)\eta_t\mathcal{G}(\mathbf{W}_t).$$

which completes the proof.

$\square$

**Lemma C.15** (Explicit $t_0$ and $c_2$ (with closed-form Big-$\mathcal{O}$) for Assumption 3.4). *Let $c > 0$ and $a \in (0,1]$. Define* $\eta_t := c\, t^{-a}$ *for all $t \geq 1$, and $\eta_t = 0$ for all $t \leq -1$. Assume the initial value satisfies*

$$0 \leq \eta_0 \leq \eta_1 = c.$$

*Fix any $\beta \in (0,1)$ and $c_1 > 0$, and denote $\lambda := \log(1/\beta) > 0$. Define the explicit times*

$$t_{\mathrm{head}} := \left\lceil \left( \frac{2^{a+1}c_1 c}{\lambda} \right)^{1/a} \right\rceil,$$

$$t_{\mathrm{tail}} := \begin{cases} \left\lceil \left( \frac{8c_1 c}{(1-a)\lambda} \right)^{1/a} \right\rceil, & a \in (0,1), \\ \left\lceil \frac{32 c_1 c}{\lambda} \log\left( \frac{32 c_1 c}{\lambda} \right) \right\rceil, & a = 1, \end{cases}$$

$$t_{\mathrm{poly}} := \left\lceil \frac{16a}{\lambda} \log\left( \frac{16a}{\lambda} \right) \right\rceil,$$

$$t_{\eta_0} := \left\lceil \frac{8c_1 \eta_0}{\lambda} \right\rceil,$$

$$t_0 := \max\{t_{\mathrm{head}}, t_{\mathrm{tail}}, t_{\mathrm{poly}}, t_{\eta_0}, 3\}.$$

*Then for all $t \geq t_0$,*

$$\sum_{s=0}^{t} \beta^s \left( \exp\left( c_1 \sum_{\tau=1}^{s} \eta_{t-\tau} \right) - 1 \right) \leq c_2 \eta_t. \tag{14}$$

*Moreover, without comparing which of $t_{\mathrm{head}}, t_{\mathrm{tail}}, t_{\mathrm{poly}}, t_{\eta_0}$ dominates, $t_0$ admits the* strict closed-form Big-$\mathcal{O}$ upper bounds

$$\text{if } a \in (0,1): \quad t_0 = \mathcal{O}\left( \left( \frac{c_1}{\lambda} \right)^{1/a} + \frac{1}{\lambda} \log\left( \frac{1}{\lambda} \right) + \frac{c_1 \eta_0}{\lambda} \right), \tag{15}$$

$$\text{if } a = 1: \quad t_0 = \mathcal{O}\left( \frac{c_1}{\lambda} \log\left( \frac{c_1}{\lambda} \right) + \frac{1}{\lambda} \log\left( \frac{1}{\lambda} \right) + \frac{c_1 \eta_0}{\lambda} \right). \tag{16}$$

*In particular,*

$$c_2 = \mathcal{O}\left( \frac{c_1}{(1-\beta)^2} + \frac{1}{c(1-\beta)} \right),$$

*and in the regime where $c_1$ is large and $\beta \to 1$ (so $\lambda \to 0$), the dominant scaling is $c_2 = \mathcal{O}\left( \frac{c_1}{(1-\beta)^2} \right)$.*

*Proof.* Fix $t \in \mathbb{N}_+$. Let

$$S(t) := \sum_{s=0}^{t} \beta^s \left( e^{E_{s,t}} - 1 \right), \qquad E_{s,t} := c_1 \sum_{\tau=1}^{s} \eta_{t-\tau}.$$

Split at $s = \lfloor t/2 \rfloor$:

$$S(t) = S_{\mathrm{head}}(t) + S_{\mathrm{tail}}(t),$$

where

$$S_{\mathrm{head}}(t) := \sum_{s=0}^{\lfloor t/2 \rfloor} \beta^s (e^{E_{s,t}} - 1), \quad S_{\mathrm{tail}}(t) := \sum_{s=\lfloor t/2 \rfloor+1}^{t} \beta^s (e^{E_{s,t}} - 1).$$

**Step 1 (Head bound: $S_{\mathrm{head}}(t) \leq \frac{2^{a+1}c_1}{(1-\beta)^2} \eta_t$ for $t \geq t_{\mathrm{head}}$).** For $1 \leq s \leq \lfloor t/2 \rfloor$ and $1 \leq \tau \leq s$, we have $t - \tau \geq t/2$. Since $\eta_u = c\, u^{-a}$ is non-increasing on $u \geq 1$,

$$\eta_{t-\tau} \leq c\,(t/2)^{-a} = 2^a \eta_t.$$

Hence $E_{s,t} \leq c_1 s(2^a \eta_t)$. Let $k_t := 2^a c_1 \eta_t$. Then

$$\beta^s(e^{E_{s,t}} - 1) \leq \beta^s(e^{k_t s} - 1) = (\beta e^{k_t})^s - \beta^s.$$

If $t \geq t_{\mathrm{head}}$, then by definition of $t_{\mathrm{head}}$,

$$\eta_t = c t^{-a} \leq c\, t_{\mathrm{head}}^{-a} \leq \frac{\lambda}{2^{a+1}c_1}, \quad \text{i.e.,} \quad k_t \leq \frac{\lambda}{2},$$

hence $\beta e^{k_t} \le e^{-\lambda}e^{\lambda/2} = e^{-\lambda/2} < 1$. Therefore,

$$S_{\text{head}}(t) \le \sum_{s=0}^{\infty} \left((\beta e^{k_t})^s - \beta^s\right) = \frac{1}{1 - \beta e^{k_t}} - \frac{1}{1-\beta} = \frac{\beta(e^{k_t} - 1)}{(1 - \beta e^{k_t})(1-\beta)}.$$

Using $e^x - 1 \le x e^x$ and $e^{k_t} \le e^{\lambda/2} = \beta^{-1/2}$, we obtain

$$S_{\text{head}}(t) \le \frac{\beta \cdot k_t e^{k_t}}{(1 - \beta e^{k_t})(1-\beta)} \le \frac{\sqrt{\beta}\, k_t}{(1 - \sqrt{\beta})(1-\beta)}.$$

Since $1 - \sqrt{\beta} = \frac{1-\beta}{1+\sqrt{\beta}} \ge \frac{1-\beta}{2}$,

$$S_{\text{head}}(t) \le \frac{2\sqrt{\beta}}{(1-\beta)^2}\, k_t \le \frac{2^{a+1}c_1}{(1-\beta)^2}\, \eta_t, \qquad \forall\, t \ge t_{\text{head}}.$$

**Step 2 (Tail bound: $S_{\text{tail}}(t) \le \frac{1}{c(1-\beta)}\, \eta_t$ for $t \ge \max\{t_{\text{tail}}, t_{\text{poly}}, t_{\eta_0}, 3\}$).** For $s \ge \lfloor t/2 \rfloor + 1$, we have $\beta^s \le \beta^{t/2} = e^{-(\lambda/2)t}$ and $e^{E_{s,t}} - 1 \le e^{E_{t,t}}$. Thus

$$S_{\text{tail}}(t) \le \sum_{s=\lfloor t/2 \rfloor+1}^{\infty} \beta^s e^{E_{t,t}} = \frac{\beta^{\lfloor t/2 \rfloor+1}}{1-\beta} e^{E_{t,t}} \le \frac{1}{1-\beta} e^{-(\lambda/2)t} e^{E_{t,t}}. \tag{17}$$

Moreover,

$$E_{t,t} = c_1 \sum_{\tau=1}^{t} \eta_{t-\tau} = c_1 \sum_{u=0}^{t-1} \eta_u = c_1 \eta_0 + c_1 c \sum_{u=1}^{t-1} u^{-a}.$$

Using the integral bound,

$$\sum_{u=1}^{t-1} u^{-a} \le \begin{cases} 1 + \int_1^t x^{-a} dx = 1 + \frac{t^{1-a}-1}{1-a} \le \frac{t^{1-a}}{1-a}, & a \in (0,1), \\ 1 + \int_1^t \frac{1}{x} dx = 1 + \log t \le 2\log t, & a = 1, \quad t > 3 \end{cases}$$

we obtain

$$E_{t,t} \le c_1 \eta_0 + \begin{cases} \frac{c_1 c}{1-a} t^{1-a}, & a \in (0,1), \\ 2c_1 c \log t, & a = 1. \end{cases}$$

*Case $a \in (0,1)$.* If $t \ge t_{\text{tail}}$, then by definition of $t_{\text{tail}}$ we have $t^a \ge \frac{8c_1 c}{(1-a)\lambda}$, hence $\frac{c_1 c}{1-a} t^{1-a} \le \frac{\lambda}{8} t$. Substituting this into (17) yields

$$S_{\text{tail}}(t) \le \frac{1}{1-\beta} \exp\left(-\frac{\lambda}{2}t + \frac{\lambda}{8}t + c_1 \eta_0\right) = \frac{1}{1-\beta} \exp\left(-\frac{3\lambda}{8}t + c_1 \eta_0\right), \qquad \forall\, t \ge t_{\text{tail}}.$$

*Case $a = 1$.* If $t \ge \max\{t_{\text{tail}}, 3\}$, then $2c_1 c \log t \le \frac{\lambda}{8} t$ by construction, hence

$$S_{\text{tail}}(t) \le \frac{1}{1-\beta} \exp\left(-\frac{3\lambda}{8}t + c_1 \eta_0\right), \qquad \forall\, t \ge \max\{t_{\text{tail}}, 3\}.$$

If also $t \ge t_{\eta_0}$, then by definition of $t_{\eta_0}$ we have $c_1 \eta_0 \le \frac{\lambda}{8} t$, and therefore

$$S_{\text{tail}}(t) \le \frac{1}{1-\beta} \exp\left(-\frac{3\lambda}{8}t + \frac{\lambda}{8}t\right) = \frac{1}{1-\beta} e^{-\lambda t/4}.$$

Finally, if $t \ge \max\{t_{\text{poly}}, 3\}$, then by definition of $t_{\text{poly}}$ we have $e^{-\lambda t/4} \le t^{-a}$, hence for all $t \ge \max\{t_{\text{poly}}, 3\}$,

$$S_{\text{tail}}(t) \le \frac{1}{1-\beta} t^{-a} = \frac{1}{c(1-\beta)} \eta_t, \qquad \forall\, t \ge \max\{t_{\text{tail}}, t_{\eta_0}, t_{\text{poly}}, 3\}.$$

**Step 3 (Combine and define $c_2$).** Let $t_0 = \max\{t_{\text{head}}, t_{\text{tail}}, t_{\text{poly}}, t_{\eta_0}, 3\}$. Then for all $t \geq t_0$,

$$S(t) = S_{\text{head}}(t) + S_{\text{tail}}(t) \leq \left( \frac{2^{a+1} c_1}{(1-\beta)^2} + \frac{1}{c(1-\beta)} \right) \eta_t,$$

which proves (14).

**Step 4 (Closed-form Big-$\mathcal{O}$ bound for $t_0$, built into the lemma).** We bound $t_0$ *without comparing* which component dominates:

$$t_0 = \max\{t_{\text{head}}, t_{\text{tail}}, t_{\text{poly}}, t_{\eta_0}, 3\} \leq t_{\text{head}} + t_{\text{tail}} + t_{\text{poly}} + t_{\eta_0} + 3.$$

Each term has a direct closed-form Big-$\mathcal{O}$ bound (using $\lceil x \rceil \leq x + 1$):

$$t_{\text{head}} = \left\lceil \left( \frac{2^{a+1} c_1 c}{\lambda} \right)^{1/a} \right\rceil = \mathcal{O}\left( \left( \frac{c_1}{\lambda} \right)^{1/a} \right),$$

$$t_{\text{poly}} = \left\lceil \frac{16a}{\lambda} \log\left( \frac{16a}{\lambda} \right) \right\rceil = \mathcal{O}\left( \frac{1}{\lambda} \log\left( \frac{1}{\lambda} \right) \right),$$

$$t_{\eta_0} = \left\lceil \frac{8 c_1 \eta_0}{\lambda} \right\rceil = \mathcal{O}\left( \frac{c_1 \eta_0}{\lambda} \right),$$

and

$$t_{\text{tail}} = \begin{cases} \left\lceil \left( \frac{8 c_1 c}{(1-a)\lambda} \right)^{1/a} \right\rceil = \mathcal{O}\left( \left( \frac{c_1}{\lambda} \right)^{1/a} \right), & a \in (0,1), \\ \left\lceil \frac{32 c_1 c}{\lambda} \log\left( \frac{32 c_1 c}{\lambda} \right) \right\rceil = \mathcal{O}\left( \frac{c_1}{\lambda} \log\left( \frac{c_1}{\lambda} \right) \right), & a = 1. \end{cases}$$

Substituting these bounds into $t_0 \leq t_{\text{head}} + t_{\text{tail}} + t_{\text{poly}} + t_{\eta_0} + 3$ yields (15) for $a \in (0,1)$ and (16) for $a = 1$. The Big-$\mathcal{O}$ statement for $c_2$ follows from Step 3 by absorbing fixed constants. $\square$

# D  Proof in Section 4.1

**Lemma D.1** (Descent Lemma for Stochastic $l_p$-Steepest Descent Algorithm without Momentum). *Suppose that Assumption 3.1, 3.2, and 3.3 hold, it holds for all $t \geq 0$,*

$$L(\mathbf{W}_{t+1}) \leq L(\mathbf{W}_t) - \eta_t(\gamma - 4(m-1)R - 2\eta_t R^2 e^{2R\eta_0}) \mathcal{G}(\mathbf{W}_t)$$

*Proof.* By Lemma C.2, let $\tilde{\boldsymbol{\Delta}}_t = \mathbf{W}_{t+1} - \mathbf{W}_t$, and define $\mathbf{W}_{t,t+1,\zeta} := \mathbf{W}_t + \zeta(\mathbf{W}_{t+1} - \mathbf{W}_t)$. We choose $\zeta^*$ such that $\mathbf{W}_{t,t+1,\zeta^*}$ satisfies (7), then we have:

$$L(\mathbf{W}_{t+1}) = L(\mathbf{W}_t) + \underbrace{\langle \nabla L(\mathbf{W}_t), \mathbf{W}_{t+1} - \mathbf{W}_t \rangle}_{A}$$

$$+ \frac{1}{2n} \sum_{i \in [n]} \underbrace{\mathbf{x}_i^\top \tilde{\boldsymbol{\Delta}}_t^\top \left( \text{diag}(\mathbb{S}(\mathbf{W}_{t,t+1,\zeta^*} \mathbf{x}_i)) - \mathbb{S}(\mathbf{W}_{t,t+1,\zeta^*} \mathbf{x}_i) \mathbb{S}(\mathbf{W}_{t,t+1,\zeta^*} \mathbf{x}_i)^\top \right) \tilde{\boldsymbol{\Delta}}_t \mathbf{x}_i}_{B}. \tag{18}$$

For Term A:

$$\langle \nabla L(\mathbf{W}_t), \mathbf{W}_{t+1} - \mathbf{W}_t \rangle = \langle \nabla L(\mathbf{W}_t) - \nabla L_{\mathcal{B}_t}(\mathbf{W}_t), \mathbf{W}_{t+1} - \mathbf{W}_t \rangle + \langle \nabla L_{\mathcal{B}_t}(\mathbf{W}_t), \mathbf{W}_{t+1} - \mathbf{W}_t \rangle$$

$$= -\eta_t \langle \nabla L(\mathbf{W}_t) - \nabla L_{\mathcal{B}_t}(\mathbf{W}_t), \Delta_t \rangle - \eta_t \langle \nabla L_{\mathcal{B}_t}(\mathbf{W}_t), \Delta_t \rangle$$

$$\overset{(a)}{\leq} \eta_t \|\nabla L(\mathbf{W}_t) - \nabla L_{\mathcal{B}_t}(\mathbf{W}_t)\|_* \|\Delta_t\| - \eta_t \|\nabla L_{\mathcal{B}_t}(\mathbf{W}_t)\|_*$$

$$\overset{(b)}{\leq} \eta_t \|\nabla L(\mathbf{W}_t) - \nabla L_{\mathcal{B}_t}(\mathbf{W}_t)\|_* - \eta_t \|\nabla L_{\mathcal{B}_t}(\mathbf{W}_t) - \nabla L(\mathbf{W}_t) + \nabla L(\mathbf{W}_t)\|_*$$

$$\overset{(c)}{\leq} \eta_t \|\nabla L(\mathbf{W}_t) - \nabla L_{\mathcal{B}_t}(\mathbf{W}_t)\|_{\text{sum}} - \eta_t (\|\nabla L(\mathbf{W}_t)\|_* - \|\nabla L(\mathbf{W}_t) - \nabla L_{\mathcal{B}_t}(\mathbf{W}_t)\|_*)$$

$$= 2\eta_t \|\nabla L(\mathbf{W}_t) - \nabla L_{\mathcal{B}_t}(\mathbf{W}_t)\|_{\text{sum}} - \eta \|\nabla L(\mathbf{W}_t)\|_*$$

$$\overset{(d)}{\leq} 4\eta_t (m-1) R \mathcal{G}(\mathbf{W}_t) - \eta_t \gamma \mathcal{G}(\mathbf{W}_t),$$

where (a) is by Cauchy Schwarz inequality and $\langle \nabla L_{\mathcal{B}_t}(\mathbf{W}_t), \boldsymbol{\Delta} \rangle = \|\nabla L_{\mathcal{B}_t}(\mathbf{W}_t)\|_*$, (b) is by $\|\boldsymbol{\Delta}\| \leq 1$, (c) is via Lemma C.1 and Triangle inequality and (d) is via Lemma C.5 and Lemma C.11.

For Term B, let $\mathbf{v} = \tilde{\boldsymbol{\Delta}}_t \mathbf{x}_i$ and $\mathbf{s} = \mathbb{S}(\mathbf{W}_{t,t+1,\zeta^*} \mathbf{x}_i)$, and apply Lemma C.3. Notice that $\|\tilde{\boldsymbol{\Delta}}_t\| \leq \eta_t$ and $\|\mathbf{x}_i\|_* \leq \|\mathbf{x}_i\|_1 \leq R$. Then we get:

$$TermB \leq 4\|\tilde{\boldsymbol{\Delta}}_t\|^2 \|\mathbf{x}_i\|_*^2 (1 - \mathbb{S}_{y_i}(\mathbf{W}_{t,t+1,\zeta^*} \mathbf{x}_i)) \leq 4\eta_t^2 R^2 (1 - \mathbb{S}_{y_i}(\mathbf{W}_{t,t+1,\zeta^*} \mathbf{x}_i)),$$

Combining Terms A, B together, we obtain

$$L(\mathbf{W}_{t+1}) \leq L(\mathbf{W}_t) - \gamma \eta_t \mathcal{G}(\mathbf{W}_t) + 4\eta_t(m-1)R\mathcal{G}(\mathbf{W}_t) + 2\eta_t^2 R^2 \frac{1}{n} \sum_{i \in [n]} (1 - \mathbb{S}_{y_i}(\mathbf{W}_{t,t+1,\zeta^*} \mathbf{x}_i))$$

$$= L(\mathbf{W}_t) - \gamma \eta_t \mathcal{G}(\mathbf{W}_t) + 4\eta_t(m-1)R\mathcal{G}(\mathbf{W}_t) + 2\eta_t^2 R^2 \mathcal{G}(\mathbf{W}_{t,t+1,\zeta^*})$$

$$\leq L(\mathbf{W}_t) - \gamma \eta_t \mathcal{G}(\mathbf{W}_t) + 4\eta_t(m-1)R\mathcal{G}(\mathbf{W}_t) + 2\eta_t^2 R^2 \sup_{\zeta \in [0,1]} \mathcal{G}(\mathbf{W}_{t,t+1,\zeta})$$

$$= L(\mathbf{W}_t) - \gamma \eta_t \mathcal{G}(\mathbf{W}_t) + 4\eta_t(m-1)R\mathcal{G}(\mathbf{W}_t) + 2\eta_t^2 R^2 \mathcal{G}(\mathbf{W}_t) \sup_{\zeta \in [0,1]} \frac{\mathcal{G}(\mathbf{W}_t + \zeta\tilde{\boldsymbol{\Delta}}_t)}{\mathcal{G}(\mathbf{W}_t)}$$

$$\overset{(a)}{\leq} L(\mathbf{W}_t) - \gamma \eta_t \mathcal{G}(\mathbf{W}_t) + 4\eta_t(m-1)R\mathcal{G}(\mathbf{W}_t) + 2\eta_t^2 R^2 \mathcal{G}(\mathbf{W}_t) \sup_{\zeta \in [0,1]} e^{2R\zeta\|\tilde{\boldsymbol{\Delta}}_t\|}$$

$$\overset{(b)}{\leq} L(\mathbf{W}_t) - \gamma \eta_t \mathcal{G}(\mathbf{W}_t) + 4\eta_t(m-1)R\mathcal{G}(\mathbf{W}_t) + 2\eta_t^2 R^2 e^{2R\eta_0} \mathcal{G}(\mathbf{W}_t)$$

$$= L(\mathbf{W}_t) - \eta_t(\gamma - 4(m-1)R - 2\eta_t R^2 e^{2R\eta_0})\mathcal{G}(\mathbf{W}_t) \tag{19}$$

where (a) is by Lemma C.9 and (b) is by $\|\tilde{\boldsymbol{\Delta}}_t\| \leq \eta_t$. $\qquad \square$

From Eq.19, we can see that in **Large batch setting**, where $4(m-1)R < \gamma \to b > \frac{4Rn}{\gamma+4R}$, the loss starts to monotonically decrease after $\eta_t$ satisfies $\eta_t \leq \frac{\gamma-4(m-1)R}{2R^2 e^{2R\eta_0}}$ for a decreasing learning rate schedule.

The following lemma establishes the convergence of the training loss. This result guarantees that the iterates eventually enter the region of linear separability (i.e., $L(\mathbf{W}_t) \leq \frac{\log 2}{n}$), satisfying the precondition for the margin maximization analysis.

**Lemma D.2** (Loss convergence). *Suppose Assumptions 3.1, 3.2, and 3.3 hold, and the batch size satisfies the large batch condition (i.e., $\rho := \gamma - 4(m-1)R > 0$). Let $\tilde{L} := \frac{\log 2}{n}$. Then, there exists a time index $t_2$ such that for all $t > t_2$, $L(\mathbf{W}_t) \leq \tilde{L}$. Specifically, the condition for $t_2$ is determined by the learning rate accumulation:*

$$\sum_{s=t_1}^{t_2} \eta_s \geq \frac{4L(\mathbf{W}_0) + 8R \sum_{s=0}^{t_1-1} \eta_s}{\rho \tilde{L}}, \tag{20}$$

*where $t_1$ is the time step ensuring monotonic descent.*

*Proof.* **Determination of $t_1$ (Start of Monotonicity).** Recall the descent inequality from Lemma D.1:

$$L(\mathbf{W}_{t+1}) \leq L(\mathbf{W}_t) - \eta_t(\rho - \eta_t\alpha_1)\mathcal{G}(\mathbf{W}_t),$$

where $\alpha_1 = 2R^2 e^{2R\eta_0}$. We choose $t_1$ such that for all $t \geq t_1$, the effective descent term dominates the curvature noise, i.e., $\eta_t\alpha_1 \leq \frac{\rho}{2}$. Considering $\eta_t = \Theta(t^{-a})$, we set $t_1$ such that $\eta_{t_1} \leq \frac{\rho}{2\alpha_1}$. Then, for all $t \geq t_1$:

$$L(\mathbf{W}_{t+1}) \leq L(\mathbf{W}_t) - \frac{\rho}{2}\eta_t\mathcal{G}(\mathbf{W}_t). \tag{21}$$

Rearranging this equation and summing from $t_1$ to $t_2$, and using $L(\mathbf{W}_{t_2+1}) \geq 0$, we obtain:

$$\frac{\rho}{2} \sum_{s=t_1}^{t_2} \eta_s \mathcal{G}(\mathbf{W}_s) \leq L(\mathbf{W}_{t_1}) - L(\mathbf{W}_{t_2+1}) \leq L(\mathbf{W}_{t_1}). \tag{22}$$

**Determination of $t_2$ (Crossing the Threshold).** First, we bound the initial loss at $t_1$ using Lemma C.6:

$$|L(\mathbf{W}_{t_1}) - L(\mathbf{W}_0)| \leq 2R\|\mathbf{W}_{t_1} - \mathbf{W}_0\| \leq 2R\sum_{s=0}^{t_1-1}\eta_s\|\boldsymbol{\Delta}_s\| \leq 2R\sum_{s=0}^{t_1-1}\eta_s,$$

where we used the fact that the normalized update satisfies $\|\boldsymbol{\Delta}_s\| \leq 1$. Thus, $L(\mathbf{W}_{t_1}) \leq L(\mathbf{W}_0) + 2R\sum_{s=0}^{t_1-1}\eta_s$.

Next, let $t^* = \operatorname{argmin}_{s\in[t_1,t_2]}\mathcal{G}(\mathbf{W}_s)$. From Eq. (22), we have:

$$\mathcal{G}(\mathbf{W}_{t^*})\sum_{s=t_1}^{t_2}\eta_s \leq \sum_{s=t_1}^{t_2}\eta_s\mathcal{G}(\mathbf{W}_s) \leq \frac{2}{\rho}L(\mathbf{W}_{t_1}).$$

Substituting the bound for $L(\mathbf{W}_{t_1})$:

$$\mathcal{G}(\mathbf{W}_{t^*}) \leq \frac{2\left(L(\mathbf{W}_0) + 2R\sum_{s=0}^{t_1-1}\eta_s\right)}{\rho\sum_{s=t_1}^{t_2}\eta_s}.$$

We require $\mathbf{W}_{t^*}$ to be deep enough in the separable region. Specifically, we want to satisfy the condition of Lemma C.7 (ii). A sufficient condition is $\mathcal{G}(\mathbf{W}_{t^*}) \leq \frac{\tilde{L}}{2} = \frac{\log 2}{2n}(\leq \frac{1}{2n})$. Setting the upper bound of $\mathcal{G}(\mathbf{W}_{t^*})$ to be less than or equal to $\frac{\tilde{L}}{2}$, we derive the sufficient condition for $t_2$:

$$\frac{2L(\mathbf{W}_0) + 4R\sum_{s=0}^{t_1-1}\eta_s}{\rho\sum_{s=t_1}^{t_2}\eta_s} \leq \frac{\tilde{L}}{2} \iff \sum_{s=t_1}^{t_2}\eta_s \geq \frac{4L(\mathbf{W}_0) + 8R\sum_{s=0}^{t_1-1}\eta_s}{\rho\tilde{L}}.$$

Once $t_2$ satisfies the above condition, there exists $t^* \in [t_1, t_2]$ such that $\mathcal{G}(\mathbf{W}_{t^*}) \leq \frac{\tilde{L}}{2}$. By Lemma C.7 (ii), this implies $L(\mathbf{W}_{t^*}) \leq 2\mathcal{G}(\mathbf{W}_{t^*}) \leq \tilde{L}$. Since the loss is monotonically decreasing for all $t \geq t_1$ (Eq. (21)), for any $t > t_2$ (which implies $t > t^*$), we have:

$$L(\mathbf{W}_t) \leq L(\mathbf{W}_{t^*}) \leq \tilde{L} = \frac{\log 2}{n}.$$

This confirms that for all $t > t_2$, the iterates remain in the low-loss separable region. $\qquad\square$

**Lemma D.3** (Unnormalized Margin). *Consider the same setting as Lemma D.2. Let $t_2$ be the time index guaranteed by Lemma D.2 such that $L(\mathbf{W}_t) \leq \frac{\log 2}{n}$ for all $t > t_2$. Define the effective margin $\rho := \gamma - 4(m-1)R$. Then, for all $t > t_2$, the minimum unnormalized margin satisfies:*

$$\min_{i\in[n],c\neq y_i}(\mathbf{e}_{y_i} - \mathbf{e}_c)^\top\mathbf{W}_t\mathbf{x}_i \geq \rho\sum_{s=t_2}^{t-1}\eta_s\frac{\mathcal{G}(\mathbf{W}_s)}{L(\mathbf{W}_s)} - \alpha_1\sum_{s=t_2}^{t-1}\eta_s^2, \tag{23}$$

*where $\alpha_1 = 2R^2 e^{2R\eta_0}$ is a constant.*

*Proof.* Recall the descent inequality derived in Lemma D.1:

$$L(\mathbf{W}_{s+1}) \leq L(\mathbf{W}_s) - \eta_s(\rho - \eta_s\alpha_1)\mathcal{G}(\mathbf{W}_s).$$

Rearranging and using the inequality $1 - x \leq e^{-x}$:

$$\begin{aligned}
L(\mathbf{W}_{s+1}) &\leq L(\mathbf{W}_s)\left(1 - \eta_s\rho\frac{\mathcal{G}(\mathbf{W}_s)}{L(\mathbf{W}_s)} + \eta_s^2\alpha_1\frac{\mathcal{G}(\mathbf{W}_s)}{L(\mathbf{W}_s)}\right)\\
&\stackrel{(a)}{\leq} L(\mathbf{W}_s)\exp\left(-\rho\eta_s\frac{\mathcal{G}(\mathbf{W}_s)}{L(\mathbf{W}_s)} + \alpha_1\eta_s^2\right),
\end{aligned}$$

where (a) uses $\frac{\mathcal{G}(\mathbf{W}_s)}{L(\mathbf{W}_s)} \leq 1$ (from Lemma C.7) to bound the quadratic term coefficient. Applying this recursively from $t_2$ to $t$:

$$L(\mathbf{W}_t) \leq L(\mathbf{W}_{t_2}) \exp\left(-\rho \sum_{s=t_2}^{t-1} \eta_s \frac{\mathcal{G}(\mathbf{W}_s)}{L(\mathbf{W}_s)} + \alpha_1 \sum_{s=t_2}^{t-1} \eta_s^2\right). \tag{24}$$

Now, consider the unnormalized margin $z_{\min}(t) := \min_{i\in[n],c\neq y_i}(\mathbf{e}_{y_i} - \mathbf{e}_c)^\top \mathbf{W}_t \mathbf{x}_i$. Since $t > t_2$, we have $L(\mathbf{W}_t) \leq \frac{\log 2}{n}$, which implies strict separability, so $z_{\min}(t) \geq 0$.

We lower bound the total loss $nL(\mathbf{W}_t)$ by focusing on the specific sample and class that achieve the minimum margin.

$$
\begin{aligned}
nL(\mathbf{W}_t) &= \sum_{j\in[n]} \log\left(1 + \sum_{c\neq y_j} e^{-(\mathbf{e}_{y_j} - \mathbf{e}_c)^\top \mathbf{W}_t \mathbf{x}_j}\right) \\
&\geq \max_{j\in[n]} \log\left(1 + \sum_{c\neq y_j} e^{-(\mathbf{e}_{y_j} - \mathbf{e}_c)^\top \mathbf{W}_t \mathbf{x}_j}\right) \\
&\geq \max_{j\in[n]} \log\left(1 + \max_{c\neq y_j} e^{-(\mathbf{e}_{y_j} - \mathbf{e}_c)^\top \mathbf{W}_t \mathbf{x}_j}\right) \\
&= \log\left(1 + e^{-\min_{j\in[n],c\neq y_j}(\mathbf{e}_{y_j} - \mathbf{e}_c)^\top \mathbf{W}_t \mathbf{x}_j}\right) \\
&= \log(1 + e^{-z_{\min}(t)}).
\end{aligned}
$$

Using the inequality $\log(1 + x) \geq (\log 2)x$ for $x \in [0, 1]$ (here $x = e^{-z_{\min}(t)} \leq 1$):

$$\log(1 + e^{-z_{\min}(t)}) \geq (\log 2)e^{-z_{\min}(t)}.$$

Combining these inequalities:

$$e^{-z_{\min}(t)} \leq \frac{n}{\log 2} L(\mathbf{W}_t).$$

Substituting the bound from Eq. (24) and using the specific property $L(\mathbf{W}_{t_2}) \leq \frac{\log 2}{n}$:

$$
\begin{aligned}
e^{-z_{\min}(t)} &\leq \frac{n}{\log 2} \cdot \left[\frac{\log 2}{n} \exp\left(-\rho \sum_{s=t_2}^{t-1} \eta_s \frac{\mathcal{G}(\mathbf{W}_s)}{L(\mathbf{W}_s)} + \alpha_1 \sum_{s=t_2}^{t-1} \eta_s^2\right)\right] \\
&= \exp\left(-\rho \sum_{s=t_2}^{t-1} \eta_s \frac{\mathcal{G}(\mathbf{W}_s)}{L(\mathbf{W}_s)} + \alpha_1 \sum_{s=t_2}^{t-1} \eta_s^2\right).
\end{aligned}
$$

The constant factors $\frac{n}{\log 2}$ and $\frac{\log 2}{n}$ cancel out exactly. Taking the natural logarithm on both sides and multiplying by $-1$ reverses the inequality:

$$z_{\min}(t) \geq \rho \sum_{s=t_2}^{t-1} \eta_s \frac{\mathcal{G}(\mathbf{W}_s)}{L(\mathbf{W}_s)} - \alpha_1 \sum_{s=t_2}^{t-1} \eta_s^2.$$

$\square$

**Theorem D.4** (Margin Convergence Rate of Stochastic $l_p$ Steepest Descent Without Momentum). *Suppose Assumptions 3.1, 3.2, and 3.3 hold. Assume the batch size $b$ satisfies the large batch condition: $\rho := \gamma - 4(\frac{n}{b} - 1)R > 0$ ($b > \frac{4Rn}{\gamma+4R}$). Let $t_2$ be the time index that $L(\mathbf{W}_t) \leq \frac{\log 2}{n}$ for all $t > t_2$. Then, for all $t > t_2$, the margin gap of the iterates satisfies:*

$$\gamma - 4\left(\frac{n}{b} - 1\right)R - \frac{\min_{i\in[n],c\neq y_i}(\mathbf{e}_{y_i} - \mathbf{e}_c)^\top \mathbf{W}_t \mathbf{x}_i}{\|\mathbf{W}_t\|} \leq \mathcal{O}\left(\frac{\sum_{s=t_2}^{t-1} \eta_s e^{-\frac{\rho}{4}\sum_{\tau=t_2}^{s-1}\eta_\tau} + \sum_{s=0}^{t_2-1}\eta_s + \sum_{s=t_2}^{t-1}\eta_s^2}{\sum_{s=0}^{t-1}\eta_s}\right).$$

*Proof.* First, we need to derive the convergence of the ratio $\frac{\mathcal{G}(\mathbf{W}_t)}{L(\mathbf{W}_t)}$. From Lemma D.2, we know that for all $t > t_2$, $L(\mathbf{W}_t) \leq \tilde{L} = \frac{\log 2}{n}$. By Lemma C.7 (ii), this implies $L(\mathbf{W}_t) \leq 2\mathcal{G}(\mathbf{W}_t)$. Recall the descent inequality for $t > t_2$ (where $\eta_t \alpha_1 \leq \rho/2$ is satisfied):

$$L(\mathbf{W}_{t+1}) \leq L(\mathbf{W}_t) - \frac{\rho}{2}\eta_t \mathcal{G}(\mathbf{W}_t) \leq L(\mathbf{W}_t) - \frac{\rho}{4}\eta_t L(\mathbf{W}_t) = L(\mathbf{W}_t)\left(1 - \frac{\rho}{4}\eta_t\right).$$

Applying this recursively from $t_2$ to $t$:

$$L(\mathbf{W}_t) \leq L(\mathbf{W}_{t_2})\exp\left(-\frac{\rho}{4}\sum_{s=t_2}^{t-1}\eta_s\right) \leq \tilde{L}\exp\left(-\frac{\rho}{4}\sum_{s=t_2}^{t-1}\eta_s\right). \tag{25}$$

Using Lemma C.7 (i), we lower bound the ratio:

$$\frac{\mathcal{G}(\mathbf{W}_t)}{L(\mathbf{W}_t)} \geq 1 - \frac{nL(\mathbf{W}_t)}{2} \geq 1 - \frac{n\tilde{L}}{2}e^{-\frac{\rho}{4}\sum_{s=t_2}^{t-1}\eta_s} \geq 1 - e^{-\frac{\rho}{4}\sum_{s=t_2}^{t-1}\eta_s}. \tag{26}$$

It's easy to get that the weight norm is upper bounded by:

$$\|\mathbf{W}_t\| \leq \|\mathbf{W}_0\| + \sum_{s=0}^{t-1}\eta_s.$$

From Lemma D.3, the unnormalized margin is lower bounded by:

$$z_{\min}(t) \geq \rho\sum_{s=t_2}^{t-1}\eta_s\frac{\mathcal{G}(\mathbf{W}_s)}{L(\mathbf{W}_s)} - \alpha_1\sum_{s=t_2}^{t-1}\eta_s^2.$$

Substituting the ratio bound (26) into the margin bound:

$$z_{\min}(t) \geq \rho\sum_{s=t_2}^{t-1}\eta_s\left(1 - e^{-\frac{\rho}{4}\sum_{\tau=t_2}^{s-1}\eta_\tau}\right) - \alpha_1\sum_{s=t_2}^{t-1}\eta_s^2$$

$$= \rho\sum_{s=t_2}^{t-1}\eta_s - \rho\sum_{s=t_2}^{t-1}\eta_s e^{-\frac{\rho}{4}\sum_{\tau=t_2}^{s-1}\eta_\tau} - \alpha_1\sum_{s=t_2}^{t-1}\eta_s^2.$$

We now compute the normalized margin gap:

$$\rho - \frac{z_{\min}(t)}{\|\mathbf{W}_t\|} = \frac{\rho\|\mathbf{W}_t\| - z_{\min}(t)}{\|\mathbf{W}_t\|}$$

$$\leq \frac{\rho\left(\|\mathbf{W}_0\| + \sum_{s=0}^{t_2-1}\eta_s + \sum_{s=t_2}^{t-1}\eta_s\right) - \left(\rho\sum_{s=t_2}^{t-1}\eta_s - \rho\sum_{s=t_2}^{t-1}\eta_s e^{-\frac{\rho}{4}\sum_{\tau=t_2}^{s-1}\eta_\tau} - \alpha_1\sum_{s=t_2}^{t-1}\eta_s^2\right)}{\sum_{s=0}^{t-1}\eta_s}$$

$$= \frac{\rho\|\mathbf{W}_0\| + \rho\sum_{s=0}^{t_2-1}\eta_s + \rho\sum_{s=t_2}^{t-1}\eta_s e^{-\frac{\rho}{4}\sum_{\tau=t_2}^{s-1}\eta_\tau} + \alpha_1\sum_{s=t_2}^{t-1}\eta_s^2}{\sum_{s=0}^{t-1}\eta_s}.$$

Since $\rho$ and $\alpha_1$ are constants, we can absorb them into the Big-O notation:

$$\rho - \frac{z_{\min}(t)}{\|\mathbf{W}_t\|} \leq \mathcal{O}\left(\frac{\sum_{s=t_2}^{t-1}\eta_s e^{-\frac{\rho}{4}\sum_{\tau=t_2}^{s-1}\eta_\tau} + \sum_{s=0}^{t_2-1}\eta_s + \sum_{s=t_2}^{t-1}\eta_s^2}{\sum_{s=0}^{t-1}\eta_s}\right).$$

This completes the proof. $\qquad\square$

**Corollary D.5.** *Consider a learning rate schedule of the form $\eta_t = c \cdot t^{-a}$ where $a \in (0, 1]$ and $c > 0$. Under the same setting as Theorem* D.4, *the margin gap converges with the following rates:*

$$\gamma - 4(\frac{n}{b} - 1)R - \frac{\min_{i \in [n], c \neq y_i}(\mathbf{e}_{y_i} - \mathbf{e}_c)^\top \mathbf{W}_t \mathbf{x}_i}{\|\mathbf{W}_t\|} = \begin{cases} \mathcal{O}\left(\frac{t^{1-2a}+n}{t^{1-a}}\right) & if \quad a < \frac{1}{2} \\ \mathcal{O}\left(\frac{\log t + n}{t^{1/2}}\right) & if \quad a = \frac{1}{2} \\ \mathcal{O}\left(\frac{n}{t^{1-a}}\right) & if \quad \frac{1}{2} < a < 1 \\ \mathcal{O}\left(\frac{n}{\log t}\right) & if \quad a = 1 \end{cases}$$

*Proof.* We analyze the three terms in the numerator and the denominator from Theorem D.4 separately.

**1. Denominator Estimation ($\sum \eta_s$):** Using integral approximation $\sum_{s=1}^{t} s^{-a} \approx \int_1^t x^{-a} dx$, we have:

$$\sum_{s=0}^{t-1} \eta_s = \begin{cases} \mathcal{O}(t^{1-a}) & \text{if } a < 1 \\ \mathcal{O}(\log t) & \text{if } a = 1 \end{cases}.$$

**2. Estimation of $t_2$ and $\sum_{s=0}^{t_2-1} \eta_s$:** Recall the condition for $t_2$ from Lemma D.2:

$$\sum_{s=t_1}^{t_2} \eta_s \geq \frac{C}{\rho \tilde{L}} = \frac{C \cdot n}{\rho \log 2},$$

where $C$ depends on $L(\mathbf{W}_0)$ and $R$, but is independent of $t$. Since $t_1$ is a constant independent of $n$ (determined only by $\rho$ and curvature), for large $n$, the LHS is dominated by the sum up to $t_2$. Thus:

$$\sum_{s=0}^{t_2-1} \eta_s = \mathcal{O}(n).$$

This directly bounds the second term in the numerator. (Note: This implies $t_2 \approx n^{\frac{1}{1-a}}$ for $a < 1$).

**3. Estimation of the Quadratic Term ($\sum \eta_s^2$):** Similarly, using integral approximation $\sum_{s=1}^{t} s^{-2a}$:

$$\sum_{s=t_2}^{t-1} \eta_s^2 = \begin{cases} \Theta(t^{1-2a}) & \text{if } a < 1/2 \\ \Theta(\log t) & \text{if } a = 1/2 \\ \Theta(1) & \text{if } a > 1/2 \quad \text{(converges to a constant)} \end{cases}.$$

**4. Estimation of the Exponential Decay Term:** The term $\sum_{s=t_2}^{t-1} \eta_s e^{-\frac{\rho}{4} \sum_{\tau=t_2}^{s-1} \eta_\tau}$ is bounded by a constant for all $a \in (0, 1]$. Let $S_s = \sum_{\tau=t_2}^{s-1} \eta_\tau$. The sum approximates the integral $\int e^{-\frac{\rho}{4}S} dS$, which converges. Thus, this term is $\mathcal{O}(1)$. Combining these estimates, let $G(t)$ denote the margin gap bound.

- $a < 1/2$: The numerator is dominated by $\sum \eta_s^2 \approx t^{1-2a}$ and the entry cost $\approx n$.

$$\text{Rate} = \mathcal{O}\left(\frac{t^{1-2a}+n}{t^{1-a}}\right).$$

- $a = 1/2$: The numerator noise term is $\log t$.

$$\text{Rate} = \mathcal{O}\left(\frac{\log t + n}{t^{1/2}}\right).$$

- $1/2 < a < 1$: The quadratic noise sum converges to a constant. The numerator is dominated by the entry cost $n$.

$$\text{Rate} = \mathcal{O}\left(\frac{n}{t^{1-a}}\right).$$

- $a = 1$: The denominator is $\log t$. The numerator is dominated by $n$.

$$\text{Rate} = \mathcal{O}\left(\frac{n}{\log t}\right).$$

$\square$

**Proposition D.6** (Failure of random reshuffling without momentum when $m = 2$)**.** *There exists a linearly separable dataset with $n = 2$, batch size $b = 1$, and hence $m = n/b = 2$, such that random-reshuffling SignSGD without momentum, initialized at $\mathbf{W}_0 = 0$, never strictly correctly classifies all training samples. Consequently, the empirical cross-entropy loss does not converge to zero.*

*Proof.* Fix any $0 < \varepsilon < 1$. Consider a two-class problem with feature dimension $d = 2$ and two training examples

$$(\mathbf{x}_1, y_1) = ((1, -\varepsilon), 1), \qquad (\mathbf{x}_2, y_2) = ((\varepsilon, -1), 2).$$

Let

$$\mathbf{u} := \mathbf{x}_1 = (1, -\varepsilon), \qquad \mathbf{v} := -\mathbf{x}_2 = (-\varepsilon, 1).$$

For

$$\mathbf{W} = \begin{pmatrix} \mathbf{w}_1^\top \\ \mathbf{w}_2^\top \end{pmatrix} \in \mathbb{R}^{2 \times 2},$$

define

$$\boldsymbol{\theta} := \mathbf{w}_1 - \mathbf{w}_2.$$

The two margins are

$$M_1(\mathbf{W}) := (\mathbf{e}_1 - \mathbf{e}_2)^\top \mathbf{W} \mathbf{x}_1 = \boldsymbol{\theta}^\top \mathbf{u},$$

and

$$M_2(\mathbf{W}) := (\mathbf{e}_2 - \mathbf{e}_1)^\top \mathbf{W} \mathbf{x}_2 = -\boldsymbol{\theta}^\top \mathbf{x}_2 = \boldsymbol{\theta}^\top \mathbf{v}.$$

The dataset is linearly separable. Indeed, taking

$$\boldsymbol{\theta}^\star = (1, 1)$$

gives

$$(\boldsymbol{\theta}^\star)^\top \mathbf{u} = 1 - \varepsilon > 0, \qquad (\boldsymbol{\theta}^\star)^\top \mathbf{v} = 1 - \varepsilon > 0.$$

Equivalently, one may choose $\mathbf{w}_1^\star = \boldsymbol{\theta}^\star/2$ and $\mathbf{w}_2^\star = -\boldsymbol{\theta}^\star/2$.

We now analyze stochastic steepest descent without momentum under the entry-wise $\ell_\infty$ geometry. For a single-sample mini-batch $\{i_t\}$, the update is

$$\mathbf{W}_{t+1} = \mathbf{W}_t - \eta_t \Delta_t, \qquad \Delta_t = \text{sign}\big(\nabla_\mathbf{W} \ell(\mathbf{W}_t \mathbf{x}_{i_t}; y_{i_t})\big),$$

where $\eta_t > 0$ and $\ell(\mathbf{W}\mathbf{x}; y) = -\log \mathbb{S}_y(\mathbf{W}\mathbf{x})$ is the softmax cross-entropy loss. Since $0 < \varepsilon < 1$, all feature coordinates are nonzero, and all softmax probabilities are strictly positive; hence all gradient coordinates appearing below are nonzero and the entry-wise sign is well-defined.

Let

$$\mathbf{s} := (1, -1).$$

First consider sample 1. Since $y_1 = 1$, we have

$$\nabla_\mathbf{W} \ell(\mathbf{W}\mathbf{x}_1; 1) = (\mathbb{S}(\mathbf{W}\mathbf{x}_1) - \mathbf{e}_1)\mathbf{x}_1^\top.$$

Writing $q_1 := \mathbb{S}_2(\mathbf{W}\mathbf{x}_1) > 0$, this gives

$$\nabla_{\mathbf{w}_1} \ell(\mathbf{W}\mathbf{x}_1; 1) = -q_1 \mathbf{x}_1, \qquad \nabla_{\mathbf{w}_2} \ell(\mathbf{W}\mathbf{x}_1; 1) = q_1 \mathbf{x}_1.$$

Since $\text{sign}(\mathbf{x}_1) = \mathbf{s}$, processing sample 1 yields

$$\mathbf{w}_1^+ = \mathbf{w}_1 + \eta_t \mathbf{s}, \qquad \mathbf{w}_2^+ = \mathbf{w}_2 - \eta_t \mathbf{s},$$

and therefore

$$\boldsymbol{\theta}^+ = \boldsymbol{\theta} + 2\eta_t \mathbf{s}.$$

Thus, whenever $i_t = 1$,

$$\boldsymbol{\theta}_{t+1} = \boldsymbol{\theta}_t + 2\eta_t \mathbf{s}.$$

Next consider sample 2. Since $y_2 = 2$,

$$\nabla_{\mathbf{W}} \ell(\mathbf{W}\mathbf{x}_2; 2) = (\mathbb{S}(\mathbf{W}\mathbf{x}_2) - \mathbf{e}_2)\mathbf{x}_2^\top.$$

Writing $q_2 := \mathbb{S}_1(\mathbf{W}\mathbf{x}_2) > 0$, we have

$$\nabla_{\mathbf{w}_1} \ell(\mathbf{W}\mathbf{x}_2; 2) = q_2 \mathbf{x}_2, \qquad \nabla_{\mathbf{w}_2} \ell(\mathbf{W}\mathbf{x}_2; 2) = -q_2 \mathbf{x}_2.$$

Since $\mathrm{sign}(\mathbf{x}_2) = \mathbf{s}$, processing sample 2 yields

$$\mathbf{w}_1^+ = \mathbf{w}_1 - \eta_t \mathbf{s}, \qquad \mathbf{w}_2^+ = \mathbf{w}_2 + \eta_t \mathbf{s},$$

and hence

$$\boldsymbol{\theta}^+ = \boldsymbol{\theta} - 2\eta_t \mathbf{s}.$$

Thus, whenever $i_t = 2$,

$$\boldsymbol{\theta}_{t+1} = \boldsymbol{\theta}_t - 2\eta_t \mathbf{s}.$$

Since $\boldsymbol{\theta}_0 = 0$, the above two update identities imply by induction that, for every $t \geq 0$, there exists a scalar $c_t \in \mathbb{R}$ such that

$$\boldsymbol{\theta}_t = c_t \mathbf{s}.$$

This invariant holds for every possible realization of random reshuffling.

Using this invariant, the two margins satisfy

$$M_1(\mathbf{W}_t) = \boldsymbol{\theta}_t^\top \mathbf{u} = c_t \mathbf{s}^\top \mathbf{u} = c_t(1 + \varepsilon),$$

whereas

$$M_2(\mathbf{W}_t) = \boldsymbol{\theta}_t^\top \mathbf{v} = c_t \mathbf{s}^\top \mathbf{v} = -c_t(1 + \varepsilon).$$

Therefore

$$M_2(\mathbf{W}_t) = -M_1(\mathbf{W}_t)$$

for every $t \geq 0$. Hence the two margins can never be simultaneously strictly positive. In particular,

$$\min\{M_1(\mathbf{W}_t), M_2(\mathbf{W}_t)\} \leq 0$$

for all $t \geq 0$. Thus the iterates never strictly correctly classify both training samples.

Finally, for binary softmax cross-entropy, the loss of a sample with margin $M$ is

$$\log(1 + \exp(-M)).$$

Since at least one of $M_1(\mathbf{W}_t)$ and $M_2(\mathbf{W}_t)$ is non-positive at every iteration, at least one sample has loss at least $\log 2$. Therefore

$$L(\mathbf{W}_t) = \frac{1}{2} \sum_{i=1}^{2} \ell(\mathbf{W}_t \mathbf{x}_i; y_i) \geq \frac{1}{2} \log 2$$

for every $t \geq 0$. Hence $L(\mathbf{W}_t) \not\to 0$, completing the proof. $\qquad \square$

# E    Proof in Section 4.2

**Lemma E.1** (Descent Lemma for Stochastic $l_p$-Steepest Descent Algorithm With momentum). *Suppose that Assumption 3.1, 3.2, 3.3 and 3.4 hold, constants $\alpha_1 = (4mR(1-\beta_1)c_2 + 2R^2e^{2R\eta_0})$, $\alpha_2 = 4R$. Then there exist a time $t_0$ such that for all $t \geq t_0$,*

$$L(\mathbf{W}_{t+1}) \leq L(\mathbf{W}_t) - \gamma\eta_t\mathcal{G}(\mathbf{W}_t) + 2(1-\beta_1)m(m^2-1)\eta_t R\mathcal{G}(\mathbf{W}_t) + \alpha_1\eta_t^2\mathcal{G}(\mathbf{W}_t) + \alpha_2\beta_1^{\frac{t}{2}}\eta_t\mathcal{G}(\mathbf{W}_t),$$

*Proof.* Similarly, by Lemma C.2, let $\tilde{\mathbf{\Delta}}_t = \mathbf{W}_{t+1} - \mathbf{W}_t$, and define $\mathbf{W}_{t,t+1,\zeta} := \mathbf{W}_t + \zeta(\mathbf{W}_{t+1} - \mathbf{W}_t)$. We choose $\zeta^*$ such that $\mathbf{W}_{t,t+1,\zeta^*}$ satisfies (7), then we have:

$$L(\mathbf{W}_{t+1}) = L(\mathbf{W}_t) + \underbrace{\langle \nabla L(\mathbf{W}_t), \mathbf{W}_{t+1} - \mathbf{W}_t \rangle}_{A}$$

$$+ \underbrace{\frac{1}{2n}\sum_{i\in[n]} \mathbf{x}_i^\top \tilde{\mathbf{\Delta}}_t^\top \left(\mathrm{diag}(\mathbb{S}(\mathbf{W}_{t,t+1,\zeta^*}\mathbf{x}_i)) - \mathbb{S}(\mathbf{W}_{t,t+1,\zeta^*}\mathbf{x}_i)\mathbb{S}(\mathbf{W}_{t,t+1,\zeta^*}\mathbf{x}_i)^\top\right)\tilde{\mathbf{\Delta}}_t\,\mathbf{x}_i}_{B}\,. \tag{27}$$

For Term A:

$$\langle \nabla L(\mathbf{W}_t), \mathbf{W}_{t+1} - \mathbf{W}_t \rangle = \langle \nabla L(\mathbf{W}_t) - \mathbf{M}_t, \mathbf{W}_{t+1} - \mathbf{W}_t \rangle + \langle \mathbf{M}_t, \mathbf{W}_{t+1} - \mathbf{W}_t \rangle$$

$$= -\eta_t\langle \nabla L(\mathbf{W}_{t+1}) - \mathbf{M}_t, \Delta_t \rangle - \eta_t\langle \mathbf{M}_t, \Delta_t \rangle$$

$$\overset{(a)}{\leq} \eta_t\|\nabla L(\mathbf{W}_t) - \mathbf{M}_t\|_*\|\Delta_t\| - \eta_t\|\mathbf{M}_t\|_*$$

$$\overset{(b)}{\leq} \eta_t\|\nabla L(\mathbf{W}_t) - \mathbf{M}_t\|_* - \eta_t\|\mathbf{M}_t - \nabla L(\mathbf{W}_t) + \nabla L(\mathbf{W}_t)\|_*$$

$$\overset{(c)}{\leq} \eta_t\|\nabla L(\mathbf{W}_t) - \mathbf{M}_t\|_{\mathrm{sum}} - \eta_t(\|\nabla L(\mathbf{W}_t)\|_* - \|\nabla L(\mathbf{W}_t) - \mathbf{M}_t\|_*)$$

$$= 2\eta_t\|\nabla L(\mathbf{W}_t) - \mathbf{M}_t\|_{\mathrm{sum}} - \eta\|\nabla L(\mathbf{W}_t)\|_*$$

$$\overset{(d)}{\leq} \underbrace{2\eta_t\|\nabla L(\mathbf{W}_t) - \mathbf{M}_t\|_{\mathrm{sum}}}_{A_1} -\eta_t\gamma\mathcal{G}(\mathbf{W}),$$

where (a) is by Cauchy Schwarz inequality and $\langle \mathbf{M}_t, \mathbf{\Delta} \rangle = \|\mathbf{M}_t\|_*$, (b) is by $\|\mathbf{\Delta}\| \leq 1$, (c) is via Lemma C.1 and Triangle inequality and (e) is via Lemma C.5.

To bound Term A1, we first need to decompose it using $\mathbf{M}_t = \sum_{\tau=0}^t (1-\beta_1)\beta_1^\tau \nabla L_{\mathcal{B}_{t-\tau}}(\mathbf{W}_{t-\tau})$.

$$\|\mathbf{M}_t - \nabla L(\mathbf{W}_t)\|_{\mathrm{sum}} = \left\|\sum_{\tau=0}^t (1-\beta_1)\beta_1^\tau \nabla L_{\mathcal{B}_{t-\tau}}(\mathbf{W}_{t-\tau}) - \sum_{\tau=0}^t (1-\beta_1)\beta_1^\tau \nabla L(\mathbf{W}_t) + \beta_1^{t+1}\nabla L(\mathbf{W}_t)\right\|_{\mathrm{sum}}$$

$$\leq \underbrace{\left\|\sum_{\tau=0}^t (1-\beta_1)\beta_1^\tau \nabla L_{\mathcal{B}_{t-\tau}}(\mathbf{W}_{t-\tau}) - \sum_{\tau=0}^t (1-\beta_1)\beta_1^\tau \nabla L_{\mathcal{B}_{t-\tau}}(\mathbf{W}_t)\right\|_{\mathrm{sum}}}_{A_{1,1}}$$

$$+ \underbrace{\left\|\sum_{\tau=0}^t (1-\beta_1)\beta_1^\tau \nabla L_{\mathcal{B}_{t-\tau}}(\mathbf{W}_t) - \sum_{\tau=0}^t (1-\beta_1)\beta_1^\tau \nabla L(\mathbf{W}_t)\right\|_{\mathrm{sum}}}_{A_{1,2}} + \underbrace{\left\|\beta_1^{t+1}\nabla L(\mathbf{W}_t)\right\|_{\mathrm{sum}}}_{A_{1,3}}$$

To bound term $A_{1,1}$, we rely on the single-sample stability property derived in the proof of Lemma C.10, combined with the definition of the mini-batch gradient.

First, consider the gradient difference for a single sample $i$. Following the derivation in the proof of Lemma C.10, setting the base point as $\mathbf{W}_t$ and the perturbed point as $\mathbf{W}_{t-\tau}$, we have the per-sample bound:

$$\|\nabla \ell_i(\mathbf{W}_{t-\tau}) - \nabla \ell_i(\mathbf{W}_t)\|_{\mathrm{sum}} \leq 2R\left(e^{2R\|\mathbf{W}_{t-\tau}-\mathbf{W}_t\|_{\mathrm{max}}} - 1\right)(1 - \mathbb{S}_{y_i}(\mathbf{W}_t\mathbf{x}_i)).$$

Now, we analyze the mini-batch gradient difference. Recall that $\nabla L_{\mathcal{B}_{t-\tau}}(\mathbf{W}) = \frac{1}{b}\sum_{i\in\mathcal{B}_{t-\tau}}\nabla\ell_i(\mathbf{W})$. Applying the triangle inequality and the per-sample bound above:

$$
\begin{aligned}
\left\|\nabla L_{\mathcal{B}_{t-\tau}}(\mathbf{W}_{t-\tau}) - \nabla L_{\mathcal{B}_{t-\tau}}(\mathbf{W}_t)\right\|_{\mathrm{sum}} &= \left\|\frac{1}{b}\sum_{i\in\mathcal{B}_{t-\tau}}\left(\nabla\ell_i(\mathbf{W}_{t-\tau}) - \nabla\ell_i(\mathbf{W}_t)\right)\right\|_{\mathrm{sum}} \\
&\leq \frac{1}{b}\sum_{i\in\mathcal{B}_{t-\tau}}\left\|\nabla\ell_i(\mathbf{W}_{t-\tau}) - \nabla\ell_i(\mathbf{W}_t)\right\|_{\mathrm{sum}} \\
&\leq \frac{1}{b}\sum_{i\in\mathcal{B}_{t-\tau}} 2R\left(e^{2R\|\mathbf{W}_{t-\tau}-\mathbf{W}_t\|_{\mathrm{max}}} - 1\right)\left(1 - \mathbb{S}_{y_i}(\mathbf{W}_t\mathbf{x}_i)\right).
\end{aligned}
$$

Since the term $(1 - \mathbb{S}_{y_i}(\mathbf{W}_t\mathbf{x}_i))$ is non-negative for all $i$, the sum over the mini-batch $\mathcal{B}_{t-\tau}\subset[n]$ is upper bounded by the sum over the entire dataset $[n]$. Using the relation $m = n/b$, we have:

$$
\begin{aligned}
\frac{1}{b}\sum_{i\in\mathcal{B}_{t-\tau}}\left(1 - \mathbb{S}_{y_i}(\mathbf{W}_t\mathbf{x}_i)\right) &\leq \frac{1}{b}\sum_{i\in[n]}\left(1 - \mathbb{S}_{y_i}(\mathbf{W}_t\mathbf{x}_i)\right) \\
&= \frac{n}{b}\cdot\frac{1}{n}\sum_{i\in[n]}\left(1 - \mathbb{S}_{y_i}(\mathbf{W}_t\mathbf{x}_i)\right) \\
&= m\mathcal{G}(\mathbf{W}_t).
\end{aligned}
$$

Substituting this back, we obtain the bound for the gradient difference term:

$$
\left\|\nabla L_{\mathcal{B}_{t-\tau}}(\mathbf{W}_{t-\tau}) - \nabla L_{\mathcal{B}_{t-\tau}}(\mathbf{W}_t)\right\|_{\mathrm{sum}} \leq 2mR\left(e^{2R\|\mathbf{W}_{t-\tau}-\mathbf{W}_t\|_{\mathrm{max}}} - 1\right)\mathcal{G}(\mathbf{W}_t).
$$

Next, we bound the weight difference $\|\mathbf{W}_{t-\tau} - \mathbf{W}_t\|_{\mathrm{max}}$. Since $\mathbf{W}_t - \mathbf{W}_{t-\tau} = \sum_{j=0}^{\tau-1} -\eta_{t-1-j}\mathbf{\Delta}_{t-1-j}$ and $\|\mathbf{\Delta}\|_{\mathrm{max}} \leq 1$, we have:

$$
\|\mathbf{W}_{t-\tau} - \mathbf{W}_t\|_{\mathrm{max}} \leq \sum_{j=1}^{\tau}\eta_{t-j}.
$$

Substituting this into the expression for $A_{1,1}$:

$$
\begin{aligned}
A_{1,1} &\leq \sum_{\tau=0}^{t}(1-\beta_1)\beta_1^{\tau}\left[2mR\left(e^{2R\sum_{j=1}^{\tau}\eta_{t-j}} - 1\right)\mathcal{G}(\mathbf{W}_t)\right] \\
&= 2mR(1-\beta_1)\mathcal{G}(\mathbf{W}_t)\sum_{\tau=0}^{t}\beta_1^{\tau}\left(e^{2R\sum_{j=1}^{\tau}\eta_{t-j}} - 1\right).
\end{aligned}
$$

Now, we directly apply Assumption 3.4. By setting constants $c_1 = 2R$, there exists a constant $c_2$ such that when $t > t_0$, the summation is bounded by $c_2\eta_t$:

$$
\sum_{\tau=0}^{t}\beta_1^{\tau}\left(e^{2R\sum_{j=1}^{\tau}\eta_{t-j}} - 1\right) \leq c_2\eta_t.
$$

Therefore, we obtain the final bound for $A_{1,1}$:

$$
A_{1,1} \leq 2mR(1-\beta_1)c_2\eta_t\mathcal{G}(\mathbf{W}_t). \tag{28}
$$

Then, we apply Lemma C.12 to bound Term $A_{1,2}$ and get:

$$
A_{1,2} \leq (1-\beta_1)m(m^2-1)R\mathcal{G}(\mathbf{W}_t). \tag{29}
$$

And using Lemma C.5, we have:

$$
A_{1,3} \leq 2R\beta_1^{t+1}\mathcal{G}(\mathbf{W}_t). \tag{30}
$$

Putting Eq. 28, 29, 30 together, we can get the upper bound for Term $A_1$:

$$A_1 \leq 2mR(1 - \beta_1)c_2\eta_t\mathcal{G}(\mathbf{W}_t) + (1 - \beta_1)m(m^2 - 1)R\mathcal{G}(\mathbf{W}_t) + 2R\beta_1^{t+1}\mathcal{G}(\mathbf{W}_t). \tag{31}$$

Finally, for Term B, We just follow the same steps in Lemma D.1 and get:

$$TermB \leq 2\eta_t^2 R^2 e^{2R\eta_0}\mathcal{G}(\mathbf{W}_t).$$

Combining Terms A, B together, we obtain

$$L(\mathbf{W}_{t+1}) \leq L(\mathbf{W}_t) - \gamma\eta_t\mathcal{G}(\mathbf{W}_t) + 4mR(1 - \beta_1)c_2\eta_t^2\mathcal{G}(\mathbf{W}_t)$$
$$+ 2(1 - \beta_1)m(m^2 - 1)\eta_t R\mathcal{G}(\mathbf{W}_t) + 4\eta_t R\beta_1^{t+1}\mathcal{G}(\mathbf{W}_t) + 2\eta_t^2 R^2 e^{2R\eta_0}\mathcal{G}(\mathbf{W}_t)$$

using additional notation $\alpha_1 = (4mR(1 - \beta_1)c_2 + 2R^2 e^{2R\eta_0})$, $\alpha_2 = 4R$ and inequality $\beta_1^{t+1} < \beta_1^{\frac{t}{2}}$, we can simplify the bound as:

$$L(\mathbf{W}_{t+1}) \leq L(\mathbf{W}_t) - \gamma\eta_t\mathcal{G}(\mathbf{W}_t) + 2(1 - \beta_1)m(m^2 - 1)\eta_t R\mathcal{G}(\mathbf{W}_t) + \alpha_1\eta_t^2\mathcal{G}(\mathbf{W}_t) + \alpha_2\beta_1^{\frac{t}{2}}\eta_t\mathcal{G}(\mathbf{W}_t) \tag{32}$$

which conclude the proof. $\qquad\square$

Eq. 32 implies that in **large batch or high momentum settings**, specifically where $\beta_1 \to 1$ or $b \to n$ such that the effective margin $\rho = \gamma - 2(1 - \beta_1)m(m^2 - 1)R$ is strictly positive—the loss eventually exhibits monotonic decrease for sufficiently large $t$ under a decaying learning rate schedule.

**Lemma E.2** (Loss Convergence). *Suppose Assumptions 3.1, 3.2, 3.3 and 3.4 hold. Assume the parameters satisfy the Positive Effective Margin Condition:*

$$\rho := \gamma - 2(1 - \beta_1)m(m^2 - 1)R > 0.$$

*Then, there exists a time index $t_2$ such that for all $t > t_2$, $L(\mathbf{W}_t) \leq \frac{\log 2}{n}$. The sufficient condition for $t_2$ is determined by the accumulated step sizes:*

$$\sum_{s=t_1}^{t_2} \eta_s \geq \frac{2L(\mathbf{W}_0) + 4R\sum_{s=0}^{t_1-1}\eta_s}{\rho \cdot (\frac{\log 2}{n})}, \tag{33}$$

*where $t_1$ is the time after which the descent term dominates the noise terms (i.e., $\alpha_1\eta_t + \alpha_2\beta_1^{t/2} \leq \rho/2$).*

*Proof.* The proof follows the same two-step strategy as Lemma D.2 (Determination of $t_1$ and $t_2$), adapted for the descent inequality derived in Lemma E.1.

Recall the descent inequality (Eq. 32):

$$L(\mathbf{W}_{t+1}) \leq L(\mathbf{W}_t) - \eta_t\left(\rho - \alpha_1\eta_t - \alpha_2\beta_1^{t/2}\right)\mathcal{G}(\mathbf{W}_t).$$

Since $\eta_t \to 0$ and $\beta_1 < 1$ implies $\beta_1^{t/2} \to 0$, there exists a finite time $t_1$ such that for all $t \geq t_1$, the noise terms are dominated by the effective margin:

$$\alpha_1\eta_t + \alpha_2\beta_1^{t/2} \leq \frac{\rho}{2}.$$

Consequently, for all $t \geq t_1$, the loss is strictly decreasing:

$$L(\mathbf{W}_{t+1}) \leq L(\mathbf{W}_t) - \frac{\rho}{2}\eta_t\mathcal{G}(\mathbf{W}_t). \tag{34}$$

Summing the strict descent inequality (34) from $t_1$ to $t_2$:

$$\frac{\rho}{2}\sum_{s=t_1}^{t_2}\eta_s\mathcal{G}(\mathbf{W}_s) \leq L(\mathbf{W}_{t_1}) - L(\mathbf{W}_{t_2+1}) \leq L(\mathbf{W}_{t_1}).$$

Let $t^* = \operatorname{argmin}_{s \in [t_1, t_2]} \mathcal{G}(\mathbf{W}_s)$. Then:

$$\mathcal{G}(\mathbf{W}_{t^*}) \sum_{s=t_1}^{t_2} \eta_s \leq \frac{2}{\rho} L(\mathbf{W}_{t_1}).$$

Let $\tilde{L} = \frac{\log 2}{n}$. To ensure $L(\mathbf{W}_{t^*}) \leq \tilde{L}$, it suffices to enforce $\mathcal{G}(\mathbf{W}_{t^*}) \leq \frac{\tilde{L}}{2}$ (by Lemma C.7). This leads to the condition:

$$\frac{2(L(\mathbf{W}_0) + 2R \sum_{s=0}^{t_1-1} \eta_s)}{\rho \sum_{s=t_1}^{t_2} \eta_s} \leq \frac{\tilde{L}}{2} \iff \sum_{s=t_1}^{t_2} \eta_s \geq \frac{4L(\mathbf{W}_0) + 8R \sum_{s=0}^{t_1-1} \eta_s}{\rho \tilde{L}}.$$

Under this condition, there exists $t^* \leq t_2$ with low loss. Due to monotonicity for $t \geq t_1$, for all $t > t_2$, we have $L(\mathbf{W}_t) \leq L(\mathbf{W}_{t^*}) \leq \tilde{L}$. $\qquad \square$

**Lemma E.3** (Unnormalized Margin). *Consider the same setting as Lemma E.2. Let $t_2$ be the time index guaranteed by Lemma E.2 such that $L(\mathbf{W}_t) \leq \frac{\log 2}{n}$ for all $t > t_2$. Recall the constants: $\rho = \gamma - 2(1 - \beta_1)m(m^2 - 1)R$, $\alpha_1 = 4mR(1 - \beta_1)c_2 + 2R^2 e^{2R\eta_0}$, and $\alpha_2 = 4R$. Define the constant $D := \frac{\alpha_2 \eta_0}{1 - \sqrt{\beta_1}}$.*

*Then, for all $t > t_2$, the minimum unnormalized margin satisfies:*

$$\min_{i \in [n], c \neq y_i} (\mathbf{e}_{y_i} - \mathbf{e}_c)^\top \mathbf{W}_t \mathbf{x}_i \geq \rho \sum_{s=t_2}^{t-1} \eta_s \frac{\mathcal{G}(\mathbf{W}_s)}{L(\mathbf{W}_s)} - \alpha_1 \sum_{s=t_2}^{t-1} \eta_s^2 - D. \tag{35}$$

*Proof.* The proof mirrors the structure of Lemma D.3, adapted for the momentum-dependent descent inequality.

Recall the descent inequality (Eq. 32):

$$L(\mathbf{W}_{s+1}) \leq L(\mathbf{W}_s) - \eta_s \left( \rho - \alpha_1 \eta_s - \alpha_2 \beta_1^{s/2} \right) \mathcal{G}(\mathbf{W}_s).$$

We rewrite this recurrence by factoring out $L(\mathbf{W}_s)$:

$$L(\mathbf{W}_{s+1}) \leq L(\mathbf{W}_s) \left( 1 - \eta_s \rho \frac{\mathcal{G}(\mathbf{W}_s)}{L(\mathbf{W}_s)} + \left( \alpha_1 \eta_s^2 + \alpha_2 \eta_s \beta_1^{s/2} \right) \frac{\mathcal{G}(\mathbf{W}_s)}{L(\mathbf{W}_s)} \right).$$

Using the inequality $1 + x \leq e^x$ and the property $\frac{\mathcal{G}(\mathbf{W}_s)}{L(\mathbf{W}_s)} \leq 1$ (Lemma C.7 (i)) to bound the noise terms, we have:

$$L(\mathbf{W}_{s+1}) \leq L(\mathbf{W}_s) \exp \left( -\eta_s \rho \frac{\mathcal{G}(\mathbf{W}_s)}{L(\mathbf{W}_s)} + \alpha_1 \eta_s^2 + \alpha_2 \eta_s \beta_1^{s/2} \right).$$

Applying this recursively from $t_2$ to $t$:

$$L(\mathbf{W}_t) \leq L(\mathbf{W}_{t_2}) \exp \left( -\rho \sum_{s=t_2}^{t-1} \eta_s \frac{\mathcal{G}(\mathbf{W}_s)}{L(\mathbf{W}_s)} + \alpha_1 \sum_{s=t_2}^{t-1} \eta_s^2 + \alpha_2 \sum_{s=t_2}^{t-1} \eta_s \beta_1^{s/2} \right). \tag{36}$$

Now, relate the loss to the unnormalized margin $z_{min}(t) := \min_{i \in [n], c \neq y_i} (\mathbf{e}_{y_i} - \mathbf{e}_c)^\top \mathbf{W}_t \mathbf{x}_i$. Since $t > t_2$, the data is separable ($z_i \geq 0$). Using the standard inequality derived in Lemma D.3:

$$e^{-z_{min}(t)} \leq \frac{n}{\log 2} L(\mathbf{W}_t).$$

Substituting the bound from Eq. (36) and utilizing the condition $L(\mathbf{W}_{t_2}) \leq \frac{\log 2}{n}$:

$$e^{-z_{min}(t)} \leq \frac{n}{\log 2} \cdot \left[ \frac{\log 2}{n} \exp(\dots) \right]$$

$$= \exp \left( -\rho \sum_{s=t_2}^{t-1} \eta_s \frac{\mathcal{G}(\mathbf{W}_s)}{L(\mathbf{W}_s)} + \alpha_1 \sum_{s=t_2}^{t-1} \eta_s^2 + \alpha_2 \sum_{s=t_2}^{t-1} \eta_s \beta_1^{s/2} \right).$$

Taking the negative logarithm on both sides reverses the inequality:

$$z_{min}(t) \geq \rho \sum_{s=t_2}^{t-1} \eta_s \frac{\mathcal{G}(\mathbf{W}_s)}{L(\mathbf{W}_s)} - \alpha_1 \sum_{s=t_2}^{t-1} \eta_s^2 - \alpha_2 \sum_{s=t_2}^{t-1} \eta_s \beta_1^{s/2}.$$

Finally, we bound the last term (the momentum drift noise) by a constant. Since the step size is non-increasing ($\eta_s \leq \eta_0$) and $\beta_1 < 1$, the series converges:

$$\alpha_2 \sum_{s=t_2}^{t-1} \eta_s \beta_1^{s/2} \leq \alpha_2 \eta_0 \sum_{s=0}^{\infty} (\sqrt{\beta_1})^s = \frac{\alpha_2 \eta_0}{1 - \sqrt{\beta_1}} = D.$$

Substituting this constant bound yields the desired result:

$$z_{min}(t) \geq \rho \sum_{s=t_2}^{t-1} \eta_s \frac{\mathcal{G}(\mathbf{W}_s)}{L(\mathbf{W}_s)} - \alpha_1 \sum_{s=t_2}^{t-1} \eta_s^2 - D.$$

$\square$

**Theorem E.4** (Margin Convergence Rate of Stochastic $l_p$ Steepest Descent With Momentum). *Suppose Assumptions 3.1, 3.2, 3.3, and 3.4 hold. Assume the hyperparameters (momentum $\beta_1$ and batch size $b$) satisfy the Positive Effective Margin Condition:*

$$\rho := \gamma - 2(1 - \beta_1)m(m^2 - 1)R > 0, \tag{37}$$

*where $m = n/b$. This condition holds in either of the following regimes:*

- ***High Momentum:*** *$\beta_1 \to 1$ (such that the noise term vanishes).*

- ***Large Batch:*** *$b \to n$ (implies $m \to 1$, such that $m^2 - 1 \to 0$).*

*Let $t_2$ be the time index guaranteed by Lemma E.2 such that $L(\mathbf{W}_t) \leq \frac{\log 2}{n}$. Let $D = \frac{4R\eta_0}{1 - \sqrt{\beta_1}}$ be the constant bound for momentum drift. Consider the learning rate schedule $\eta_t = c \cdot t^{-a}$ with $a \in (0, 1]$.*

*Then, for all $t > t_2$, the margin gap of the iterates satisfies:*

$$\rho - \frac{\min_{i \in [n], c \neq y_i}(\mathbf{e}_{y_i} - \mathbf{e}_c)^\top \mathbf{W}_t \mathbf{x}_i}{\|\mathbf{W}_t\|} \leq \mathcal{O}\left( \frac{\sum_{s=t_2}^{t-1} \eta_s e^{-\frac{\rho}{4} \sum_{\tau=t_2}^{s-1} \eta_\tau} + \sum_{s=0}^{t_2-1} \eta_s + \frac{m}{1-\beta_1} \sum_{s=t_2}^{t-1} \eta_s^2 + D}{\sum_{s=0}^{t-1} \eta_s} \right).$$

*Proof.* From Lemma E.2, for all $t > t_2$, we have $L(\mathbf{W}_t) \leq \frac{\log 2}{n}$. Lemma C.7 (ii) implies $L(\mathbf{W}_t) \leq 2\mathcal{G}(\mathbf{W}_t)$. Recall the descent inequality for $t > t_2$ (where the noise terms are dominated by $\rho/2$):

$$L(\mathbf{W}_{t+1}) \leq L(\mathbf{W}_t) - \frac{\rho}{2}\eta_t \mathcal{G}(\mathbf{W}_t) \leq L(\mathbf{W}_t)\left(1 - \frac{\rho}{4}\eta_t\right).$$

Recursively applying this yields exponential decay:

$$L(\mathbf{W}_t) \leq \frac{\log 2}{n} \exp\left(-\frac{\rho}{4} \sum_{s=t_2}^{t-1} \eta_s\right).$$

Using Lemma C.7 (i), the ratio is bounded as:

$$\frac{\mathcal{G}(\mathbf{W}_t)}{L(\mathbf{W}_t)} \geq 1 - \frac{nL(\mathbf{W}_t)}{2} \geq 1 - e^{-\frac{\rho}{4} \sum_{s=t_2}^{t-1} \eta_s}. \tag{38}$$

The weight growth is bounded by the cumulative step size:

$$\|\mathbf{W}_t\| \leq \|\mathbf{W}_0\| + \sum_{s=0}^{t-1} \eta_s. \tag{39}$$

From Lemma E.3, for $t > t_2$:

$$z_{\min}(t) \geq \rho \sum_{s=t_2}^{t-1} \eta_s \frac{\mathcal{G}(\mathbf{W}_s)}{L(\mathbf{W}_s)} - \alpha_1 \sum_{s=t_2}^{t-1} \eta_s^2 - D.$$

Substituting Eq. (38):

$$z_{\min}(t) \geq \rho \sum_{s=t_2}^{t-1} \eta_s \left( 1 - e^{-\frac{\rho}{4} \sum_{\tau=t_2}^{s-1} \eta_\tau} \right) - \alpha_1 \sum_{s=t_2}^{t-1} \eta_s^2 - D$$

$$= \rho \sum_{s=t_2}^{t-1} \eta_s - \rho \sum_{s=t_2}^{t-1} \eta_s e^{-\frac{\rho}{4} \sum_{\tau=t_2}^{s-1} \eta_\tau} - \alpha_1 \sum_{s=t_2}^{t-1} \eta_s^2 - D.$$

The margin gap is:

$$\text{Gap}_t = \rho - \frac{z_{\min}(t)}{\|\mathbf{W}_t\|} = \frac{\rho\|\mathbf{W}_t\| - z_{\min}(t)}{\|\mathbf{W}_t\|}$$

$$\leq \frac{\rho(\|\mathbf{W}_0\| + \sum_{s=0}^{t-1} \eta_s) - (\rho \sum_{s=t_2}^{t-1} \eta_s - \rho \sum_{s=t_2}^{t-1} \eta_s e^{-\frac{\rho}{4} \sum_{\tau=t_2}^{s-1} \eta_\tau} - \alpha_1 \sum_{s=t_2}^{t-1} \eta_s^2 - D)}{\sum_{s=0}^{t-1} \eta_s}$$

$$= \frac{\rho\|\mathbf{W}_0\| + \rho \sum_{s=0}^{t_2-1} \eta_s + \rho \sum_{s=t_2}^{t-1} \eta_s e^{-\frac{\rho}{4} \sum_{\tau=t_2}^{s-1} \eta_\tau} + \alpha_1 \sum_{s=t_2}^{t-1} \eta_s^2 + D}{\sum_{s=0}^{t-1} \eta_s}.$$

By Lemma C.15, $c_2 = \mathcal{O}(1/(1 - \beta_1)^2)$, so that $\alpha_1$ scales with $\mathcal{O}(m/(1 - \beta_1))$. Absorbing the constants $\rho, \alpha_1$ into the Big-O notation, we obtain the stated result. $\square$

**Corollary E.5.** *Consider the learning rate schedule $\eta_t = c \cdot t^{-a}$ with $a \in (0, 1]$. Under the setting of Theorem E.4, the margin gap converges with the following rates:*

$$\rho - \frac{\min_{i \in [n], c \neq y_i} (\mathbf{e}_{y_i} - \mathbf{e}_c)^\top \mathbf{W}_t \mathbf{x}_i}{\|\mathbf{W}_t\|} = \begin{cases} \mathcal{O}\left( \frac{n[\frac{m}{1-\beta_1}]^{\frac{1}{a}-1} + \frac{m}{1-\beta_1} t^{1-2a}}{t^{1-a}} \right) & \text{if} \quad a < \frac{1}{2} \\ \mathcal{O}\left( \frac{\frac{m}{1-\beta_1}(n + \log t)}{t^{1/2}} \right) & \text{if} \quad a = \frac{1}{2} \\ \mathcal{O}\left( \frac{n[\frac{m}{1-\beta_1}]^{\frac{1}{a}-1} + \frac{m}{1-\beta_1}}{t^{1-a}} \right) & \text{if} \quad \frac{1}{2} < a < 1 \\ \mathcal{O}\left( \frac{n \log(\frac{m}{1-\beta_1}) + \frac{m}{1-\beta_1}}{\log t} \right) & \text{if} \quad a = 1 \end{cases}$$

*If we view $\beta_1$ as a constant for better comparison, we have:*

$$\rho - \frac{\min_{i \in [n], c \neq y_i} (\mathbf{e}_{y_i} - \mathbf{e}_c)^\top \mathbf{W}_t \mathbf{x}_i}{\|\mathbf{W}_t\|} = \begin{cases} \mathcal{O}\left( \frac{nm^{\frac{1}{a}-1} + mt^{1-2a}}{t^{1-a}} \right) & \text{if} \quad a < \frac{1}{2} \\ \mathcal{O}\left( \frac{nm + m \log t}{t^{1/2}} \right) & \text{if} \quad a = \frac{1}{2} \\ \mathcal{O}\left( \frac{nm^{\frac{1}{a}-1}}{t^{1-a}} \right) & \text{if} \quad \frac{1}{2} < a < 1 \\ \mathcal{O}\left( \frac{n \log m}{\log t} \right) & \text{if} \quad a = 1 \end{cases}$$

*Proof.* We derive the convergence rate by systematically analyzing the order of the numerator terms in Theorem E.4 and dividing by the denominator $\sum_{s=0}^{t-1} \eta_s$. Let $\xi := \frac{m}{1-\beta_1}$ denote the scaling factor of the noise constant $\alpha_1$.

The bound in Theorem E.4 is:

$$\text{Gap}_t \leq \mathcal{O}\left( \frac{\mathcal{S}_2 + \mathcal{N}_t + D}{\sum_{s=0}^{t-1} \eta_s} \right),$$

where $\mathcal{S}_2 := \sum_{s=0}^{t_2-1} \eta_s$, $\mathcal{N}_t := \xi \sum_{s=t_2}^{t-1} \eta_s^2$ and $D = \mathcal{O}((1 - \beta_1)^{-1})$.

**Step 1: Order of the Start Time $t_1$ and Warm-up Drift $\mathcal{S}_1$.** The time $t_1$ is defined as the first iteration where the effective margin dominates the noise terms. Specifically, we require $t_1 \geq t_0$ (from Assumption 3.4), $\alpha_{\text{svr-m}}\eta_{t_1} \leq \rho/4$, and $\alpha_2 \beta_1^{t_1/2} \leq \rho/4$. Thus:

$$t_1 = \max\{t_{1,\text{poly}}, t_{1,\text{exp}}, t_0\}.$$

We analyze the order of the cumulative step size $\mathcal{S}_1 = \sum_{s=0}^{t_1-1} \eta_s$ contributed by each component. Note that for large $T$, $\sum_{s=0}^{\top} s^{-a} = \mathcal{O}(T^{1-a})$ for $a < 1$ and $\mathcal{O}(\log T)$ for $a = 1$.

1. **Polynomial Stability ($t_{1,\text{poly}}$):** From $\xi t^{-a} \leq \mathcal{O}(1)$, we have $t_{1,\text{poly}} = \Theta(\xi^{1/a})$. The contribution to the sum is:

$$\sum_{s=0}^{t_{1,\text{poly}}} \eta_s = \mathcal{O}\left((\xi^{1/a})^{1-a}\right) = \mathcal{O}\left(\xi^{\frac{1}{a}-1}\right). \quad \text{(For } a = 1 : \log \xi\text{)}.$$

2. **Exponential Stability ($t_{1,\text{exp}}$):** From $\beta_1^{t/2} \leq \mathcal{O}(1)$, we have $t_{1,\text{exp}} = \Theta(\frac{1}{\log(1/\beta_1)})$. The contribution to the sum is:

$$\sum_{s=0}^{t_{1,\text{exp}}} \eta_s = \Theta\left((\log(1/\beta_1))^{a-1}\right). \quad \text{(For } a = 1 : \log(\frac{1}{\log(1/\beta_1)})\text{)}.$$

3. **Assumption 3.4 Validity ($t_0$):** We explicitly invoke Lemma C.15. The order of $t_0$ is given in Eq. (15) and (16). Define

$$\tilde{t} = \begin{cases} (1/\log(1/\beta_1))^{\frac{1}{a}} + \frac{1}{\log(1/\beta_1)} \log(\frac{1}{\log(1/\beta_1)}), & \text{if } a \in (0,1) \\ \frac{1}{\log(1/\beta_1)} \log(\frac{1}{\log(1/\beta_1)}), & \text{if } a = 1 \end{cases}$$

The variable $\tilde{t}$ represents the integral of the learning rate up to this time $t_0$. Thus:

$$\sum_{s=0}^{t_0} \eta_s = \mathcal{O}(\tilde{t}^{1-a}). \quad \text{(For } a = 1 : \log \tilde{t}\text{)}.$$

Summing these contributions, the total warm-up drift is:

$$\mathcal{S}_1 = \mathcal{O}\left(\xi^{\frac{1}{a}-1} + (\log(1/\beta_1))^{a-1} + \tilde{t}^{1-a}\right).$$

(With logarithmic modifications for $a = 1$).

**Step 2: Order of Entry Cost $\mathcal{S}_2$.** From Lemma E.2, the condition for $t_2$ is $\sum_{s=t_1}^{t_2} \eta_s \geq \frac{C}{\gamma \tilde{L}}(2L(\mathbf{W}_0) + 4R\mathcal{S}_1)$. Since $\tilde{L} = \Theta(1/n)$, the required integral scales linearly with $n$. Therefore, the total accumulation up to $t_2$ is dominated by $n$ times the warm-up drift:

$$\mathcal{S}_2 = \sum_{s=0}^{t_2} \eta_s = \mathcal{O}(n \cdot \mathcal{S}_1).$$

**Step 3: Order of Variance Noise $\mathcal{N}_t$.** The noise term is $\mathcal{N}_t = \Theta(\xi \sum_{s=t_2}^{t-1} \eta_s^2)$. We approximate the sum by integral $\int^t x^{-2a} dx$:

$$\mathcal{N}_t = \begin{cases} \mathcal{O}(\xi t^{1-2a}) & \text{if } a < 1/2, \\ \mathcal{O}(\xi \log t) & \text{if } a = 1/2, \\ \mathcal{O}(\xi) & \text{if } a > 1/2 \quad \text{(converges to constant).} \end{cases}$$

**Step 4: Final Rate** The denominator is $\sum_{s=0}^{t-1} \eta_s = \mathcal{O}(t^{1-a})$ (or $\log t$ if $a = 1$). The numerator is $\mathcal{S}_2 + \mathcal{N}_t + D$, where $D = \mathcal{O}((1-\beta_1)^{-1})$.

Before proceeding to the case-by-case discussion, we show that several numerator terms can be absorbed into the leading ones. Recall $\xi := \frac{m}{1-\beta_1}$ and let $L := \log(1/\beta_1) > 0$.

**Absorbing $D = \mathcal{O}((1 - \beta_1)^{-1})$ into $\xi$.** Since $m \geq 1$, we have

$$\frac{1}{1 - \beta_1} \leq \frac{m}{1 - \beta_1} = \xi,$$

hence $D = \mathcal{O}((1 - \beta_1)^{-1})$ can be absorbed into $\mathcal{O}(\xi)$ throughout.

**Absorbing the remaining warm-up terms when $a \in (0, 1)$.** For $a \in (0, 1)$, the warm-up drift satisfies

$$\mathcal{S}_1 = \mathcal{O}\left(\xi^{\frac{1}{a} - 1} + L^{a-1} + \tilde{t}^{1-a}\right).$$

We show that the last two terms can be absorbed into the first term (up to constants depending only on $a$), so that

$$\mathcal{S}_1 = \mathcal{O}\left(\xi^{\frac{1}{a} - 1}\right) \qquad (a \in (0, 1)).$$

1. **Absorbing $L^{a-1}$.** Using the standard inequality $-\log x \geq 1 - x$ for $x \in (0, 1)$, we have

   $$L = \log(1/\beta_1) = -\log \beta_1 \geq 1 - \beta_1.$$

   Since $a - 1 < 0$, raising both sides to power $(a - 1)$ reverses the inequality:

   $$L^{a-1} \leq (1 - \beta_1)^{a-1} = \frac{1}{(1 - \beta_1)^{1-a}}.$$

   Moreover, because $m \geq 1$ and $\frac{1-a}{a} \geq 1 - a$ (equivalently $1/a \geq 1$),

   $$\xi^{\frac{1}{a} - 1} = \left(\frac{m}{1 - \beta_1}\right)^{\frac{1-a}{a}} = m^{\frac{1-a}{a}} (1 - \beta_1)^{-\frac{1-a}{a}} \geq (1 - \beta_1)^{-(1-a)}.$$

   Combining the two displays yields

   $$L^{a-1} \leq \xi^{\frac{1}{a} - 1}.$$

2. **Absorbing $\tilde{t}^{1-a}$.** For $a \in (0, 1)$, recall $\tilde{t} = L^{-1/a} + L^{-1} \log(1/L)$ with $L = \log(1/\beta_1)$. Since $1 - 1/a < 0$, the function $y^{1-1/a} \log y$ is bounded on $[1, \infty)$, and hence there exists a constant $C_a > 0$ such that

   $$L^{-1} \log(1/L) \leq C_a L^{-1/a}, \qquad \forall L \in (0, 1].$$

   Therefore,

   $$\tilde{t} \leq C_a L^{-1/a} \quad \Rightarrow \quad \tilde{t}^{1-a} \leq C_a L^{-(1-a)/a}.$$

   Using $L \geq 1 - \beta_1$ and $m \geq 1$, we obtain

   $$\tilde{t}^{1-a} \leq C_a (1 - \beta_1)^{-(1-a)/a} \leq C_a \left(\frac{m}{1 - \beta_1}\right)^{\frac{1-a}{a}} = C_a \xi^{\frac{1}{a} - 1}.$$

Hence, for $a \in (0, 1)$,

$$\mathcal{S}_1 = \mathcal{O}\left(\xi^{\frac{1}{a} - 1}\right), \qquad \text{and thus} \qquad \mathcal{S}_2 = \mathcal{O}\left(n \xi^{\frac{1}{a} - 1}\right).$$

**Remarks for the case $a = 1$.** When $a = 1$, the warm-up drift becomes logarithmic:

$$\mathcal{S}_1 = \mathcal{O}\left(\log \xi + \log \frac{1}{L} + \log \tilde{t}\right), \qquad \tilde{t} = \frac{1}{L} \log \frac{1}{L},$$

where $L = \log(1/\beta_1)$. Noting that

$$\log \tilde{t} = \log\left(\frac{1}{L}\right) + \log\left(\log \frac{1}{L}\right),$$

and using the fact that $\log \log(1/L) = o(\log(1/L))$ as $L \to 0$, the third term is of lower order and can be absorbed into $\log(1/L)$. Hence,

$$\mathcal{S}_1 = \mathcal{O}\left(\log \xi + \log \frac{1}{L}\right) = \mathcal{O}\left(\log \frac{m}{1 - \beta_1}\right).$$

Consequently,

$$\mathcal{S}_2 = \mathcal{O}\left(n \log \frac{m}{1 - \beta_1}\right).$$

Then We analyze by cases of $a$:

- **Case $a < 1/2$:** The denominator is $t^{1-a}$. The numerator is dominated by $\mathcal{S}_2$ (scaled by $n$) and the growing noise $\xi t^{1-2a}$.

$$\text{Rate} = \mathcal{O}\left(\frac{n\left([\frac{m}{1-\beta_1}]^{\frac{1}{a}-1} + (\log(1/\beta_1))^{a-1} + \tilde{t}^{1-a}\right) + \frac{m}{1-\beta_1}t^{1-2a} + (1-\beta_1)^{-1}}{t^{1-a}}\right),$$

$$= \mathcal{O}\left(\frac{n[\frac{m}{1-\beta_1}]^{\frac{1}{a}-1} + \frac{m}{1-\beta_1}t^{1-2a}}{t^{1-a}}\right).$$

- **Case $a = 1/2$:** The denominator is $t^{1/2}$. The noise grows as $\xi \log t$.

$$\text{Rate} = \mathcal{O}\left(\frac{n\left(\frac{m}{1-\beta_1} + (\log(1/\beta_1))^{-1/2} + \tilde{t}^{1/2}\right) + \frac{m}{1-\beta_1}\log t + (1-\beta_1)^{-1}}{t^{1/2}}\right),$$

$$= \mathcal{O}\left(\frac{\frac{m}{1-\beta_1}(n + \log t)}{t^{1/2}}\right).$$

- **Case $1/2 < a < 1$:** The denominator is $t^{1-a}$. The noise sum converges to a constant $\mathcal{O}(\xi)$, which is absorbed into the constant terms (or explicitly kept as $\xi$).

$$\text{Rate} = \mathcal{O}\left(\frac{n\left([\frac{m}{1-\beta_1}]^{\frac{1}{a}-1} + (\log(1/\beta_1))^{a-1} + \tilde{t}^{1-a}\right) + \frac{m}{1-\beta_1} + (1-\beta_1)^{-1}}{t^{1-a}}\right),$$

$$= \mathcal{O}\left(\frac{n[\frac{m}{1-\beta_1}]^{\frac{1}{a}-1} + \frac{m}{1-\beta_1}}{t^{1-a}}\right).$$

- **Case $a = 1$:** The denominator is $\log t$. The powers in $\mathcal{S}_1$ become logarithms.

$$\text{Rate} = \mathcal{O}\left(\frac{n\left(\log(\frac{m}{1-\beta_1}) + \log(\frac{1}{\log(1/\beta_1)}) + \log \tilde{t}\right) + \frac{m}{1-\beta_1} + (1-\beta_1)^{-1}}{\log t}\right),$$

$$= \mathcal{O}\left(\frac{n\log(\frac{m}{1-\beta_1}) + \frac{m}{1-\beta_1}}{\log t}\right).$$

This concludes the derivation of the rates. $\qquad\square$

# F   Proof in Section 4.3 (without Momentum)

**Lemma F.1** (Descent Lemma for SVR-Stochastic $l_p$-Steepest Descent Algorithm without Momentum). *Suppose Assumptions 3.1, 3.2, and 3.3 hold. Let $t_0 = \Theta(m^{1/a})$ and define $\alpha_1 := 16R^2 m(m-1)(1+m)^a$ (where $m = n/b$) as the constant*

*derived from the variance reduction bound. Consider the learning rate schedule $\eta_t = c \cdot t^{-a}$ with $a \in (0, 1]$. Then, for all iterations $t \geq t_0$, the loss satisfies the following descent inequality:*

$$L(\mathbf{W}_{t+1}) \leq L(\mathbf{W}_t) - \eta_t \gamma \mathcal{G}(\mathbf{W}_t) + \alpha_1 \eta_t^2 \mathcal{G}(\mathbf{W}_t) + 2\eta_t^2 R^2 e^{2R\eta_0} \mathcal{G}(\mathbf{W}_t).$$

*Proof.* Similarly, let $\tilde{\boldsymbol{\Delta}}_t = \mathbf{W}_{t+1} - \mathbf{W}_t$, and define $\mathbf{W}_{t,t+1,\zeta} := \mathbf{W}_t + \zeta(\mathbf{W}_{t+1} - \mathbf{W}_t)$. We choose $\zeta^*$ such that $\mathbf{W}_{t,t+1,\zeta^*}$ satisfies (7), then we have:

$$L(\mathbf{W}_{t+1}) = L(\mathbf{W}_t) + \underbrace{\langle \nabla L(\mathbf{W}_t), \mathbf{W}_{t+1} - \mathbf{W}_t \rangle}_{A}$$

$$+ \underbrace{\frac{1}{2n} \sum_{i \in [n]} \mathbf{x}_i^\top \tilde{\boldsymbol{\Delta}}_t^\top \left( \mathrm{diag}(\mathbb{S}(\mathbf{W}_{t,t+1,\zeta^*} \mathbf{x}_i)) - \mathbb{S}(\mathbf{W}_{t,t+1,\zeta^*} \mathbf{x}_i) \mathbb{S}(\mathbf{W}_{t,t+1,\zeta^*} \mathbf{x}_i)^\top \right) \tilde{\boldsymbol{\Delta}}_t \mathbf{x}_i}_{B} . \tag{40}$$

For Term A, we just follow the same steps in Lemma D.1, E.1 and get:

$$\langle \nabla L(\mathbf{W}_t), \mathbf{W}_{t+1} - \mathbf{W}_t \rangle \leq 2\eta_t \| \nabla L(\mathbf{W}_t) - \mathbf{V}_t \|_{\mathrm{sum}} - \eta_t \gamma \mathcal{G}(\mathbf{W}).$$

Notice that $\mathbf{V}_t = \nabla L_{\mathcal{B}_t}(\mathbf{W}_t) - \nabla L_{\mathcal{B}_t}(\tilde{\mathbf{W}}_t) + \nabla L(\tilde{\mathbf{W}}_t)$ and apply Lemma C.13:

$$\| \nabla L(\mathbf{W}_t) - \mathbf{V}_t \|_{\mathrm{sum}} = \left\| \nabla L(\mathbf{W}_t) - \nabla L_{\mathcal{B}_t}(\mathbf{W}_t) - \nabla L_{\mathcal{B}_t}(\tilde{\mathbf{W}}_t) + \nabla L(\tilde{\mathbf{W}}_t) \right\|_{\mathrm{sum}}$$

$$\leq 2(m-1)R(e^{2R\|\mathbf{W}_t - \tilde{\mathbf{W}}_t\|_{\mathrm{max}}} - 1)\mathcal{G}(\mathbf{W}_t)$$

Recall that $\tilde{\mathbf{W}}_t$ is the snapshot weight at the beginning of the current epoch. Since there are at most $m$ steps between $\mathbf{W}_t$ and $\tilde{\mathbf{W}}_t$, we can bound the max-norm distance as:

$$\left\| \mathbf{W}_t - \tilde{\mathbf{W}}_t \right\|_{\mathrm{max}} \leq \sum_{j=\mathrm{start\_of\_epoch}}^{t-1} \eta_j \leq m\eta_{t-m},$$

where we used the monotonicity of the learning rate ($\eta_j \leq \eta_{t-m}$).

Now, applying the inequality $e^x - 1 \leq xe^x$ (valid for $x \geq 0$) with $x = 2R\left\| \mathbf{W}_t - \tilde{\mathbf{W}}_t \right\|_{\mathrm{max}}$, we obtain:

$$e^{2R\|\mathbf{W}_t - \tilde{\mathbf{W}}_t\|_{\mathrm{max}}} - 1 \leq 2R\left\| \mathbf{W}_t - \tilde{\mathbf{W}}_t \right\|_{\mathrm{max}} e^{2R\|\mathbf{W}_t - \tilde{\mathbf{W}}_t\|_{\mathrm{max}}}$$

$$\leq 2Rm\eta_{t-m} e^{2Rm\eta_{t-m}}.$$

Substituting this back into the bound for $\| \nabla L(\mathbf{W}_t) - \mathbf{V}_t \|_{\mathrm{sum}}$:

$$\| \nabla L(\mathbf{W}_t) - \mathbf{V}_t \|_{\mathrm{sum}} \leq 2(m-1)R \left[ 2Rm\eta_{t-m} e^{2Rm\eta_{t-m}} \right] \mathcal{G}(\mathbf{W}_t)$$

$$= 4R^2 m(m-1) e^{2Rm\eta_{t-m}} \eta_{t-m} \mathcal{G}(\mathbf{W}_t).$$

Next, we handle the learning rate terms under the schedule $\eta_t = c \cdot t^{-a}$. For the exponential term, Let $t_0$ be any time such that $t_0 = m + \left\lceil \left( \frac{2Rmc}{\ln 2} \right)^{1/a} \right\rceil$. In particular, for all $t \geq t_0$, we have $2Rm\eta_{t-m} \leq \ln 2$ and hence $e^{2Rm\eta_{t-m}} \leq 2$.

For the linear term $\eta_{t-m}$, for any $t > m$, we have the ratio:

$$\frac{\eta_{t-m}}{\eta_t} = \left( \frac{t}{t-m} \right)^a = \left( 1 + \frac{m}{t-m} \right)^a \leq (1+m)^a,$$

where the inequality holds because $t - m \geq 1$. Thus, $\eta_{t-m} \leq (1+m)^a \eta_t$.

Substituting these bounds, we get:

$$\|\nabla L(\mathbf{W}_t) - \mathbf{V}_t\|_{\text{sum}} \leq 4R^2 m(m-1) \cdot 2 \cdot (1+m)^a \eta_t \mathcal{G}(\mathbf{W}_t)$$
$$= 8R^2 m(m-1)(1+m)^a \eta_t \mathcal{G}(\mathbf{W}_t).$$

Finally, substituting this back into the expression for Term A:

$$\langle \nabla L(\mathbf{W}_t), \mathbf{W}_{t+1} - \mathbf{W}_t \rangle \leq 2\eta_t \left[ 8R^2 m(m-1)(1+m)^a \eta_t \mathcal{G}(\mathbf{W}_t) \right] - \eta_t \gamma \mathcal{G}(\mathbf{W}_t)$$
$$= -\eta_t \gamma \mathcal{G}(\mathbf{W}_t) + \alpha_1 \eta_t^2 \mathcal{G}(\mathbf{W}_t),$$

where we define the constant $\alpha_1 := 16R^2 m(m-1)(1+m)^a$. Finally, for Term B, We just follow the same steps in Lemma D.1 and get:

$$TermB \leq 2\eta_t^2 R^2 e^{2R\eta_0} \mathcal{G}(\mathbf{W}_t).$$

Combining Terms A, B together, we obtain

$$L(\mathbf{W}_{t+1}) \leq L(\mathbf{W}_t) - \eta_t \gamma \mathcal{G}(\mathbf{W}_t) + \alpha_1 \eta_t^2 \mathcal{G}(\mathbf{W}_t) + 2\eta_t^2 R^2 e^{2R\eta_0} \mathcal{G}(\mathbf{W}_t). \tag{41}$$

$\square$

From Eq. 41, we observe a crucial advantage of the variance reduction mechanism: the noise terms scale with $\eta_t^2$, Consequently, unlike the standard stochastic setting (Random Reshuffling) which relies on a **Large Batch** condition to ensure a positive effective margin, SVR requires no constraints on the batch size. As long as $t$ is sufficiently large, the loss is guaranteed to decrease monotonically.

**Lemma F.2** (Loss convergence). *Suppose Assumptions 3.1, 3.2, and 3.2 hold, let $\tilde{L} := \frac{\log 2}{n}$. Then, there exists a time index $t_2$ such that for all $t > t_2$, $L(\mathbf{W}_t) \leq \tilde{L}$. Specifically, the condition for $t_2$ is determined by the learning rate accumulation:*

$$\sum_{s=t_1}^{t_2} \eta_s \geq \frac{4L(\mathbf{W}_0) + 8R \sum_{s=0}^{t_1-1} \eta_s}{\gamma \tilde{L}}, \tag{42}$$

*where $t_1$ is the time step ensuring monotonic descent.*

*Proof.* **Determination of $t_1$ (Start of Monotonicity).** Recall the descent inequality derived in Lemma F.1:

$$L(\mathbf{W}_{t+1}) \leq L(\mathbf{W}_t) - \eta_t \left( \gamma - \eta_t \alpha_{\text{svr}} \right) \mathcal{G}(\mathbf{W}_t),$$

where $\alpha_{\text{svr}} := \alpha_1 + 2R^2 e^{2R\eta_0}$ encapsulates all second-order noise terms. To ensure strict descent, we require the first-order margin term to dominate the second-order variance and curvature terms. Specifically, we choose $t_1$ such that for all $t \geq t_1$:

$$\eta_t \alpha_{\text{svr}} \leq \frac{\gamma}{2} \iff \eta_t \leq \frac{\gamma}{2\alpha_{\text{svr}}}.$$

Since the learning rate $\eta_t = c \cdot t^{-a}$ is monotonically decreasing to zero, such a finite time $t_1$ always exists, regardless of the magnitude of $\alpha_{\text{svr}}$ (and thus independent of the batch size $m$). For all $t \geq t_1$, the loss satisfies:

$$L(\mathbf{W}_{t+1}) \leq L(\mathbf{W}_t) - \frac{\gamma}{2} \eta_t \mathcal{G}(\mathbf{W}_t). \tag{43}$$

**Determination of $t_2$.** The subsequent analysis for $t_2$ (the time to enter the separable region $L(\mathbf{W}_t) \leq \tilde{L}$) follows the exact same logic as in the proof of Lemma D.2, by replacing the effective margin $\rho$ with the full margin $\gamma$. By summing Eq. (43) and bounding the initial loss growth, we obtain the stated condition for $t_2$. $\square$

**Lemma F.3** (Unnormalized Margin Growth for SVR). *Consider the same setting as Lemma F.2. Let $t_2$ be the time index guaranteed by Lemma F.2 such that $L(\mathbf{W}_t) \leq \frac{\log 2}{n}$ for all $t > t_2$. Recall the total noise constant defined in Lemma F.1: $\alpha_{svr} := \alpha_1 + 2R^2 e^{2R\eta_0}$. Then, for all $t > t_2$, the minimum unnormalized margin satisfies:*

$$\min_{i \in [n], c \neq y_i} (\mathbf{e}_{y_i} - \mathbf{e}_c)^\top \mathbf{W}_t \mathbf{x}_i \geq \gamma \sum_{s=t_2}^{t-1} \eta_s \frac{\mathcal{G}(\mathbf{W}_s)}{L(\mathbf{W}_s)} - \alpha_{svr} \sum_{s=t_2}^{t-1} \eta_s^2. \tag{44}$$

*Proof.* The proof follows the exact same logic as Lemma D.3. We start from the SVR descent inequality derived in Lemma F.1:

$$L(\mathbf{W}_{s+1}) \leq L(\mathbf{W}_s) - \eta_s(\gamma - \eta_s\alpha_{\text{svr}})\mathcal{G}(\mathbf{W}_s).$$

By replacing the effective margin $\rho$ with the full margin $\gamma$, and the curvature constant $\alpha_1$ with the total SVR noise constant $\alpha_{\text{svr}}$, the recursive derivation for the lower bound of the unnormalized margin remains valid. $\square$

**Theorem F.4** (Margin Convergence Rate of SVR-Stochastic $l_p$ Steepest Descent). *Suppose Assumptions 3.1, 3.2, and 3.3 hold. Consider the learning rate schedule $\eta_t = c \cdot t^{-a}$ with $a \in (0,1]$. Recall the total noise constant $\alpha_{svr} := \alpha_1 + 2R^2 e^{2R\eta_0}$, which scales with the batch size parameter as $\alpha_{svr} = \Theta(m^{2+a})$. Let $t_2$ be the time index guaranteed by Lemma F.2 such that $L(\mathbf{W}_t) \leq \frac{\log 2}{n}$ for all $t > t_2$.*

*Then, for all $t > t_2$, the SVR algorithm recovers the full max-margin solution, and the margin gap satisfies:*

$$\gamma - \frac{\min_{i\in[n],c\neq y_i}(\mathbf{e}_{y_i} - \mathbf{e}_c)^\top \mathbf{W}_t\mathbf{x}_i}{\|\mathbf{W}_t\|} \leq \mathcal{O}\left(\frac{\sum_{s=t_2}^{t-1}\eta_s e^{-\frac{\gamma}{4}\sum_{\tau=t_2}^{s-1}\eta_\tau} + \sum_{s=0}^{t_2-1}\eta_s + m^{2+a}\sum_{s=t_2}^{t-1}\eta_s^2}{\sum_{s=0}^{t-1}\eta_s}\right).$$

*Remark: Unlike standard Random Reshuffling, the convergence to the full margin $\gamma$ is guaranteed regardless of the batch size. However, the magnitude of the noise term $\alpha_{svr}$ (scaling with $m^{2+a}$) is significantly larger, potentially affecting the pre-asymptotic convergence speed.*

*Proof.* The proof parallels the derivation of Theorem D.4. First, combining Lemma F.2 with the SVR descent property (Lemma F.1) ensures that for $t > t_2$, the ratio $\frac{\mathcal{G}(\mathbf{W}_t)}{L(\mathbf{W}_t)}$ converges to 1 exponentially with a rate determined by the full margin $\gamma$. Second, substituting the unnormalized margin lower bound from Lemma F.3 and the standard weight upper bound ($\|\mathbf{W}_t\| \leq \|\mathbf{W}_0\| + \sum \eta_s$) into the definition of the normalized margin gap yields the stated bound. Mathematically, this is identical to the result in Theorem D.4, obtained by substituting the effective margin $\rho$ with the full margin $\gamma$ and the noise coefficient with $\alpha_{\text{svr}}$. $\square$

**Corollary F.5.** *Consider the learning rate schedule $\eta_t = c \cdot t^{-a}$ with $a \in (0,1]$. Under the setting of Theorem F.4, the margin gap converges with the following rates:*

$$\gamma - \frac{\min_{i\in[n],c\neq y_i}(\mathbf{e}_{y_i} - \mathbf{e}_c)^\top \mathbf{W}_t\mathbf{x}_i}{\|\mathbf{W}_t\|} = \begin{cases} \mathcal{O}\left(\frac{nm^{\frac{2-a-a^2}{a}}+m^{2+a}t^{1-2a}}{t^{1-a}}\right) & \text{if} \quad a < \frac{1}{2} \\ \mathcal{O}\left(\frac{nm^{5/2}+m^{5/2}\log t}{t^{1/2}}\right) & \text{if} \quad a = \frac{1}{2} \\ \mathcal{O}\left(\frac{nm^{\frac{2-a-a^2}{a}}+m^{2+a}}{t^{1-a}}\right) & \text{if} \quad \frac{1}{2} < a < 1 \\ \mathcal{O}\left(\frac{n\log m^3+m^3}{\log t}\right) & \text{if} \quad a = 1 \end{cases}$$

*Proof.* The proof is identical to that of Corollary D.5, except that $\alpha_{\text{svr}} = \Theta(m^{2+a})$. $\square$

## G   Proof in Section 4.3 (with Momentum)

**Lemma G.1** (Descent Lemma for SVR-Stochastic $l_p$-Steepest Descent Algorithm with Momentum). *Suppose that Assumption 3.1, 3.2, 3.3 and 3.4 hold. Consider the learning rate schedule $\eta_t = c \cdot t^{-a}$ with $a \in (0,1]$. Define the SVR-Momentum noise constant $\alpha_{svr\text{-}m}$ and initialization constant $\alpha_2$ as:*

$$\alpha_{svr\text{-}m} := 4(m-1)(1-\beta_1)Rc_2' + 32(m^2-m)(1+2m)^a R^2(1-\beta_1^m)$$
$$+ 4R(1-\beta_1)c_2 + 2R^2 e^{2R\eta_0},$$
$$\alpha_2 := 4R.$$

*Then, there exist $t_0$ and for all iterations $t > t_0$, the loss satisfies the following descent inequality:*

$$L(\mathbf{W}_{t+1}) \leq L(\mathbf{W}_t) - \eta_t\gamma\mathcal{G}(\mathbf{W}_t) + \alpha_{svr\text{-}m}\eta_t^2\mathcal{G}(\mathbf{W}_t) + \alpha_2\eta_t\beta_1^{\frac{t}{2}}\mathcal{G}(\mathbf{W}_t).$$

*Proof.* Similarly, let $\tilde{\boldsymbol{\Delta}}_t = \mathbf{W}_{t+1} - \mathbf{W}_t$, and define $\mathbf{W}_{t,t+1,\zeta} := \mathbf{W}_t + \zeta(\mathbf{W}_{t+1} - \mathbf{W}_t)$. We choose $\zeta^*$ such that $\mathbf{W}_{t,t+1,\zeta^*}$ satisfies (7), then we have:

$$
\begin{aligned}
L(\mathbf{W}_{t+1}) = L(\mathbf{W}_t) + \underbrace{\langle \nabla L(\mathbf{W}_t), \mathbf{W}_{t+1} - \mathbf{W}_t \rangle}_{A} \\
+ \underbrace{\frac{1}{2n} \sum_{i \in [n]} \mathbf{x}_i^\top \tilde{\boldsymbol{\Delta}}_t^\top \left( \mathrm{diag}(\mathbb{S}(\mathbf{W}_{t,t+1,\zeta^*} \mathbf{x}_i)) - \mathbb{S}(\mathbf{W}_{t,t+1,\zeta^*} \mathbf{x}_i) \mathbb{S}(\mathbf{W}_{t,t+1,\zeta^*} \mathbf{x}_i)^\top \right) \tilde{\boldsymbol{\Delta}}_t \, \mathbf{x}_i}_{B} .
\end{aligned}
\tag{45}
$$

For Term A, we just follow the same steps in Lemma D.1, E.1 and get:

$$
\langle \nabla L(\mathbf{W}_t), \mathbf{W}_{t+1} - \mathbf{W}_t \rangle \leq 2\eta_t \| \nabla L(\mathbf{W}_t) - \mathbf{M}_t^V \|_{\mathrm{sum}} - \eta_t \gamma \mathcal{G}(\mathbf{W}).
$$

Notice that $\mathbf{V}_t = \nabla L_{\mathcal{B}_t}(\mathbf{W}_t) - \nabla L_{\mathcal{B}_t}(\tilde{\mathbf{W}}_t) + \nabla L(\tilde{\mathbf{W}}_t)$ and $\mathbf{M}_t^V = \sum_{\tau=0}^t (1-\beta_1)\beta_1^\tau \mathbf{V}_{t-\tau}$.

$$
\begin{aligned}
\| \mathbf{M}_t^V - \nabla L(\mathbf{W}_t) \|_{\mathrm{sum}} &= \left\| \sum_{\tau=0}^t (1-\beta_1)\beta_1^\tau \mathbf{V}_{t-\tau} - \sum_{\tau=0}^t (1-\beta_1)\beta_1^\tau \nabla L(\mathbf{W}_t) + \beta_1^{t+1} \nabla L(\mathbf{W}_t) \right\|_{\mathrm{sum}} \\
&\leq \underbrace{\left\| \sum_{\tau=0}^t (1-\beta_1)\beta_1^\tau \mathbf{V}_{t-\tau} - \sum_{\tau=0}^t (1-\beta_1)\beta_1^\tau \nabla L(\mathbf{W}_{t-\tau}) \right\|_{\mathrm{sum}}}_{A_{1,1}} \\
&\quad + \underbrace{\left\| \sum_{\tau=0}^t (1-\beta_1)\beta_1^\tau \nabla L(\mathbf{W}_{t-\tau}) - \sum_{\tau=0}^t (1-\beta_1)\beta_1^\tau \nabla L(\mathbf{W}_t) \right\|_{\mathrm{sum}}}_{A_{1,2}} + \underbrace{\left\| \beta_1^{t+1} \nabla L(\mathbf{W}_t) \right\|_{\mathrm{sum}}}_{A_{1,3}}
\end{aligned}
$$

For Term $A_{1,1}$, applying Lemma C.14, we have:

$$
A_{1,1} \leq 2(m-1)(1-\beta_1)Rc_2'\eta_t \mathcal{G}(\mathbf{W}_t) + 16(m^2-m)(1+2m)^a R^2(1-\beta_1^m)\eta_t \mathcal{G}(\mathbf{W}_t),
$$

We apply Lemma C.10 to the gradient difference term. Using the exponential bound and the fact that $\|\mathbf{W}_{t-\tau} - \mathbf{W}_t\|_{\max} \leq \sum_{j=1}^\tau \eta_{t-j}$:

$$
\| \nabla L(\mathbf{W}_{t-\tau}) - \nabla L(\mathbf{W}_t) \|_{\mathrm{sum}} \leq 2R \left( e^{2R \sum_{j=1}^\tau \eta_{t-j}} - 1 \right) \mathcal{G}(\mathbf{W}_t).
$$

Substituting this into the sum and applying Assumption 3.4, for $t > t_0$:

$$
\begin{aligned}
A_{1,2} &\leq 2R(1-\beta_1)\mathcal{G}(\mathbf{W}_t) \sum_{\tau=0}^t \beta_1^\tau \left( e^{2R \sum_{j=1}^\tau \eta_{t-j}} - 1 \right) \\
&\leq 2R(1-\beta_1)c_2 \eta_t \mathcal{G}(\mathbf{W}_t).
\end{aligned}
$$

*Remark: $c_2'$ in $A_{1,1}$ and $c_2$ in $A_{1,2}$ are different. While $c_2'$ is a function of $(R, m, a, \beta_1)$, $c_2$ is a function of $(R, \beta_1)$.*

Using Lemma C.5 and notice that $\beta_1^{t+1} \leq \beta_1^{\frac{t}{2}}$, we have:

$$
A_{1,3} \leq 2R\beta_1^{t+1} \mathcal{G}(\mathbf{W}_t) \leq 2R\beta_1^{\frac{t}{2}} \mathcal{G}(\mathbf{W}_t).
$$

Summing the bounds for $A_{1,1}$, $A_{1,2}$, and $A_{1,3}$, and multiplying by the outer factor $2\eta_t$ from the Term A inequality:

$$
\begin{aligned}
A &\leq 4(m-1)(1-\beta_1)Rc_2'\eta_t^2 \mathcal{G}(\mathbf{W}_t) + 32(m^2-m)(1+2m)^a R^2(1-\beta_1^m)\eta_t^2 \mathcal{G}(\mathbf{W}_t) \\
&\quad + 4R(1-\beta_1)c_2 \eta_t^2 \mathcal{G}(\mathbf{W}_t) + 4R\beta_1^{\frac{t}{2}} \eta_t \mathcal{G}(\mathbf{W}_t) - \gamma\eta_t \mathcal{G}(\mathbf{W}_t)
\end{aligned}
$$

Finally, for Term B, We just follow the same steps in Lemma D.1 and get:

$$TermB \leq 2\eta_t^2 R^2 e^{2R\eta_0} \mathcal{G}(\mathbf{W}_t).$$

Combining Terms A, B together and using the notation $\alpha_{\text{svr-m}} := 4(m-1)(1-\beta_1)Rc_2' + 32(m^2-m)(1+2m)^a R^2(1-\beta_1^m) + 4R(1-\beta_1)c_2 + 2R^2 e^{2R\eta_0}$, $\alpha_2 := 4R$ we obtain

$$L(\mathbf{W}_{t+1}) \leq L(\mathbf{W}_t) - \eta_t \gamma \mathcal{G}(\mathbf{W}_t) + \alpha_{\text{svr-m}}\eta_t^2 \mathcal{G}(\mathbf{W}_t) + \alpha_2 \eta_t \beta_1^{\frac{t}{2}} \mathcal{G}(\mathbf{W}_t). \tag{46}$$

Note that by Lemma C.15, $c_2'$ scales with $\frac{m^{1+a}}{(1-\beta_1)^2}$ so that $\alpha_{\text{svr-m}} = \Theta(\frac{m^{2+a}}{1-\beta_1})$. $\qquad\square$

From Eq. 46, we observe that similat to SVR algorithm without momentum, SVR with momentum requires no constraints on the batch size. Also it requires no constraints on $\beta_1$. As long as $t$ is sufficiently large, the loss is guaranteed to decrease monotonically.

**Lemma G.2** (Loss Convergence for SVR-Stochastic $l_p$-Steepest Descent Algorithm with Momentum). *Under the same setting as Lemma G.1, let $\tilde{L} := \frac{\log 2}{n}$. There exists a finite time index $t_1$ such that for all $t \geq t_1$, the loss is monotonically decreasing. Furthermore, there exists $t_2 \geq t_1$ such that for all $t > t_2$, $L(\mathbf{W}_t) \leq \tilde{L}$. The condition for $t_2$ is determined by:*

$$\sum_{s=t_1}^{t_2} \eta_s \geq \frac{2L(\mathbf{W}_0) + 4R\sum_{s=0}^{t_1-1} \eta_s}{\gamma \tilde{L}}. \tag{47}$$

*Proof.* **Determination of $t_1$:** The descent inequality Eq. (46) guarantees strict descent when the effective margin term dominates the noise:

$$\alpha_{\text{svr-m}}\eta_t + \alpha_2 \beta_1^{t/2} \leq \frac{\gamma}{2}.$$

Since $\eta_t \to 0$ and $\beta_1^{t/2} \to 0$ as $t \to \infty$, such a $t_1$ always exists regardless of the batch size $m$ or momentum $\beta_1$ (provided $\beta_1 < 1$).

**Determination of $t_2$:** For $t \geq t_1$, the descent inequality simplifies to $L(\mathbf{W}_{t+1}) \leq L(\mathbf{W}_t) - \frac{\gamma}{2}\eta_t \mathcal{G}(\mathbf{W}_t)$. The rest of the proof follows exactly as in Lemma D.2 and Lemma F.2, using the full margin $\gamma$. $\qquad\square$

**Lemma G.3** (Unnormalized Margin Growth for SVR-Stochastic $l_p$-Steepest Descent Algorithm with Momentum). *Consider the same setting as Lemma G.2. Let $t_2$ be the time index such that $L(\mathbf{W}_t) \leq \tilde{L}$ for all $t > t_2$. Define the constant upper bound for the momentum initialization drift as $Q := \frac{\alpha_2 \eta_0}{1-\sqrt{\beta_1}}$. Then, for all $t > t_2$, the minimum unnormalized margin satisfies:*

$$\min_{i\in[n], c\neq y_i} (\mathbf{e}_{y_i} - \mathbf{e}_c)^\top \mathbf{W}_t \mathbf{x}_i \geq \gamma \sum_{s=t_2}^{t-1} \eta_s \frac{\mathcal{G}(\mathbf{W}_s)}{L(\mathbf{W}_s)} - \alpha_{svr\text{-}m} \sum_{s=t_2}^{t-1} \eta_s^2 - D. \tag{48}$$

*Proof.* The proof mirrors Lemma E.3. We start from the descent inequality (Eq. (46)). By factoring out $L(\mathbf{W}_s)$ and using $1 - x \leq e^{-x}$, we obtain the recursive bound on the loss involving both $\eta_s^2$ and $\eta_s \beta_1^{s/2}$ noise terms. Converting this to the margin lower bound yields the stated result, where the geometric series $\sum \eta_s \beta_1^{s/2}$ is bounded by the constant $D$. $\qquad\square$

**Theorem G.4** (Margin Convergence Rate of SVR-Stochastic $l_p$-Steepest Descent Algorithm with Momentum). *Suppose that Assumption 3.1, 3.2, 3.3 and 3.4 hold. Consider the learning rate schedule $\eta_t = c \cdot t^{-a}$ with $a \in (0, 1]$. Define $\alpha_{svr\text{-}m} = 4(m-1)(1-\beta_1)Rc_2' + 32(m^2-m)(1+2m)^a R^2(1-\beta_1^m) + 4R(1-\beta_1)c_2 + 2R^2 e^{2R\eta_0}$, which scales with $\Theta\left(\frac{m^{2+a}}{1-\beta_1}\right)$, and $D = \frac{4R\eta_0}{1-\sqrt{\beta_1}}$. Let $t_2$ be the time index guaranteed by Lemma G.2.*

*Then, for all $t > t_2$, the algorithm converges to the full max-margin solution $\gamma$, with the margin gap satisfying:*

$$\gamma - \frac{\min_{i\in[n], c\neq y_i}(\mathbf{e}_{y_i}-\mathbf{e}_c)^\top \mathbf{W}_t \mathbf{x}_i}{\|\mathbf{W}_t\|} \leq \mathcal{O}\left(\frac{\sum_{s=t_2}^{t-1}\eta_s e^{-\frac{\gamma}{4}\sum_{\tau=t_2}^{s-1}\eta_\tau} + \sum_{s=0}^{t_2-1}\eta_s + \frac{m^{2+a}}{1-\beta_1}\sum_{s=t_2}^{t-1}\eta_s^2 + D}{\sum_{s=0}^{t-1}\eta_s}\right).$$

*Remark: SVR-Stochastic $l_p$-steepest descent algorithm with momentum requires neither large-batch nor high momentum condition (unlike the algorithm without SVR).*

*Proof.* The proof is structurally identical to Theorem E.4 and Theorem F.4. The ratio $\frac{\mathcal{G}}{\mathcal{L}}$ converges to 1 exponentially with rate $\gamma/4$. The unnormalized margin grows according to Lemma G.3, driven by the full margin $\gamma$. Substituting these into the normalized margin gap definition yields the upper bound, where the noise terms $\alpha_{\text{svr-m}} \sum \eta^2$ and $D$ appear in the numerator. $\qquad \square$

**Corollary G.5.** *Consider the learning rate schedule $\eta_t = c \cdot t^{-a}$ with $a \in (0,1]$. Under the setting of Theorem G.4, the margin gap converges with the following rates:*

$$\gamma - \frac{\min_{i \in [n], c \neq y_i} (\mathbf{e}_{y_i} - \mathbf{e}_c)^\top \mathbf{W}_t \mathbf{x}_i}{\|\mathbf{W}_t\|} = \begin{cases} \mathcal{O}\left( \frac{n(\frac{m^{2+a}}{1-\beta_1})^{\frac{1}{a}-1} + \frac{m^{2+a}}{1-\beta_1} t^{1-2a}}{t^{1-a}} \right) & \text{if } a < \frac{1}{2} \\[2ex] \mathcal{O}\left( \frac{\frac{m^{5/2}}{1-\beta_1}(n + \log t)}{t^{1/2}} \right) & \text{if } a = \frac{1}{2} \\[2ex] \mathcal{O}\left( \frac{n(\frac{m^{2+a}}{1-\beta_1})^{\frac{1}{a}-1} + \frac{m^{2+a}}{1-\beta_1}}{t^{1-a}} \right) & \text{if } \frac{1}{2} < a < 1 \\[2ex] \mathcal{O}\left( \frac{n \log \frac{m^3}{1-\beta_1} + \frac{m^3}{1-\beta_1}}{\log t} \right) & \text{if } a = 1 \end{cases}$$

*If we view $\beta_1$ as a constant for better comparison, we have:*

$$\gamma - \frac{\min_{i \in [n], c \neq y_i} (\mathbf{e}_{y_i} - \mathbf{e}_c)^\top \mathbf{W}_t \mathbf{x}_i}{\|\mathbf{W}_t\|} = \begin{cases} \mathcal{O}\left( \frac{nm^{\frac{2-a-a^2}{a}} + m^{2+a} t^{1-2a}}{t^{1-a}} \right) & \text{if } a < \frac{1}{2} \\[2ex] \mathcal{O}\left( \frac{nm^{5/2} + m^{5/2} \log t}{t^{1/2}} \right) & \text{if } a = \frac{1}{2} \\[2ex] \mathcal{O}\left( \frac{nm^{\frac{2-a-a^2}{a}} + m^{2+a}}{t^{1-a}} \right) & \text{if } \frac{1}{2} < a < 1 \\[2ex] \mathcal{O}\left( \frac{n \log m^3 + m^3}{\log t} \right) & \text{if } a = 1 \end{cases}$$

*Proof.* The proof is identical to that of Corollary E.5, except that $\alpha_{\text{svr-m}} = \Theta(\frac{m^{2+a}}{1-\beta_1})$ and start time $\tilde{t}$ changed according to Lemma C.14:

$$\tilde{t} = \begin{cases} \left( \frac{m^{1+a}}{\log(1/\beta_1)} \right)^{\frac{1}{a}} + \frac{1}{\log(1/\beta_1)} \log \left( \frac{1}{\log(1/\beta_1)} \right), & a \in (0,1), \\[2ex] \frac{m^{1+a}}{\log(1/\beta_1)} \log \left( \frac{m^{1+a}}{\log(1/\beta_1)} \right), & a = 1. \end{cases}$$

$\qquad \square$

# H  Implicit Bias of Stochastic $l_p$ Steepest Descent in Small Batch Regime

In this section, we consider an extreme scenario where the batch size is 1. We will show that in this regime, the implicit bias of stochastic $l_p$ steepest descent differs fundamentally from its full-batch counterpart.

## H.1  Spectral-SGD and Normalized-SGD are Equivalent when Batchsize=1

For the spectral descent direction, the optimizer is not unique in general. Following Muon's definition, for a gradient matrix $\mathbf{G}_t = \mathbf{U}_t \mathbf{\Sigma}_t \mathbf{V}_t^\top$, we take the canonical spectral descent direction to be $-\mathbf{U}_t \mathbf{V}_t^\top$.

**Lemma H.1** (Equivalence of Spectral and Frobenius Descent for Single-Sample)**.** *Consider the update using a single sample $(\mathbf{x}_i, y_i)$. The steepest descent directions defined by the Spectral norm ($\|\cdot\|_{S_\infty}$) and the Frobenius norm ($\|\cdot\|_{S_2}$) are identical:*

$$\mathbf{\Delta}^{\text{Spec}} = \mathbf{\Delta}^{\text{Frob}} \in \text{argmax}_{\|\mathbf{\Delta}\|_{S_\infty} \leq 1} \langle -\nabla \ell_i(\mathbf{W}), \mathbf{\Delta} \rangle \cap \text{argmax}_{\|\mathbf{\Delta}\|_{S_2} \leq 1} \langle -\nabla \ell_i(\mathbf{W}), \mathbf{\Delta} \rangle.$$

*Proof.* Recall that the negative gradient for sample $i$ is the outer product $-\nabla \ell_i(\mathbf{W}) = (\mathbf{e}_{y_i} - \mathbf{s}_i)\mathbf{x}_i^\top$. Being an outer product of two vectors, this matrix has rank 1. Its singular value decomposition is simply $\sigma_1 \mathbf{u}\mathbf{v}^\top$, where $\sigma_1 = \|\mathbf{e}_{y_i} - \mathbf{s}_i\|_2 \|\mathbf{x}_i\|_2$, $\mathbf{u} = \frac{\mathbf{e}_{y_i} - \mathbf{s}_i}{\|\mathbf{e}_{y_i} - \mathbf{s}_i\|_2}$, and $\mathbf{v} = \frac{\mathbf{x}_i}{\|\mathbf{x}_i\|_2}$.

For the Frobenius norm (Schatten 2-norm), the optimal direction is the normalized gradient:

$$\boldsymbol{\Delta}^{\text{Frob}} = \frac{-\nabla\ell_i(\mathbf{W})}{\|\nabla\ell_i(\mathbf{W})\|_{S_2}} = \frac{(\mathbf{e}_{y_i} - \mathbf{s}_i)\mathbf{x}_i^\top}{\sigma_1} = \mathbf{u}\mathbf{v}^\top.$$

For the Spectral norm (Schatten $\infty$-norm), the inner product is maximized when $\boldsymbol{\Delta}$ aligns with the top left and right singular vectors:

$$\boldsymbol{\Delta}^{\text{spec}} = \mathbf{u}\mathbf{v}^\top.$$

Since both optimization problems yield the same rank-1 matrix $\mathbf{u}\mathbf{v}^\top$, the update directions are identical. $\qquad\square$

## H.2 Two Special Cases for SignSGD and Normalized-SGD (Batch size=1)

### H.2.1 DATA CONSTRUCTION

We consider a dataset $\mathcal{D} = \{(\mathbf{x}_i, y_i)\}_{i=1}^n$ with $K$ classes constructed under an *orthogonal scale-skewed* setting. Specifically, for the $i$-th sample with label $y_i$, the input feature is aligned with the canonical basis vector of its class, given by $\mathbf{x}_i = \alpha_i \mathbf{e}_{y_i}$, where $\mathbf{e}_{y_i} \in \mathbb{R}^K$ denotes the standard basis vector and $\alpha_i > 0$ represents the arbitrary, heterogeneous scale of the sample.

### H.2.2 IMPLICIT BIAS OF SIGNSGD (BATCH SIZE = 1) ON ORTHOGONAL SCALE-SKEWED DATASET

In this section, we analyze the implicit bias of SignSGD with a batch size of $b = 1$ (per-sample update) under the Random Reshuffling scheme on the Orthogonal Scale-Skewed dataset.

**Definitions.** For a sample $i$ with label $y_i$, let us define a *voting vector* $\mathbf{v}_i \in \mathbb{R}^k$ corresponding to the target alignment. Specifically:

$$\mathbf{v}_i[c] = \begin{cases} 1 & \text{if } c = y_i, \\ -1 & \text{if } c \neq y_i. \end{cases}$$

We define the *Stochastic Sign Bias Matrix* as the average of the individual sign-gradient directions over the dataset:

$$\bar{\mathbf{W}}_{\text{sign}} := \sum_{i=1}^n \mathbf{v}_i \, \text{sign}(\mathbf{x}_i)^\top.$$

**Theorem H.2** (Global Convergence and Implicit Bias of SignSGD). *Consider the SignSGD algorithm with batch size $b = 1$ initialized at $\mathbf{W}_0 = \mathbf{0}$ and trained on the Orthogonal Scale-Skewed dataset $\mathcal{D}$. The step size satisfies $\eta_t = c \cdot t^{-a}$ for $t \geq 1$ with constants $c > 0$ and $a \in (0, 1]$. Then:*

1. ***Loss Convergence:*** *The training loss converges to zero asymptotically:*

$$\lim_{t\to\infty} \mathcal{L}(\mathbf{W}_t) = 0.$$

2. ***Implicit Bias:*** *The parameter direction converges to the Stochastic Sign Bias Matrix:*

$$\lim_{t\to\infty} \frac{\mathbf{W}_t}{\|\mathbf{W}_t\|_F} = \frac{\bar{\mathbf{W}}_{sign}}{\|\bar{\mathbf{W}}_{sign}\|_F}.$$

*Proof.* First, we characterize the update direction for a single sample $(\mathbf{x}_i, y_i)$. The gradient of the Cross-Entropy loss is $\nabla\ell_i(\mathbf{W}_t) = (\mathbb{S}(\mathbf{W}_t\mathbf{x}_i) - \mathbf{e}_{y_i})\mathbf{x}_i^\top$. The update direction depends on $-\text{sign}(\nabla\ell_i(\mathbf{W}_t))$. Using the property $\text{sign}(\mathbf{u}\mathbf{v}^\top) = \text{sign}(\mathbf{u})\,\text{sign}(\mathbf{v})^\top$, and observing that the softmax probability is always less than 1, we have:

$$-\text{sign}(\nabla\ell_i(\mathbf{W}_t)) = \text{sign}(\mathbf{e}_{y_i} - \mathbb{S}(\mathbf{W}_t\mathbf{x}_i))\,\text{sign}(\mathbf{x}_i)^\top = \mathbf{v}_i\,\text{sign}(\mathbf{x}_i)^\top =: \mathbf{M}_i^{\text{sign}}.$$

Crucially, this matrix $\mathbf{M}_i^{\text{sign}}$ depends only on the data sample $i$ and is independent of the weights $\mathbf{W}_t$.

**Loss Convergence.** Since the update is always $\eta_\tau \mathbf{M}_{n_\tau}^{\text{sign}}$. The weights at time $t$ are:

$$\mathbf{W}_t = \sum_{\tau=0}^{t-1} \eta_\tau \mathbf{M}_{n_\tau}^{\text{sign}}.$$

$\mathbf{M}_n^{\text{sign}}$ is non-zero only at column $y_n$. For the column $y$ of $\mathbf{W}_t$, let $\mathcal{T}_y(t)$ be the set of time steps where class $y$ was sampled. The diagonal entry accumulates $+\eta$, while off-diagonal entries accumulate $-\eta$. Let $S_y(t) = \sum_{\tau \in \mathcal{T}_y(t)} \eta_\tau$.

$$(\mathbf{W}_t)_{:,y} = S_y(t) \cdot (2\mathbf{e}_y - \mathbf{1}).$$

For sample $i$ (class $y$), logits are $\mathbf{z} = \alpha_i(\mathbf{W}_t)_{:,y}$. Target logit: $z_y = \alpha_i S_y(t)(2-1) = \alpha_i S_y(t)$. Non-target logit $(k \neq y)$: $z_k = \alpha_i S_y(t)(0-1) = -\alpha_i S_y(t)$. The margin is $\Delta z_i(t) = z_y - z_k = 2\alpha_i S_y(t)$. Since $\eta_t$ is not summable ($a \leq 1$) and Random Reshuffling ensures constant visits, $S_y(t) \to \infty$. Thus, the margin diverges to $+\infty$. $\ell_i(\mathbf{W}_t) = \log(1 + (K-1)e^{-2\alpha_i S_y(t)})$. As $S_y(t) \to \infty$, $\ell_i \to 0$. So $L = \frac{1}{n}\sum_{i=1}^n \ell_i \to 0$.

**Implicit Bias.** The algorithm operates in epochs. In epoch $r$, the data indices are permuted by $\sigma_r$. For analysis, let us consider $t > T_0$ so the signs are stable. Let $\mathbf{W}_{rn}$ denote the weights at the start of epoch $r$. The weight update after one full epoch is:

$$\mathbf{W}_{(r+1)n} = \mathbf{W}_{rn} + \sum_{k=1}^n \eta_{rn+k-1} \mathbf{M}_{\sigma_r(k)}^{\text{sign}}$$

$$= \mathbf{W}_{rn} + \eta_{rn} \sum_{k=1}^n \mathbf{M}_{\sigma_r(k)}^{\text{sign}} + \sum_{k=1}^n (\eta_{rn+k-1} - \eta_{rn}) \mathbf{M}_{\sigma_r(k)}^{\text{sign}}.$$

Since $\sigma_r$ is a permutation of $\{1, \ldots, n\}$, the sum of update matrices is invariant to the order:

$$\sum_{k=1}^n \mathbf{M}_{\sigma_r(k)}^{\text{sign}} = \sum_{i=1}^n \mathbf{M}_i^{\text{sign}} = \bar{\mathbf{W}}_{\text{sign}}.$$

Thus, the recurrence relation is:

$$\mathbf{W}_{(r+1)n} = \mathbf{W}_{rn} + \eta_{rn} \bar{\mathbf{W}}_{\text{sign}} + E_r, \quad \text{where } E_r = \sum_{k=1}^n (\eta_{rn+k-1} - \eta_{rn}) \mathbf{M}_{\sigma_r(k)}^{\text{sign}}.$$

We bound the drift term $E_r$. Let $f(t) = c \cdot t^{-a}$. By the Mean Value Theorem, for $k \in \{1, \ldots, n\}$, there exists $\xi \in [rn, rn+n]$ such that $|\eta_{rn+k-1} - \eta_{rn}| \leq |f'(\xi)| \cdot n \leq ca(rn)^{-a-1}n$. The Frobenius norm of the update matrix is constant: $\|\mathbf{M}_i^{\text{sign}}\|_F = \|\mathbf{v}_i\|_2 \|\text{sign}(\mathbf{x}_i)\|_2 \leq \sqrt{K}\sqrt{1}$. Let $C_M = \sqrt{K}$. Then, the accumulated error in epoch $r$ is bounded by:

$$\|E_r\|_F \leq \sum_{k=1}^n |\eta_{rn+k-1} - \eta_{rn}| \|\mathbf{M}_{\sigma_r(k)}^{\text{sign}}\|_F \leq n \cdot \frac{can}{(rn)^{a+1}} C_M = \frac{caC_M n^2}{(rn)^{a+1}}.$$

By the triangle inequality and the bound on $\|E_r\|_F$, we have

$$\left\| \sum_{r=0}^{R-1} E_r \right\|_F \leq \sum_{r=0}^{R-1} \|E_r\|_F \leq \sum_{r=1}^{R-1} \frac{caC_M n^2}{(rn)^{a+1}} = \frac{caC_M n^2}{n^{a+1}} \sum_{r=1}^{R-1} \frac{1}{r^{a+1}}.$$

Since $a \in (0, 1]$, we have $a + 1 > 1$, hence the series $\sum_{r=1}^\infty r^{-(a+1)}$ converges. Therefore there exists a constant $C_E < \infty$ such that for all $R$,

$$\left\| \sum_{r=0}^{R-1} E_r \right\|_F \leq C_E.$$

Recalling that $S_R = \sum_{r=0}^{R-1} \eta_{rn} \to \infty$, it follows that

$$\lim_{R \to \infty} \frac{\left\| \sum_{r=0}^{R-1} E_r \right\|_F}{S_R} \leq \lim_{R \to \infty} \frac{C_E}{S_R} = 0.$$

Therefore, the accumulated error term becomes negligible compared to the signal direction $\bar{\mathbf{W}}_{\text{sign}}$. The parameter direction converges to:

$$\lim_{R \to \infty} \frac{\mathbf{W}_{Rn}}{\|\mathbf{W}_{Rn}\|_F} = \frac{\bar{\mathbf{W}}_{\text{sign}}}{\|\bar{\mathbf{W}}_{\text{sign}}\|_F}.$$

Since the parameter change within each epoch is $O(\eta_{rn} n)$ while $\|\mathbf{W}_{rn}\|_F \to \infty$, the relative deviation inside an epoch vanishes, and hence convergence of the subsequence $\{\mathbf{W}_{rn}\}$ implies convergence of the full sequence $\{\mathbf{W}_t\}$. $\qquad \square$

### H.2.3 IMPLICIT BIAS OF NORMALIZED-SGD (BATCH SIZE = 1) ON ORTHOGONAL SCALE-SKEWED DATASET

**Algorithm.** We analyze Per-sample Normalized-SGD with random shuffling and without replacement in the epoch. Training proceeds in epochs $r = 0, 1, \ldots$; at the start of each epoch, the dataset indices are shuffled via a random permutation $\sigma_r$. At the global time step $t$ (the $k$-th iteration of epoch $r$), the parameter is updated using the sample $i = \sigma_r(k)$ as follows:

$$\mathbf{W}_{t+1} = \mathbf{W}_t - \eta_t \frac{\nabla \ell_i(\mathbf{W}_t)}{\|\nabla \ell_i(\mathbf{W}_t)\|_F},$$

where $\ell_i(\cdot)$ is the Cross-Entropy loss and the learning rate follows a decay schedule $\eta_t = c \cdot t^{-a}$ with $a \in (0, 1]$.

**Geometric Definitions.** To analyze the optimization trajectory, we define the geometric properties of the target solution. For a sample $i$ with label $y_i$, let the *normalized centered label vector* be $\bar{\mathbf{u}}_{y_i} \triangleq (\mathbf{e}_{y_i} - \frac{1}{K}\mathbf{1})/\|\mathbf{e}_{y_i} - \frac{1}{K}\mathbf{1}\|_2$, and the *normalized input direction* be $\bar{\mathbf{x}}_i \triangleq \mathbf{x}_i/\|\mathbf{x}_i\|_2 = \mathbf{e}_{y_i}$ (due to the data construction). We define the *Canonical Update Matrix* for sample $i$ as:

$$\mathbf{M}_i \triangleq \bar{\mathbf{u}}_{y_i} \bar{\mathbf{x}}_i^\top.$$

The *Specific Bias Matrix* is defined as the sum of these canonical updates over the dataset: $\bar{\mathbf{W}} \triangleq \sum_{i=1}^{n} \mathbf{M}_i$.

**Theorem H.3** (Loss Convergence and Implicit Bias of Spectral-SGD/Normalized-SGD). *Consider the Random Reshuffling Per-sample Normalized-SGD algorithm initialized at $\mathbf{W}_0 = \mathbf{0}$ and trained on the Orthogonal Scale-Skewed dataset $\mathcal{D}$. The step size satisfies $\eta_t = c \cdot t^{-a}$ for $t \geq 1$ with constants $c > 0$ and $a \in (0, 1]$. Then, the algorithm satisfies:*

1. *__Loss Convergence:__ The training loss converges to zero:*

$$\lim_{t \to \infty} \mathcal{L}(\mathbf{W}_t) = 0.$$

2. *__Implicit Bias:__ The parameter matrix direction converges to the normalized Specific Bias Matrix $\bar{\mathbf{W}}$:*

$$\lim_{t \to \infty} \frac{\mathbf{W}_t}{\|\mathbf{W}_t\|_F} = \frac{\bar{\mathbf{W}}}{\|\bar{\mathbf{W}}\|_F}.$$

*Proof.* The proof relies on the invariant geometric update property established in Lemma H.4, which states that for any step $t$ utilizing sample $i$, the update is strictly $\mathbf{M}_i$. We analyze the convergence of the loss and the direction separately.

**Lemma H.4** (Invariant Normalized Gradient). *Consider the Algorithm on the orthogonal scale-skewed dataset with $\mathbf{W}_0 = \mathbf{0}$. For any time step $t \geq 0$, assuming the loss is not exactly zero, the normalized negative gradient direction is invariant to the current parameter magnitude $\|\mathbf{W}_t\|_F$ and the sample scale $\alpha_i$. Specifically, if sample $i$ is selected at step $t$, the update direction is strictly equal to the Canonical Update Matrix:*

$$-\frac{\nabla \ell_i(\mathbf{W}_t)}{\|\nabla \ell_i(\mathbf{W}_t)\|_F} = \mathbf{M}_i.$$

*Proof.* The proof relies on mathematical induction to establish a symmetry property of the weight matrix $\mathbf{W}_t$.

Let the current sample be indexed by $i$ with label $y = y_i$ and input $\mathbf{x}_i = \alpha_i \mathbf{e}_{y_i}$. Let $\mathbf{p} \in \mathbb{R}^K$ be the softmax probability vector where $p_k = \frac{\exp((\mathbf{W}_t \mathbf{x}_i)_k)}{\sum_j \exp((\mathbf{W}_t \mathbf{x}_i)_j)}$. The gradient of the Cross-Entropy loss is given by $\nabla \ell_i(\mathbf{W}_t) = (\mathbf{p} - \mathbf{e}_y)\mathbf{x}_i^\top$. The normalized negative gradient is:

$$-\frac{\nabla \ell_i(\mathbf{W}_t)}{\|\nabla \ell_i(\mathbf{W}_t)\|_F} = \frac{\mathbf{e}_y - \mathbf{p}}{\|\mathbf{e}_y - \mathbf{p}\|_2} \cdot \frac{\mathbf{x}_i^\top}{\|\mathbf{x}_i\|_2}. \tag{49}$$

Observing that $\frac{\mathbf{x}_i^\top}{\|\mathbf{x}_i\|_2} = \bar{\mathbf{x}}_i^\top$ matches the right component of $\mathbf{M}_i$, we must prove the left component matches $\bar{\mathbf{u}}_y$.

We claim that for all $t \geq 0$, the matrix $\mathbf{W}_t$ satisfies the *Column-wise Off-diagonal Equality* property: for any column $j \in \{1, \ldots, K\}$, all off-diagonal entries are equal. That is, $(\mathbf{W}_t)_{p,j} = (\mathbf{W}_t)_{q,j}$ for all $p, q \neq j$.

*Base Case ($t = 0$):* $\mathbf{W}_0 = \mathbf{0}$, so all entries are 0. The property holds.

*Inductive Step:* Assume the property holds for $\mathbf{W}_t$. Consider the update with sample $i$ (class $y$). The logits are $\mathbf{z} = \mathbf{W}_t \mathbf{x}_i = \alpha_i (\mathbf{W}_t)_{:,y}$. By the induction hypothesis, the column $(\mathbf{W}_t)_{:,y}$ has equal off-diagonal entries. Thus, for all non-target classes $k \neq y$, the logits $z_k$ are identical. Consequently, the softmax probabilities for non-target classes are identical:

$$p_k = \frac{e^{z_k}}{e^{z_y} + \sum_{j \neq y} e^{z_j}} = \epsilon, \quad \forall k \neq y.$$

The gradient update is proportional to $(\mathbf{e}_y - \mathbf{p})\mathbf{x}_i^\top$. Since $\mathbf{x}_i$ is zero everywhere except index $y$, only the $y$-th column of $\mathbf{W}$ is updated. The update vector for this column is proportional to $\mathbf{v} = \mathbf{e}_y - \mathbf{p}$. Its components are $v_y = 1 - p_y$ and $v_k = -\epsilon$ for $k \neq y$. Since the update adds the same value $(-\eta_t \cdot C \cdot \epsilon)$ to all off-diagonal entries of column $y$, and they were previously equal, they remain equal in $\mathbf{W}_{t+1}$. The property is preserved.

Using the symmetry $p_k = \epsilon$ for $k \neq y$, and the fact $\sum p_j = 1$, we have $p_y + (K - 1)\epsilon = 1$, implies $1 - p_y = (K - 1)\epsilon$. The vector $(\mathbf{e}_y - \mathbf{p})$ can be written as:

$$\mathbf{e}_y - \mathbf{p} = \begin{bmatrix} -\epsilon \\ \vdots \\ 1 - p_y \\ \vdots \\ -\epsilon \end{bmatrix} = \begin{bmatrix} -\epsilon \\ \vdots \\ (K-1)\epsilon \\ \vdots \\ -\epsilon \end{bmatrix} = K\epsilon \left( \mathbf{e}_y - \frac{1}{K}\mathbf{1} \right).$$

Let $\lambda = K\epsilon$. Since the loss is non-zero, $\epsilon > 0 \implies \lambda > 0$. Substituting this into Eq. (49):

$$\frac{\mathbf{e}_y - \mathbf{p}}{\|\mathbf{e}_y - \mathbf{p}\|_2} = \frac{\lambda(\mathbf{e}_y - \frac{1}{K}\mathbf{1})}{\lambda\|\mathbf{e}_y - \frac{1}{K}\mathbf{1}\|_2} = \frac{\mathbf{e}_y - \frac{1}{K}\mathbf{1}}{\|\mathbf{e}_y - \frac{1}{K}\mathbf{1}\|_2} \equiv \bar{\mathbf{u}}_y.$$

Thus, the normalized gradient is strictly $\mathbf{M}_i$. $\qquad\square$

**Loss Convergence.** Recall that for sample $i$, the input is $\mathbf{x}_i = \alpha_i \mathbf{e}_{y_i}$ and the update matrix is $\mathbf{M}_i = \bar{\mathbf{u}}_{y_i} \mathbf{e}_{y_i}^\top$. Since $\mathbf{M}_i$ is non-zero only in the $y_i$-th column, the update at step $t$ only affects the column of $\mathbf{W}_t$ corresponding to the label of the current sample. Thus, the dynamics of the $K$ columns of $\mathbf{W}$ are mutually independent.

Let $(\mathbf{W}_t)_{:,c}$ denote the $c$-th column of $\mathbf{W}_t$. Since $\mathbf{W}_0 = \mathbf{0}$, the column at time $t$ is the sum of all historical updates applied to class $c$. Let $\mathcal{T}_c(t) = \{\tau < t \mid \text{sample at step } \tau \text{ has label } c\}$ be the set of time steps where class $c$ was sampled. We have:

$$(\mathbf{W}_t)_{:,c} = \sum_{\tau \in \mathcal{T}_c(t)} \eta_\tau (\mathbf{M}_{i_\tau})_{:,c} = \left( \sum_{\tau \in \mathcal{T}_c(t)} \eta_\tau \right) \bar{\mathbf{u}}_c.$$

Define the cumulative step size for class $c$ as $S_c(t) \triangleq \sum_{\tau \in \mathcal{T}_c(t)} \eta_\tau$.

Consider an arbitrary sample $i$ with label $y$ and scale $\alpha_i$. The logit vector is $\mathbf{z} = \mathbf{W}_t \mathbf{x}_i = \alpha_i (\mathbf{W}_t)_{:,y}$. Substituting the column expression:

$$\mathbf{z} = \alpha_i S_y(t) \bar{\mathbf{u}}_y.$$

We analyze the margin between the target class $y$ and any non-target class $k \neq y$. Recall $\bar{\mathbf{u}}_y = \lambda(\mathbf{e}_y - \frac{1}{K}\mathbf{1})$ for some $\lambda > 0$.

$$z_y = \alpha_i S_y(t) \cdot \lambda(1 - 1/K),$$
$$z_k = \alpha_i S_y(t) \cdot \lambda(0 - 1/K).$$

The margin is $\Delta z(t) = z_y - z_k = \alpha_i S_y(t)\lambda$. Under Random Reshuffling with data completeness, class $y$ is visited at least once per epoch. Since $\eta_t = \Theta(t^{-a})$ with $a \leq 1$, the series $\sum \eta_t$ diverges. Consequently, $\lim_{t\to\infty} S_y(t) = +\infty$, implying $\lim_{t\to\infty} \Delta z(t) = +\infty$.

The Cross-Entropy loss is strictly decreasing with respect to the margin:

$$\ell_i(\mathbf{W}_t) = \log\left(1 + \sum_{k\neq y} e^{-\Delta z(t)}\right).$$

As $\Delta z(t) \to \infty$, the term $e^{-\Delta z(t)} \to 0$. Thus, $\lim_{t\to\infty} \ell_i(\mathbf{W}_t) = \log(1) = 0$.

**Implicit Bias** We analyze the trajectory using a deterministic recurrence relation at the epoch level.

Let $\mathbf{W}_{rN}$ denote the weights at the start of epoch $r$. The weights at the start of the next epoch are:

$$\mathbf{W}_{(r+1)n} = \mathbf{W}_{rn} + \sum_{k=1}^{n} \eta_{rn+k-1}\mathbf{M}_{\sigma_r(k)}$$

$$= \mathbf{W}_{rn} + \eta_{rn}\sum_{k=1}^{n}\mathbf{M}_{\sigma_r(k)} + \sum_{k=1}^{n}(\eta_{rn+k-1} - \eta_{rn})\mathbf{M}_{\sigma_r(k)}.$$

Crucially, since $\sigma_r$ is a permutation of $\{1,\ldots,n\}$, the sum of update matrices is invariant and exactly equals $\bar{\mathbf{W}} = \sum_{i=1}^{n}\mathbf{M}_i$. Thus, the recurrence relation is:

$$\mathbf{W}_{(r+1)n} = \mathbf{W}_{rn} + \eta_{rn}\bar{\mathbf{W}} + E_r, \quad \text{where } E_r = \sum_{k=1}^{n}(\eta_{rn+k-1} - \eta_{rn})\mathbf{M}_{\sigma_r(k)}. \tag{50}$$

We bound the drift term $E_r$. Let $f(t) = ct^{-a}$. By the Mean Value Theorem, for $k \in \{1,\ldots,n\}$, there exists $\xi \in [rn, rn+n]$ such that $|\eta_{rn+k-1} - \eta_{rn}| \leq |f'(\xi)| \cdot n \leq ca(rn)^{-a-1}n$. The Frobenius norm of the update matrix is constant: $\|\mathbf{M}_i\|_F = \|\bar{\mathbf{u}}_{y_i}\|_2\|\bar{\mathbf{x}}_i\|_2 = 1 \cdot 1 = 1$. Then, the accumulated error in epoch $r$ is bounded by:

$$\|E_r\|_F \leq \sum_{k=1}^{n}|\eta_{rn+k-1} - \eta_{rn}|\|\mathbf{M}_{\sigma_r(k)}\|_F \leq n \cdot \frac{can}{(rn)^{a+1}} \cdot 1 = \frac{can^2}{(rn)^{a+1}}.$$

Summing Eq. (50) from epoch 0 to $R-1$ (with $\mathbf{W}_0 = \mathbf{0}$):

$$\mathbf{W}_{Rn} = \left(\sum_{r=0}^{R-1}\eta_{rn}\right)\bar{\mathbf{W}} + \sum_{r=0}^{R-1}E_r.$$

By the triangle inequality and the bound on $\|E_r\|_F$, we analyze the accumulated error series:

$$\left\|\sum_{r=0}^{R-1}E_r\right\|_F \leq \sum_{r=0}^{R-1}\|E_r\|_F \leq \sum_{r=1}^{R-1}\frac{can^2}{(rn)^{a+1}} + \|E_0\|_F = \frac{can^2}{n^{a+1}}\sum_{r=1}^{R-1}\frac{1}{r^{a+1}} + \|E_0\|_F.$$

Since $a \in (0,1]$, we have $a+1 > 1$, hence the series $\sum_{r=1}^{\infty} r^{-(a+1)}$ converges. Therefore, there exists a constant $C_E < \infty$ such that for all $R$:

$$\left\|\sum_{r=0}^{R-1}E_r\right\|_F \leq C_E.$$

Let $S_R = \sum_{r=0}^{R-1}\eta_{rn}$. Since $a \leq 1$, the learning rate series diverges, so $S_R \to \infty$. It follows that the ratio of error to signal vanishes:

$$\lim_{R\to\infty}\frac{\left\|\sum_{r=0}^{R-1}E_r\right\|_F}{S_R} \leq \lim_{R\to\infty}\frac{C_E}{S_R} = 0.$$

Therefore, the accumulated error term becomes negligible compared to the signal direction $\bar{\mathbf{W}}$. The parameter direction converges to the direction of the accumulated signal:

$$\lim_{R\to\infty} \frac{\mathbf{W}_{Rn}}{\|\mathbf{W}_{Rn}\|_F} = \lim_{R\to\infty} \frac{S_R\bar{\mathbf{W}} + \sum E_r}{\|S_R\bar{\mathbf{W}} + \sum E_r\|_F} = \frac{\bar{\mathbf{W}}}{\|\bar{\mathbf{W}}\|_F}.$$

Since the parameter change within each epoch is $O(\eta_{rn}n)$ while $\|\mathbf{W}_{rn}\|_F \to \infty$, the relative deviation inside an epoch vanishes, and hence convergence of the subsequence $\{\mathbf{W}_{rn}\}$ implies convergence of the full sequence $\{\mathbf{W}_t\}$.

$\square$

# I   Extension to Exponential Loss

In this section, we demonstrate that our main convergence results (Theorems 4.1, 4.4, and 4.7) extend to the Exponential Loss. The analysis follows the unified framework established in the previous sections. We show that by choosing the Exponential Loss itself as the proxy function, i.e., $\mathcal{G}(\mathbf{W}) = L(\mathbf{W})$, all the required geometric and stochastic properties (Lemmas C.2–C.13) are satisfied, often with tighter constants than the Cross-Entropy case.

## I.1   Setup and Definitions

We consider the multi-class Exponential Loss defined as:

$$L_{\exp}(\mathbf{W}) := \frac{1}{n} \sum_{i\in[n]} \sum_{c\neq y_i} e^{-(\mathbf{e}_{y_i}-\mathbf{e}_c)^\top \mathbf{W}\mathbf{x}_i}. \tag{51}$$

For this loss, we define the proxy function simply as the loss itself:

$$\mathcal{G}_{\exp}(\mathbf{W}) := L_{\exp}(\mathbf{W}). \tag{52}$$

Note that under this definition, the compatibility ratio $\frac{\mathcal{G}_{\exp}(\mathbf{W})}{L_{\exp}(\mathbf{W})} \equiv 1$, which trivially satisfies Lemma C.7.

## I.2   Verification of Geometric Properties

We verify that $L_{\exp}$ satisfies the gradient bounds, smoothness, and stability conditions required by our descent lemmas.

**Lemma I.1** (Geometric Properties of Exponential Loss). *Under Assumption 3.2 ($\|\mathbf{x}_i\|_1 \leq R$), for any $\mathbf{W}, \mathbf{\Delta} \in \mathbb{R}^{k\times d}$:*

*(i) **Gradient Bound:** $\gamma\mathcal{G}_{\exp}(\mathbf{W}) \leq \|\nabla L_{\exp}(\mathbf{W})\|_* \leq 2R\mathcal{G}_{\exp}(\mathbf{W})$.*

*(ii) **Hessian/Smoothness:** The quadratic form is bounded by the proxy:*

$$\mathbf{x}_i^\top \mathbf{\Delta}^\top \nabla^2 \ell_i(\mathbf{W})\mathbf{\Delta}\mathbf{x}_i \leq 4R^2\|\mathbf{\Delta}\|^2 e^{-(\mathbf{e}_{y_i}-\mathbf{e}_c)^\top \mathbf{W}\mathbf{x}_i}.$$

*Consequently, the descent lemma second-order term is bounded by $2R^2\|\mathbf{\Delta}\|^2\mathcal{G}_{\exp}(\mathbf{W})$.*

*(iii) **Stability (Ratio Property):** $\mathcal{G}_{\exp}(\mathbf{W}+\mathbf{\Delta}) \leq e^{2R\|\mathbf{\Delta}\|_{\max}}\mathcal{G}_{\exp}(\mathbf{W})$.*

*Proof.* **(i) Gradient.** The gradient is given by $\nabla L_{\exp}(\mathbf{W}) = -\frac{1}{n}\sum_i \sum_{c\neq y_i} e^{-(\mathbf{e}_{y_i}-\mathbf{e}_c)^\top \mathbf{W}\mathbf{x}_i}(\mathbf{e}_{y_i}-\mathbf{e}_c)\mathbf{x}_i^\top$. Taking the dual norm (using $\|\mathbf{x}_i\|_1 \leq R$ and $\|\mathbf{e}_{y_i}-\mathbf{e}_c\|_1 \leq 2$):

$$\begin{aligned}
\|\nabla L_{\exp}(\mathbf{W})\|_* &\leq \frac{1}{n}\sum_{i,c\neq y_i} e^{-(\mathbf{e}_{y_i}-\mathbf{e}_c)^\top \mathbf{W}\mathbf{x}_i}\|(\mathbf{e}_{y_i}-\mathbf{e}_c)\mathbf{x}_i^\top\| \\
&\leq 2R\cdot\frac{1}{n}\sum_{i,c\neq y_i} e^{-(\mathbf{e}_{y_i}-\mathbf{e}_c)^\top \mathbf{W}\mathbf{x}_i} = 2R\mathcal{G}_{\exp}(\mathbf{W}).
\end{aligned}$$

The lower bound follows from the separability Assumption 3.1 similarly to Lemma C.5.

**(ii) Hessian.** Let $z_{i,c} = (\mathbf{e}_{y_i} - \mathbf{e}_c)^\top \mathbf{W} \mathbf{x}_i$. The second derivative of $e^{-z}$ is $e^{-z}$. Thus, $\nabla^2 L_{\exp} \preceq \frac{1}{n} \sum_{i,c \neq y_i} e^{-z_{i,c}} \|(\mathbf{e}_{y_i} - \mathbf{e}_c)\mathbf{x}_i^\top\|^2$. Since $\|(\mathbf{e}_{y_i} - \mathbf{e}_c)\mathbf{x}_i^\top\| \leq 2R$, the quadratic term scales with the loss itself.

**(iii) Stability.** Note that $L_{\exp}(\mathbf{W} + \boldsymbol{\Delta})$ is a sum of terms of the form $e^{-(\mathbf{e}_{y_i} - \mathbf{e}_c)^\top (\mathbf{W} + \boldsymbol{\Delta})\mathbf{x}_i}$.

$$
\begin{aligned}
e^{-(\mathbf{e}_{y_i} - \mathbf{e}_c)^\top (\mathbf{W} + \boldsymbol{\Delta})\mathbf{x}_i} &= e^{-(\mathbf{e}_{y_i} - \mathbf{e}_c)^\top \mathbf{W} \mathbf{x}_i} e^{-(\mathbf{e}_{y_i} - \mathbf{e}_c)^\top \boldsymbol{\Delta} \mathbf{x}_i} \\
&\leq e^{-z_{i,c}} e^{\|\mathbf{e}_{y_i} - \mathbf{e}_c\|_1 \|\boldsymbol{\Delta}\|_\infty \|\mathbf{x}_i\|_1} \\
&\leq e^{-z_{i,c}} e^{2R\|\boldsymbol{\Delta}\|_{\max}}.
\end{aligned}
$$

Summing over $i, c$ yields $\mathcal{G}_{\exp}(\mathbf{W} + \boldsymbol{\Delta}) \leq e^{2R\|\boldsymbol{\Delta}\|_{\max}} \mathcal{G}_{\exp}(\mathbf{W})$. $\qquad \square$

**Lemma I.2** (Gradient Stability for Exp Loss). *For any two weight matrices $\mathbf{W}, \mathbf{W}' \in \mathbb{R}^{k \times d}$, let $\boldsymbol{\Delta} = \mathbf{W}' - \mathbf{W}$. Suppose the data satisfies $\|\mathbf{x}_i\|_1 \leq R$. Then, the entry-wise 1-norm of the gradient difference is bounded by:*

$$
\|\nabla L_{\exp}(\mathbf{W}') - \nabla L_{\exp}(\mathbf{W})\|_1 \leq 2R \left( e^{2R\|\boldsymbol{\Delta}\|_{\max}} - 1 \right) \mathcal{G}_{\exp}(\mathbf{W}).
$$

*Proof.* The gradient of the exponential loss is given by $\nabla L_{\exp}(\mathbf{W}) = -\frac{1}{n} \sum_i \sum_{c \neq y_i} e^{-z_{i,c}} (\mathbf{e}_{y_i} - \mathbf{e}_c)\mathbf{x}_i^\top$, where $z_{i,c} = (\mathbf{e}_{y_i} - \mathbf{e}_c)^\top \mathbf{W} \mathbf{x}_i$. Consider the gradient difference term-by-term:

$$
\begin{aligned}
\|\nabla L_{\exp}(\mathbf{W}') - \nabla L_{\exp}(\mathbf{W})\|_1 &\leq \frac{1}{n} \sum_{i=1}^n \sum_{c \neq y_i} \left| e^{-z'_{i,c}} - e^{-z_{i,c}} \right| \cdot \|(\mathbf{e}_{y_i} - \mathbf{e}_c)\mathbf{x}_i^\top\|_1 \\
&\leq \frac{2R}{n} \sum_{i=1}^n \sum_{c \neq y_i} \left| e^{-z'_{i,c}} - e^{-z_{i,c}} \right|.
\end{aligned}
$$

Let $\delta_{i,c} = z'_{i,c} - z_{i,c} = (\mathbf{e}_{y_i} - \mathbf{e}_c)^\top \boldsymbol{\Delta} \mathbf{x}_i$. The magnitude of the perturbation is bounded by:

$$
|\delta_{i,c}| \leq \|\mathbf{e}_{y_i} - \mathbf{e}_c\|_1 \|\boldsymbol{\Delta} \mathbf{x}_i\|_\infty \leq 2\|\boldsymbol{\Delta}\|_\infty \|\mathbf{x}_i\|_1 \leq 2R\|\boldsymbol{\Delta}\|_{\max}.
$$

Using the elementary inequality $|e^{-a} - e^{-b}| = e^{-a}|e^{-(b-a)} - 1| \leq e^{-a}(e^{|b-a|} - 1)$, we have:

$$
\left| e^{-z'_{i,c}} - e^{-z_{i,c}} \right| \leq e^{-z_{i,c}} \left( e^{|\delta_{i,c}|} - 1 \right) \leq e^{-z_{i,c}} \left( e^{2R\|\boldsymbol{\Delta}\|_{\max}} - 1 \right).
$$

Substituting this back into the sum:

$$
\|\nabla L_{\exp}(\mathbf{W}') - \nabla L_{\exp}(\mathbf{W})\|_1 \leq 2R \left( e^{2R\|\boldsymbol{\Delta}\|_{\max}} - 1 \right) \underbrace{\frac{1}{n} \sum_{i=1}^n \sum_{c \neq y_i} e^{-z_{i,c}}}_{\mathcal{G}_{\exp}(\mathbf{W})}.
$$

This completes the proof. $\qquad \square$

## I.3 Verification of Stochastic Properties

Crucially for our stochastic analysis, the gradient noise and momentum accumulation for Exponential Loss are also naturally bounded by the proxy function.

**Lemma I.3** (Stochastic Noise Bound for Exp Loss). *The mini-batch gradient noise $\boldsymbol{\Xi} = \nabla L_{\mathcal{B}}(\mathbf{W}) - \nabla L(\mathbf{W})$ satisfies:*

$$
\|\boldsymbol{\Xi}\|_{\text{sum}} \leq 2(m-1)R\mathcal{G}_{\exp}(\mathbf{W}).
$$

*This ensures that Lemmas C.11, C.12, and C.13 hold directly for Exponential Loss.*

*Proof.* The single-sample gradient norm is $\|\nabla \ell_i(\mathbf{W})\|_{\text{sum}} = \|\sum_{c \neq y_i} e^{-z_{i,c}}(\mathbf{e}_{y_i} - \mathbf{e}_c)\mathbf{x}_i^\top\|_1 \leq 2R \sum_{c \neq y_i} e^{-z_{i,c}}$. Summing over all $i$:

$$
\sum_{i=1}^n \|\nabla \ell_i(\mathbf{W})\|_{\text{sum}} \leq 2nR L_{\exp}(\mathbf{W}) = 2nR\mathcal{G}_{\exp}(\mathbf{W}).
$$

Using the finite population correction argument from Lemma C.11, the mini-batch noise is bounded by $\frac{m-1}{n} \sum \|\nabla \ell_i\|_{\text{sum}} \leq 2(m-1)R\mathcal{G}_{\exp}(\mathbf{W})$. $\qquad \square$

## I.4   Conclusion

Since $L_{\exp}$ and $\mathcal{G}_{\exp}$ satisfy all the geometric inequalities and stochastic noise bounds (Lemma used in the proofs of Theorems 1–4, the convergence rates derived for Cross-Entropy apply directly to Exponential Loss. Specifically, the algorithms converge to the max-margin solution with the same asymptotic rates, as the self-bounding property of the exponential function provides strictly tighter control over the optimization trajectory.

## J   Empirical Validation of Theorem 4.10

**Setup for Per-Sample Regime.**   To validate our theoretical findings regarding the unique implicit bias in the batch-size-one regime, we construct a synthetic *Orthogonal Scale-Skewed* dataset with $n = 500$ samples, $K = 10$ classes, and dimension $d = 10$. The data is generated to explicitly decouple class frequency from sample hardness: class counts are sampled from a multinomial distribution to introduce label imbalance, while the feature scale $\alpha_i$ for each sample $\mathbf{x}_i = \alpha_i \mathbf{e}_{y_i}$ is drawn uniformly from heterogeneous class-specific ranges to introduce scale variation. We train Per-sample SignSGD and Per-sample Normalized-SGD (corresponding to $b = 1$) for $T = 5,000$ iterations, initialized at $\mathbf{W}_0 = \mathbf{0}$. We use a polynomial learning rate schedule $\eta_t = \eta_0 t^{-a}$ with decay rate $a = 0.5$ and $\eta_0 = 0.5$. We evaluate convergence by measuring the cosine similarity to the theoretical bias direction $\bar{\mathbf{W}}$ (Definition 4.9) and the true max-margin solution $\mathbf{W}^*$, as well as the relative error to the optimal margin $\gamma^*$ under $\ell_2$, $\ell_\infty$, and spectral norm geometries.

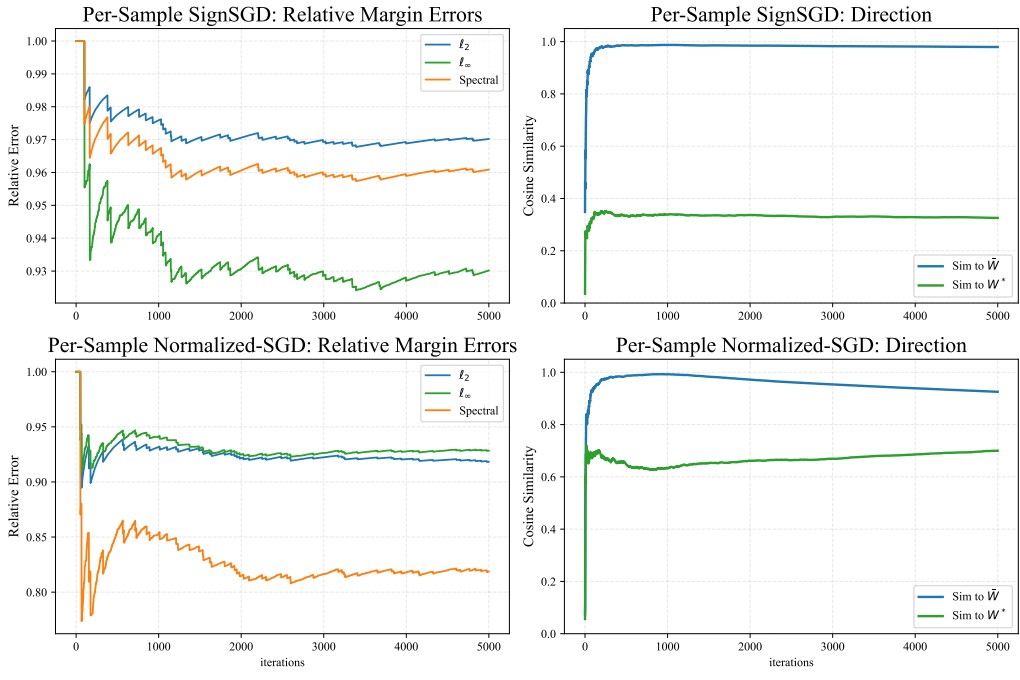

*Figure 2*

**Empirical validation of theory.**   Figure 2 confirms our theoretical predictions for the per-sample regime ($b = 1$). The left column shows the relative margin error with respect to the optimal margin $\gamma^*$ under $\ell_2$, $\ell_\infty$, and spectral norm geometries. Regardless of the norm considered, the relative error remains high and does not converge to zero, indicating that per-sample stochastic updates fail to maximize the margin under any of these geometries. In contrast, the right column demonstrates the directional convergence of the iterates. While the cosine similarity to the true max-margin solution $\mathbf{W}^*$ (green) stagnates well below 1, the similarity to our theoretically derived bias direction $\bar{\mathbf{W}}$ (blue) steadily approaches 1. This provides strong empirical evidence that the implicit bias is governed by the sample-averaged direction $\bar{\mathbf{W}}$ rather than the geometric max-margin solution.

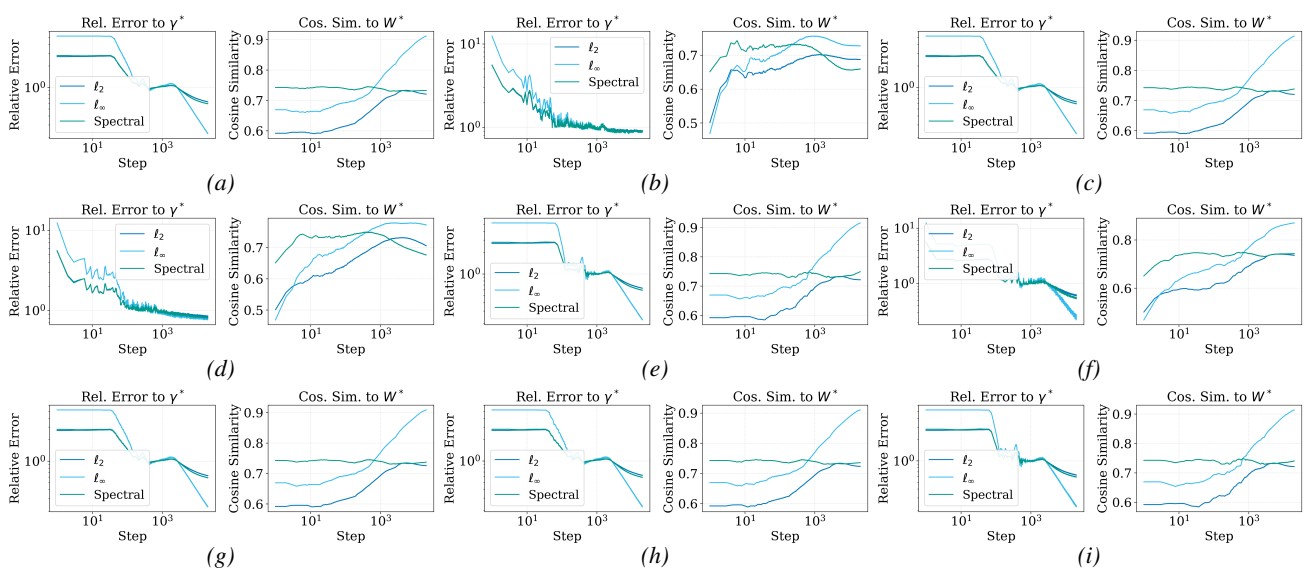

*Figure 3.* Empirical validation of the implicit bias of normalized steepest descent under the $\ell_\infty$ norm. (a) SignSGD with full-batch size $b = 200$. (b) SignSGD with mini-batch size $b = 20$. (c) Signum with momentum $\beta_1 = 0.5$ and full-batch size $b = 200$. (d) Signum with momentum $\beta_1 = 0.5$ and mini-batch size $b = 20$. (e) Signum with momentum $\beta_1 = 0.99$ and full-batch size $b = 200$. (f) Signum with momentum $\beta_1 = 0.99$ and mini-batch size $b = 20$. (g) VR-SignSGD with mini-batch size $b = 20$. (h) VR-Signum with momentum $\beta_1 = 0.5$ and mini-batch size $b = 20$. (i) VR-Signum with momentum $\beta_1 = 0.99$ and mini-batch size $b = 20$.

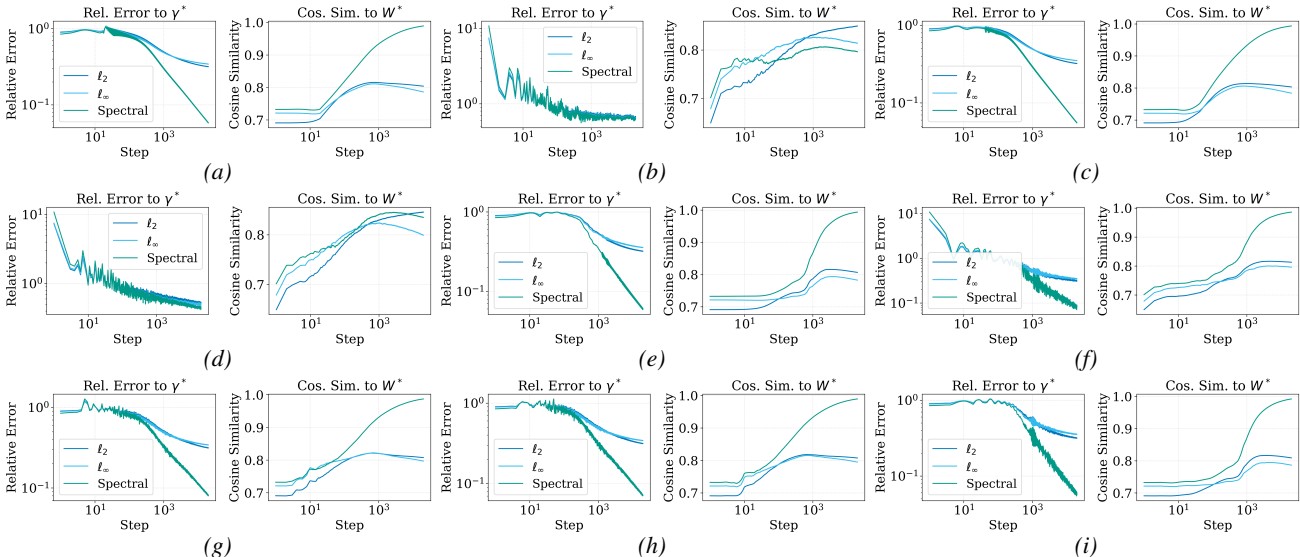

*Figure 4.* Empirical validation of the implicit bias of normalized steepest descent under the Spectral norm. (a) Spectral-SGD with full-batch size $b = 200$. (b) Spectral-SGD with mini-batch size $b = 20$. (c) Muon with momentum $\beta_1 = 0.5$ and full-batch size $b = 200$. (d) Muon with momentum $\beta_1 = 0.5$ and mini-batch size $b = 20$. (e) Muon with momentum $\beta_1 = 0.99$ and full-batch size $b = 200$. (f) Muon with momentum $\beta_1 = 0.99$ and mini-batch size $b = 20$. (g) VR-Spectral-SGD with mini-batch size $b = 20$. (h) VR-Muon with momentum $\beta_1 = 0.5$ and mini-batch size $b = 20$. (i) VR-Muon with momentum $\beta_1 = 0.99$ and mini-batch size $b = 20$.

# K   Additional Experimental Results

In this section, we present additional experimental results for SignSGD and Signum under the $\ell_\infty$ geometry, as well as Spectral-SGD and Muon under the spectral norm, complementing the $\ell_2$-norm results reported in the main text.

These results in Figure 3 and 4 exhibit the same qualitative behavior as in the $\ell_2$ case discussed in the main text.

## K.1 MNIST dataset results

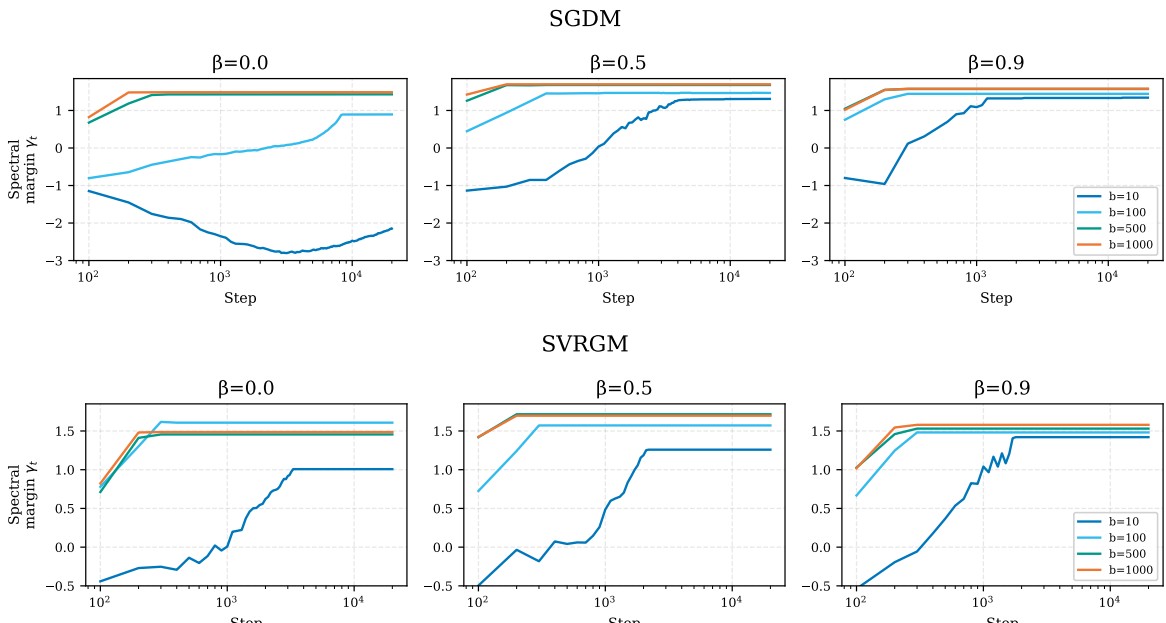

*Figure 5.* Spectral margin $\gamma^{\mathbf{W_1}, \mathbf{W_2}}$ along training on MNIST for a two-layer neural network trained with Spectral-SGD/Muon. The top row shows SGDM (Spectral-SGD with momentum), and the bottom row shows SVRGM (its variance-reduced counterpart).We observe that for SGDM (top), small batch sizes lead to a collapse of the spectral margin and deviate significantly from the full-batch behavior, while increasing momentum progressively restores the margin toward the full-batch regime. In contrast, SVRGM (bottom) consistently recovers a margin close to the full-batch solution across all batch sizes and momentum values, even when $b$ is small, highlighting the stabilizing effect of variance reduction.

