# OpenReview forum: "The Implicit Bias of Steepest Descent with Mini-batch Stochastic Gradient"
_ICML.cc/2026/Conference — ICML 2026 regular_

### Official Review · Reviewer_BEoD · 2026-03-08

**Soundness:** 4
**Presentation:** 4
**Significance:** 4
**Originality:** 3
**Overall Recommendation:** 5
**Confidence:** 4

**Summary:**

This paper studies the convergence of a multi-class linear classifier under logistic loss. The authors rigorously prove that without momentum, SGD approximates the max-margin solution under large batches; with momentum, smaller batch sizes can also lead to convergence. Additionally, they prove that variance reduction scheme guarantees convergence, independent of batch size and momentum. They further show a batch size 1 example to demonstrate that small batch sizes bias towards weighted average over samples, while large/full batch size bias towards max-margin solution. Experimental verification of theory is also provided.

**Compliance With Llm Reviewing Policy:**

Affirmed.

**Final Justification:**

This is a technically solid paper that studies several important and fundamental aspects of training. One potential limitation, if there has to be any, is that the model analyzed is relatively simple, and thus the proof techniques do not appear to be substantially new.

The authors address additional details of their theory interpretation in the rebuttal. All my concerns have been addressed, and I will maintain my evaluation.

**Key Questions For Authors:**

This paper is well-written and has complete results. I don't really have many questions. Below are some of the points that I feel could be further clarified:
- For the positive effective margin condition: I wonder how much difference is the bound on batch size $b$ compared to the large batch condition. It would be better if there were explicit expressions, or at least some instances that characterize the gap if the requirement on $b$ cannot be solved explicitly.
- What is the intuition of the large batch size threshold without momentum? More specifically, are there any technical evidence/intuition of what happens when batch size is small under the same condition as Therorem 4.1, since it cannot be inferred from the setting of Theorem 4.3 as the authors discussed in Remark 4.4?

**Limitations:**

yes

**Strengths And Weaknesses:**

This paper is well written, and complete. The theoretical results are rich and intuitive. Empirical validation is also sufficient.

The paper mainly discusses the effect of different training strategies, including changing batch size, adding momentum, and adding variance reduction. These are all important perspectives in training and generalization. The results are nice; the proof techniques may not be new.

---

> ### Author Rebuttal · Authors · 2026-03-30
>
> We sincerely thank you for your insightful feedback and thoughtful suggestions on our work. Here, we address your comments/suggestions below:
>
> ---
> **Comparison of batch-size scaling under two conditions:** Thank you for raising this point. We agree that the difference between the large-batch condition and the positive effective margin condition should be made more explicit. For the no-momentum case (Theorem 4.1), the requirement is $\rho=\gamma-4\Big(\frac{n}{b}-1\Big)R>0$, which is equivalent to $b>\frac{4Rn}{\gamma+4R}.$ Thus, the required batch size is explicitly $b_{\mathrm{LB}}=\Omega\Big(n\frac{R}{\gamma+R}\Big).$ For the momentum case, the positive effective margin condition is $\rho=\gamma-2(1-\beta_1)m(m^2-1)R>0.$ Unlike the no-momentum case, the condition is not a pure batch-size threshold, but rather a batch-momentum trade-off. Fix $\beta_1$ then we have $b=\Omega\Big(n(\frac{(1-\beta_1)R}{\gamma})^{1/3}\Big).$ Compared with the large batch condition $b_{\mathrm{LB}}=\Omega\Big(\frac{Rn}{\gamma+R}\Big)$, momentum changes the sufficient batch-size scaling from essentially linear in $n$ to a cubic-root dependence on $(1-\beta_1)$. In particular, as $\beta_1\to 1$, the required batch size can become much smaller; e.g., if $1-\beta_1=O(n^{-3})$, then the above condition allows $b=\Omega(1)$. We agree that this comparison is useful for readers, and in the final version we will add an explicit discussion.
>
> ---
> **Intuition and necessity of the large-batch threshold:** The large-batch requirement in Theorem 4.1 comes from a signal-to-noise trade-off in the descent direction. In the no-momentum setting, the update direction is given by the mini-batch steepest direction, which deviates from the full gradient due to sampling noise. In Appendix D, this deviation appears explicitly in the descent inequality as an error term of order $(m-1)RG(W_t).$
> This term competes directly with the margin signal $\gamma G(W_t)$, leading to the effective margin $\rho=\gamma-4(m-1)R.$ Thus, the large-batch condition $\rho>0$ ensures that the update direction maintains a positive projection onto the max-margin separator, which is the key invariant driving the proof.
>
> When the batch size is small (i.e., $m=n/b$ is large) and no additional stabilization such as momentum or variance reduction is used, this noise term dominates the signal, and the effective margin becomes non-positive and the proof mechanism breaks down. Technically, we cannot guarantee the same the monotonic decrease of the loss and consistent progress toward the max-margin direction. Importantly, this is not only a proof issue: our batch-size-one result in Theorem 4.9 shows that in the extreme small-batch regime, plain stochastic steepest descent can indeed converge to a fundamentally different implicit bias from the full-batch max-margin solution. Hence, the threshold in Theorem 4.1 reflects a genuine limitation of plain mini-batch updates without additional stabilization.
>
> ---
> **Why Theorem 4.3 does not cover the $\beta_1=0$ case:** As stated in Remark 4.4, The positive effective margin condition in Theorem 4.3 does not directly recover the large-batch condition by setting $\beta_1=0$. This is because the analysis of Theorem 4.3 relies on momentum-specific temporal averaging of stochastic errors, where momentum-specific exponential weighting allows the proof to exploit the epoch-wise zero-sum structure of random reshuffling $\sum_{k=0}^{m-1} \Big(\nabla L_{B_{tm+k}}(W)-\nabla L(W)\Big)=0$. When $\beta_1=0$, this temporal averaging disappears, and the argument reduces to controlling only the instantaneous mini-batch error, so the same proof no longer applies. We view this as a limitation of the momentum-specific proof template rather than of the result itself. Importantly, this does not affect the main conclusion: momentum introduces a distinct stabilization mechanism which enables approximate max-margin recovery in small batch-size regimes where plain mini-batch updates do not.

---

> > ### Author Rebuttal · Reviewer_BEoD · 2026-04-01
> >
> > I would like to thank the authors for the detailed reply.
> >
> > This addresses my concerns. Please consider updating the corresponding parts of the paper. I will keep my score.

---

> > > ### Author Response · Authors · 2026-04-04
> > >
> > > Dear Reviewer BEoD,
> > >
> > > Thank you for your kind feedback and positive evaluation. We are glad to hear that our rebuttal has addressed your concerns, and we will include these clarifications on the batch-size conditions and their intuition in the revised manuscript.
> > >
> > > Thank you again for your effort in reviewing our work.
> > >
> > > Best,\
> > > Authors

---

### Official Review · Reviewer_WgdY · 2026-03-13

**Soundness:** 3
**Presentation:** 4
**Significance:** 2
**Originality:** 3
**Overall Recommendation:** 4
**Confidence:** 3

**Summary:**

This paper studies the implicit bias of mini-batch stochastic steepest descent methods in multi-class classification with linearly separable data. The authors show how different method and hyperparameter choices can affect the solution the method converges to. More specifically:
- How batch size affects whether stochastic steepest descent recovers the same max-margin solution as full-batch gradient descent.
- How momentum can compensate for small batch sizes through a batch–momentum trade-off.
- How variance reduction can fully restore full-batch implicit bias regardless of batch size.
- A unified framework covering SignSGD, Normalized-SGD, Muon, and their variants under entry-wise and Schatten-p norms.

Finally, they provide a toy example of single-sample gradient steps, showing that it converges to a completely different solution from full-batch. This suggests that the implicit bias of mini-batch steepest descent methods is not uniform and depends on the batch size.

**Compliance With Llm Reviewing Policy:**

Affirmed.

**Final Justification:**

The authors addressed my main concerns and will add the discussion from the rebuttal to the paper.

**Key Questions For Authors:**

1. Is it possible to generalize the results to datasets that are not linearly separable?
2. Practically, we use very small batch sizes compared to the number of data points. Is there a way to analyze this setting as well?
3. Can we extend the analysis to sampling without replacement?

**Limitations:**

Yes

**Strengths And Weaknesses:**

**Strengths:**
- The paper provides a unified framework for analyzing the implicit bias of stochastic steepest methods, covering SignSGD, Normalized-SGD, Muon, and their variants using different techniques like momentum and variance reduction.
- The result on variance reduction is new, and the fact that SVRG-style correction recovers the exact full-batch max-margin solution for any batch size is interesting.
- Removing the dependence on the dimensionality of the problem is a solid theoretical contribution that improves over previous works.
- The writing is clear and easy to follow.


**Weaknesses:**
- In the introduction, the motivation is to understand optimizers like AdamW or Muon, which are relevant for large-scale language model training. However, the theory only holds for linear models and linearly separable data. Although it is acceptable to study simplified settings in theory, providing real-world experiments that align with the findings would strengthen the paper.
- The batch size condition in Theorem 4.1 seems very strong. In the case of small $\gamma$, the condition is roughly $b \approx n$, which basically means this is not a stochastic optimization problem anymore. Could the authors elaborate more on this condition and whether it can be relaxed?
- The authors show that in a specially designed case of batch size one, steepest descent methods converge to a fundamentally different solution. However, this setting cannot be recovered by Theorem 4.1 due to the constraint on batch size. It seems the analysis is not yet completely tight enough to cover these edge-case scenarios.

---

> ### Author Rebuttal · Authors · 2026-03-30
>
> We sincerely thank you for your insightful feedback and thoughtful suggestions on our work. Here, we address your comments/suggestions below:
>
> ---
> **Linear model & separable-data limitation:** Our analysis follows the standard setting in the implicit bias literature, which focuses on linear models with linearly separable data, where max-margin characterization is well-defined. Even in this classical setting, the implicit bias of stochastic algorithms is not yet fully understood, leaving important gaps in how stochasticity affects the limiting solutions.
>
> To address this, and to enable a fair comparison, we focus on isolating how stochastic optimization, including batch size, momentum, and variance reduction, shapes implicit bias under normalized steepest descent (NSD) with different geometries. To our knowledge, this is the first work that systematically characterizes how these factors affect convergence toward or deviation from max-margin solutions. We view this as a necessary step before tackling more complex settings.
>
> Regarding non-separable data, [1] shows that GD admits a decomposition into a separable component (yielding a max-margin direction) and a strongly convex component (yielding a finite offset), leading to convergence along a ray. Extending our analysis to incorporate this decomposition is a natural and promising direction for future work.
>
> Regarding real-world experiments, we further validate our findings on a two-layer ReLU NN trained on MNIST, following the setup of [2]. We observe qualitatively consistent behaviors with our theoretical predictions. Due to space limitations, detailed results are provided in [link](https://anonymous.4open.science/r/implicit_bias_real_data-57C5/real_data.png). We will include these results and further details in the final version.
>
> [1] Ji and Telgarsky. "The implicit bias of gradient descent on nonseparable data", 2019.\
> [2] Fan et al., "Implicit bias of spectral descent...", 2025.
>
> ---
> **Strong batch-size condition:** We agree that the batch-size requirement in Theorem 4.1 can be strong especially when the max-margin $\gamma$ is small and we do not claim it is tight. Importantly, this condition is not merely a proof artifact, but reflects an inherent difficulty of stochastic steepest descent without additional stabilization. Specifically, it arises as a sufficient condition ensuring a positive effective margin $\rho$, which is needed to overcome the alignment mismatch between the full gradient and the mini-batch-induced steepest direction $\langle \nabla L(W_t), \Delta_t \rangle$ and to enter the monotone loss-decreasing regime.
>
> Following a similar setup as in our response to Reviewer rYD4, we add a new experiment on an extremely hard separable dataset with $\gamma \approx 0$ under $\ell_{\infty}$ steepest descent (n = 200, 120k iterations). We observe that b=50 fails to classify all samples, while b=100 succeeds in classifying all samples, yet still stays noticeably away from the full-batch max-margin solution (see curves in [link](https://anonymous.4open.science/r/ICML_implicit_bias-hard-data-batch-condition-DD6E/hard_linf_dif_batchsize.png)). This suggests that insufficient batch sizes can lead to qualitative deviations from the full-batch regime, indicating the condition reflects a real stability requirement, rather than merely a proof artifact. We believe this condition may be further relaxed with a sharper, potentially data-dependent analysis, but leave this for future work.
>
> ---
> **Small-batch regime with distinct behavior:** Regarding small batch sizes used in practice, our batch-size-one analysis shows that the limiting direction can fundamentally differ from the full-batch max-margin solution, being driven by sample-wise effects rather than max-margin geometry. This highlights that the small-batch regime can exhibit fundamentally different behavior. We would also like to clarify that this batch-size-one case is not intended to be covered by Theorem 4.1; rather, it serves to illustrate a genuinely different asymptotic regime. We agree that characterizing the limiting behavior of $W_t$ for general datasets under very small batch sizes remains an important and interesting open problem for future work.
>
> ---
> **Sampling scheme clarification / extension:** We would like to clarify that our analysis already focuses on sampling without replacement (random reshuffling), as stated in Sec. 3.1. This sampling scheme plays a key role in our analysis through the epoch-wise zero-sum property: $\sum_{k=0}^{m-1} \Big(\nabla L_{B_{tm+k}}(W)-\nabla L(W)\Big)=0,$ which allows us to control the cumulative stochastic deviation over each epoch and is crucial in establishing the alignment and margin growth results. Extending to the with-replacement setting does not directly enjoy this cancellation property, and would likely require different tools. We leave this as an interesting direction for future work.

---

> > ### Author Rebuttal · Reviewer_WgdY · 2026-04-04
> >
> > Thanks for your reply. I am happy to keep my positive score.

---

> > > ### Author Response · Authors · 2026-04-04
> > >
> > > Dear Reviewer WgdY,
> > >
> > > We are glad to hear that our rebuttal has addressed your concerns, and we sincerely appreciate your positive evaluation of our work. In particular, we appreciate your comments on real-world experiments and batch-size conditions. We will include these additional experiments and clarifications in the revised manuscript.
> > >
> > > Thank you again for your effort in reviewing our work.
> > >
> > > Best,\
> > > Authors

---

### Official Review · Reviewer_rYD4 · 2026-03-15

**Soundness:** 3
**Presentation:** 2
**Significance:** 2
**Originality:** 2
**Overall Recommendation:** 4
**Confidence:** 3

**Summary:**

This paper study stochastic steepest descent with and without momentum and its variance reduction version. The authors show the implicit bias results for the mini-batch version of these algorithms, implemented in multi-class classification. Compared to Fan et. al. (2025), the analysis improves the dependence on the dimension, which originates from a different way of bounding a gradient difference term. For the variance reduction case, theoretical results recover the exact full-batch implicit bias, but the convergence rate is slower than that of standard stochastic methods without variance reduction.

**Compliance With Llm Reviewing Policy:**

Affirmed.

**Final Justification:**

The rebuttal addressed my questions, and I suggest that the authors include the additional discussions in their next version.

**Key Questions For Authors:**

Please see my questions above about stochastic literature & variance reduction.

**Limitations:**

The authors explained the limitations & societal impact.

**Strengths And Weaknesses:**

The majority of the theoretical setting is similar to the reference Fan et. al. (2025) as they consider the same problem, similar algorithms, same type of implicit bias results for normalized steepest descent (NSD) and momentum steepest descent (NMD). The difference is the minibatch SGD vs full batch GD, and the authors demonstrated large batch results for the version without momentum, where small batch applies for momentum and variance reduction.

While the idea is not new, the contributions of this paper are good. The theoretical results seem solid. While I did not check the proofs, they seem to be correct. The paper is well written.

While the authors compared their work with normalized steepest descent and momentum steepest descent (deterministic/full batch version), the prior literature on implicit bias on the stochastic/mini-batch version is not very clear. Similar questions apply to variance reduction methods. The authors should address this. There have been many prior works on implicit bias for SGD, but I am not sure how these works connect to this paper's setting.

Another weakness is the worse convergence rate for variance reduction methods, which is unusual. I am not sure if this reflects the actual rate of the algorithms. Does the empirical experiment support the theoretical rate? Or is it possible to obtain a better rate?

I am happy to revise the review if the authors can address these questions.

---

> ### Author Rebuttal · Authors · 2026-03-30
>
> We sincerely thank you for your insightful feedback and thoughtful suggestions on our work. Here, we address your comments/suggestions below:
>
> ----
> ### insufficient discussion of prior work
> Prior works on the implicit bias of stochastic algorithms are indeed broad. Due to space constraints, here we focus on the most relevant line studying linear separable settings with max-margin characterization. [1] shows that mini-batch SGD converges to the $\ell_2$ max-margin direction under a sufficiently small stepsize controlling stochastic noise (depending on batch size). [2] further shows that SGD with momentum preserves the same $\ell_2$ max-margin direction under a small constant stepsize depending on both batch size and momentum. [3] establishes a similar max-margin result for stochastic AdaGrad-Norm under general mini-batch noise assumption. [4] shows that per-sample Adam can induce an implicit bias that differs from its full-batch counterpart and that Signum exhibits similar behavior in our paper under a restrictive fixed mini-batch cycle. In contrast, our work provides a unified analysis of stochastic normalized steepest descent (NSD) under general geometries and is, to the best of our knowledge, the first to systematically characterize how multiple stochastic factors, including batch size, momentum, and variance reduction, interact to shape the implicit bias.
>
> Regarding variance reduction methods, they have been extensively studied in stochastic optimization for reducing gradient variance and accelerating convergence (e.g., [5-7]). In this work, we focus on an SVRG-style estimator[6]. To the best of our knowledge, its role in shaping implicit bias has not been explicitly studied. We show that variance reduction recovers the exact full-batch implicit bias regardless of batch size and momentum, while it may lead to a slower margin convergence rate.
>
> Due to limited rebuttal space, we have focused on the most relevant works here, and will include a more comprehensive discussion in the final version.
>
> [1] Nacson et al., "Stochastic gradient descent on separable data...", 2019.\
> [2] Wang et al., "Momentum doesn't change the implicit bias", 2021.\
> [3] Jin et al., "The Implicit Bias of Stochastic AdaGrad-Norm on Separable Data", 2024.\
> [4] Baek et al., "Implicit Bias of Per-sample Adam on Separable Data...", 2025.\
> [5] Le Roux et al., "A stochastic gradient method with an exponential convergence...", 2012.\
> [6] Johnson & Zhang, "Accelerating stochastic gradient descent using predictive...", 2013.\
> [7] Defazio et al., "SAGA: A fast incremental gradient method...", 2014.
>
> ---
> ### convergence rate for variance reduction methods
> We clarify that the convergence rate studied in our paper is the margin convergence rate, rather than the standard optimization convergence rates (e.g., objective decrease or stationarity). Therefore, it is not contradictory that variance reduction accelerates these optimization metrics, while it may lead to slower convergence (compared with full-batch) in terms of identifying the max-margin direction.
>
> Our analysis actually provides a **worst-case upper bound** on the margin convergence rate so the slower rate for variance reduction should be viewed as a conservative guarantee, rather than a tight or universal characterization. We will clarify this point in the final version.
>
> To empirically validate this, we add a new experiment using a similar setup as in the paper but on a **harder separable dataset** (K=10, n=200, d=10), where only a small number of support-like samples $x_i$ lie close to the decision boundary while the majority are well separated. In this setting, the max-margin direction is determined by these rare hard samples, making margin identification more sensitive. We focus on the $\ell_2$ normalized-SGD setting for clarity; we observe qualitatively similar behavior under other geometries, but their curves are more numerically sensitive and less clean to present. Full experimental details and convergence curves are provided here: [link](https://anonymous.4open.science/r/implicit_bias_VR_slower-B264/VR_slower.png). We report the number of iterations required to reach target relative margin errors $\frac{|\gamma^\ast - \gamma(W_t)|}{\gamma^\ast} \in \{0.15, 0.10, 0.05\}$:
> |Method|≤ 0.15 |≤ 0.10 |≤ 0.05 |
> |-|-|-|-|
> | Full-batch N-SGD (n=200)|1300|2500|7950|
> | VR N-SGD (b=5) |2400|4250|12000|
> | Mini-batch N-SGD (b=5)|not reached|not reached|not reached|
>
> On this hard instance, VR requires more iterations than full-batch N-SGD to reach the same margin accuracy, while still converging to the same max-margin solution. This is **qualitatively consistent** with our theoretical result that VR may exhibit a more conservative margin convergence behavior as it acts as a stabilization mechanism towards max-margin direction.
>
> Finally, obtaining a sharper rate for VR remains an important open question. Our analysis does not rule out improved bounds, which we leave for future work.

---

> > ### Author Rebuttal · Reviewer_rYD4 · 2026-04-05
> >
> > Thank you for the rebuttal, I will change my score. Please include the additional discussions in the next version of your paper.

---

> > > ### Author Response · Authors · 2026-04-05
> > >
> > > Dear Reviewer rYD4,
> > >
> > > Thank you for your encouraging feedback and for updating your evaluation. We are glad that our rebuttal has addressed your concerns, and we will incorporate the additional discussions of prior work and clarifications on the variance-reduction convergence rate into the revised manuscript as suggested.
> > >
> > > Thank you again for your effort in reviewing our work.
> > >
> > > Best,\
> > > Authors

---

### Decision · Program_Chairs · 2026-04-30

**Decision:**

Accept (regular)

**Comment:**

The paper studies stochastic steepest descent with and without momentum, as well as its variance-reduced version. This topic is fundamental and deserves a deep studies.  As all reviewers find the paper interesting, I recommend acceptance.